# Zeroth-Order Optimization at the Edge of Stability

**Minhak Song** [1 2]  **Liang Zhang** [3 4]  **Bingcong Li** [3]  **Niao He** [3]  **Michael Muehlebach** [4]  **Sewoong Oh** [5]

## Abstract

Zeroth-order (ZO) methods are widely used when gradients are unavailable or prohibitively expensive, including black-box learning and memory-efficient fine-tuning of large models, yet their optimization dynamics in deep learning remain underexplored. In this work, we provide an explicit step size condition that exactly captures the (mean-square) linear stability of a family of ZO methods based on the standard two-point estimator. Our characterization reveals a sharp contrast with first-order (FO) methods: whereas FO stability is governed solely by the largest Hessian eigenvalue, mean-square stability of ZO methods depends on the entire Hessian spectrum. Since computing the full Hessian spectrum is infeasible in practical neural network training, we further derive tractable stability bounds that depend only on the largest eigenvalue and the Hessian trace. Empirically, we find that full-batch ZO methods operate at the *edge of stability*: ZO-GD, ZO-GDM, and ZO-Adam consistently stabilize near the predicted stability boundary across a range of deep learning training problems. Our results highlight an implicit regularization effect specific to ZO methods, where large step sizes primarily regularize the Hessian trace, whereas in FO methods they regularize the top eigenvalue.

## 1. Introduction

Zeroth-order (ZO) optimization methods, which rely only on function evaluations, are widely used when gradients are unavailable, unreliable, or expensive to compute. Such a setting arises in black-box learning, derivative-free control, and increasingly in modern large-model pipelines, where memory and systems constraints can make backpropagation

[1]KAIST [2]KRAFTON [3]ETH Zurich [4]Max Planck Institute for Intelligent Systems [5]University of Washington. Correspondence to: Minhak Song <minhaksong@kaist.ac.kr>.

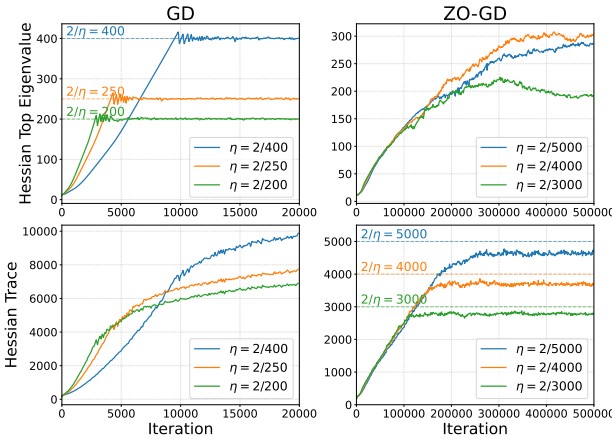

*Figure 1.* **EoS behaviors of FO and ZO methods are captured by different spectral quantities of the Hessian.** We train full-batch GD (left) and ZO-GD (right) with varying step sizes $\eta$ on a CNN for CIFAR-10. For GD, the largest eigenvalue of the Hessian $\lambda_{\max}(\boldsymbol{H}_t)$ stabilizes near $2/\eta$. For ZO-GD, the trace of the Hessian $\mathrm{Tr}(\boldsymbol{H}_t)$ instead stabilizes slightly below $2/\eta$.

costly. Recent work shows that ZO methods based on two-point function evaluations can fine-tune large language models (LLMs) with competitive accuracy while substantially reducing memory usage and compute overhead (Malladi et al., 2023; Zhang et al., 2024b). Despite their growing practical relevance, the training dynamics of ZO methods in deep learning remain far less understood than those of first-order (FO) optimizers.

For FO optimization in deep learning, one prominent empirical phenomenon is the *edge of stability* (EoS). In full-batch gradient descent (GD) with step size $\eta$ on a quadratic objective $f_{\mathrm{quad}}(\boldsymbol{x}) = \frac{1}{2}\boldsymbol{x}^\top \boldsymbol{H}\boldsymbol{x}$, the iterates diverge when $\eta > 2/\lambda_{\max}(\boldsymbol{H})$. In neural network training, however, optimization often remains well-behaved even at step sizes beyond this local quadratic threshold. Along the training trajectory $\{\boldsymbol{x}_t\}$, the top eigenvalue of the Hessian $\lambda_{\max}(\boldsymbol{H}_t)$ increases early in training and then stabilizes near the threshold $2/\eta$ over long horizons (Cohen et al., 2021). This phenomenon has motivated a growing line of work aimed at understanding its mechanism and its connections to stability, curvature, and implicit regularization in deep learning optimization dynamics (Ahn et al., 2022; Arora et al., 2022; Damian et al., 2023; Cohen et al., 2025).

In this work, we ask whether a similar phenomenon arises

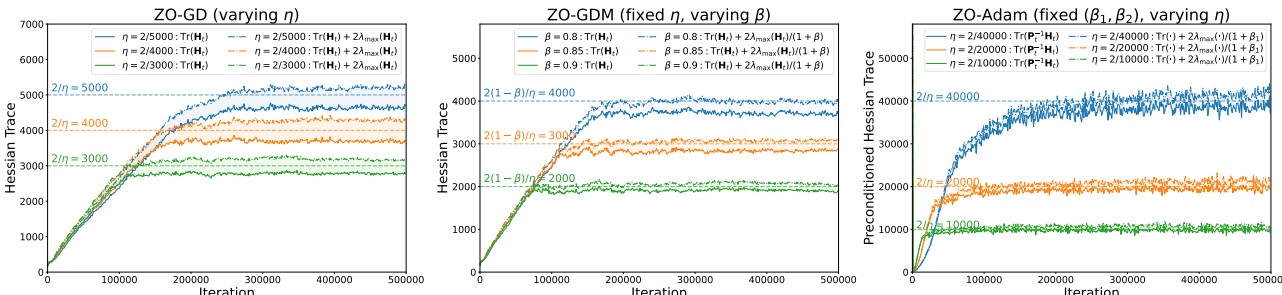

*Figure 2.* **Zeroth-order methods operate at the mean-square edge of stability.** We train full-batch ZO methods on a CNN for CIFAR-10 and track the curvature terms defining the mean-square stability interval from Section 4.2. Across all panels, each color denotes one run; the solid curve is the lower-band term, the dash-dotted curve is the upper-band term, and the dashed line is the predicted stability threshold. **Left** (ZO-GD, varying $\eta$): lower $\text{Tr}(\boldsymbol{H}_t)$, upper $\text{Tr}(\boldsymbol{H}_t) + 2\lambda_{\max}(\boldsymbol{H}_t)$, threshold $2/\eta$ in Eq. (5). **Middle** (ZO-GDM, varying $\beta$ with fixed $\eta = 10^{-4}$): lower $\text{Tr}(\boldsymbol{H}_t)$, upper $\text{Tr}(\boldsymbol{H}_t) + \frac{2}{1+\beta}\lambda_{\max}(\boldsymbol{H}_t)$, threshold $2(1-\beta)/\eta$ in Eq. (6). **Right** (ZO-Adam, varying $\eta$ with fixed $(\beta_1, \beta_2) = (0.9, 0.999)$): lower $\text{Tr}(\boldsymbol{P}_t^{-1}\boldsymbol{H}_t)$, upper $\text{Tr}(\boldsymbol{P}_t^{-1}\boldsymbol{H}_t) + \frac{2}{1+\beta_1}\lambda_{\max}(\boldsymbol{P}_t^{-1}\boldsymbol{H}_t)$, threshold $2/\eta$ in Eq. (7). Across all methods, the threshold stays within (or very close to) the stability interval throughout training, indicating mean-square EoS behavior.

in *zeroth-order* training. At first glance, this is far from obvious: ZO methods update parameters by drawing random search directions at every iteration, and their stability cannot be understood through the deterministic arguments typically used to explain FO dynamics and stability.

As a preliminary experiment, Figure 1 compares full-batch (first-order) GD and zeroth-order gradient descent (ZO-GD) with varying step sizes $\eta$. For GD, the top Hessian eigenvalue $\lambda_{\max}(\boldsymbol{H}_t)$ stabilizes near $2/\eta$, consistent with the behavior predicted by standard EoS theory. For ZO-GD, however, $\lambda_{\max}(\boldsymbol{H}_t)$ does not exhibit the same trend; instead, perhaps surprisingly, the Hessian trace $\text{Tr}(\boldsymbol{H}_t)$ stabilizes slightly below $2/\eta$. This suggests that ZO training may be governed by a different stability mechanism than FO training, and that ZO methods may exhibit their own EoS phenomenon in deep learning.

These observations motivate the following fundamental questions: ($i$) *do ZO methods operate at the edge of stability in neural network training*, and ($ii$) if so, *which curvature-related quantity governs their stability*?

We introduce a mean-square linear stability theory for ZO methods to investigate these questions. In Section 4, we provide an exact step size characterization of mean-square linear stability under the linearized dynamics for a family of ZO optimizers, including ZO-GD and its momentum and preconditioned variants. Unlike FO methods, whose stability is governed solely by the largest Hessian eigenvalue, mean-square stability of ZO methods depends on the entire Hessian spectrum. Moreover, momentum affects stability in opposite ways: increasing $\beta$ enlarges the stable regime for GD with momentum (GDM), but shrinks it for ZO-GD with momentum (ZO-GDM). Since computing the full spectrum is typically infeasible, we also derive tractable bounds that depend only on the Hessian trace and the largest eigenvalue (summarized in Table 1).

Technically, our analysis introduces a cone-preserving linear covariance operator with a rank-one global coupling term to capture the dynamics of the ZO second-moment recursion. This reduction allows one to characterize its spectral radius using the Krein–Rutman Theorem and derive the explicit mean-square stability conditions.

In Section 5, we empirically find that full-batch ZO methods operate at the *mean-square* edge of stability across architectures and tasks: ZO methods (ZO-GD, ZO-GDM, and ZO-Adam) consistently stabilize near the predicted mean-square stability boundary. Notably, this behavior is governed primarily by trace-based curvature quantities, providing new insight into how large step sizes implicitly bias ZO training toward solutions with small (preconditioned) Hessian trace.

## 2. Related work

**Zeroth-order optimization.** ZO methods optimize objectives using only function evaluations, typically via the standard two-point estimator along a random direction. Empirically, Malladi et al. (2023) report that ZO methods can fine-tune LLMs with performance comparable to FO methods while achieving up to $12\times$ memory and up to $2\times$ GPU-hour reduction, highlighting ZO optimization as a promising approach for memory-efficient fine-tuning. ZO methods have also been applied to adversarial robustness (Chen et al., 2017; Andriushchenko et al., 2020), reinforcement learning (Salimans et al., 2017), private fine-tuning (Zhang et al., 2024a) and distributed learning (Fang et al., 2022; Qin et al., 2024; Xu et al., 2024), where gradients may be noisy, unavailable, or expensive to compute or communicate.

On the theory side, most prior work studies ZO methods through the lens of classical (non)convex optimization (Jamieson et al., 2012; Duchi et al., 2015; Shamir, 2017; Wang et al., 2018; Malladi et al., 2023; Zhang et al., 2024a), emphasizing convergence under small step sizes

*Table 1.* Summary of linear stability conditions for FO and ZO methods under the linearized dynamics (Definition 1). For FO methods of GD, GDM, and Adam, we report the exact critical step size $\eta^\star$, and the mean and mean-square thresholds coincide. For ZO methods, we report lower and upper bounds on the mean-square critical step size $\eta^\star_{\mathrm{ms}}$; the critical step size in the mean, $\eta^\star_{\mathrm{mean}}$, matches the corresponding FO threshold. For Adam and ZO-Adam, the reported conditions correspond to the *frozen-preconditioner* variants and are governed by the spectrum of the preconditioned Hessian $\boldsymbol{P}^{-1}\boldsymbol{H}$. See Section 4 for details.

| Method | Linear Stability Condition |
|---|---|
| GD | $\eta^\star_{\mathrm{mean}} = \eta^\star_{\mathrm{ms}} = \dfrac{2}{\lambda_{\max}(\boldsymbol{H})}$ |
| GDM (Cohen et al., 2021) | $\eta^\star_{\mathrm{mean}} = \eta^\star_{\mathrm{ms}} = \dfrac{2(1+\beta)}{\lambda_{\max}(\boldsymbol{H})}$ |
| Adam (Cohen et al., 2022) | $\eta^\star_{\mathrm{mean}} = \eta^\star_{\mathrm{ms}} = \dfrac{2(1+\beta_1)}{(1-\beta_1)\lambda_{\max}(\boldsymbol{P}^{-1}\boldsymbol{H})}$ |
| ZO-GD (Theorem 1) | $\eta^\star_{\mathrm{ms}} \leqslant \dfrac{2}{\mathrm{Tr}(\boldsymbol{H})}$ 

 $\eta^\star_{\mathrm{ms}} \geqslant \dfrac{2}{\mathrm{Tr}(\boldsymbol{H}) + 2\lambda_{\max}(\boldsymbol{H})}$ |
| ZO-GDM (Theorem 2) | $\eta^\star_{\mathrm{ms}} \leqslant \dfrac{2(1-\beta)}{\mathrm{Tr}(\boldsymbol{H})}$ 

 $\eta^\star_{\mathrm{ms}} \geqslant \dfrac{2(1-\beta)}{\mathrm{Tr}(\boldsymbol{H}) + \frac{2\lambda_{\max}(\boldsymbol{H})}{1+\beta}}$ |
| ZO-Adam (Theorem 3) | $\eta^\star_{\mathrm{ms}} \leqslant \dfrac{2}{\mathrm{Tr}(\boldsymbol{P}^{-1}\boldsymbol{H})}$ 

 $\eta^\star_{\mathrm{ms}} \geqslant \dfrac{2}{\mathrm{Tr}(\boldsymbol{P}^{-1}\boldsymbol{H}) + \frac{2\lambda_{\max}(\boldsymbol{P}^{-1}\boldsymbol{H})}{1+\beta_1}}$ |

and smoothness assumptions. Notably, Zhang et al. (2025a) study the implicit bias of ZO-GD under smooth convex objectives and show that it favors solutions with small Hessian trace.

**Edge of stability.** Recent empirical work shows that FO methods often train near instability. In particular, Cohen et al. (2021) identify the edge of stability (EoS) in full-batch GD, where $\lambda_{\max}(\boldsymbol{H}_t)$ grows early in training and then equilibrates near $2/\eta$. This observation has motivated extensive follow-up work on the mechanisms and implications of EoS (Ahn et al., 2022; Arora et al., 2022; Wang et al., 2022; Damian et al., 2023; Song & Yun, 2023; Zhu et al., 2023; Yoo et al., 2025; Cohen et al., 2025). EoS-type behavior has also been studied for momentum and adaptive optimizers (Cohen et al., 2022), mini-batch stochastic gradient descent (SGD) (Lee & Jang, 2023; Andreyev & Beneventano, 2024), and other families of FO optimizers, including sharpness-aware minimization (Foret et al., 2021; Long & Bartlett, 2024) and schedule-free methods (Defazio et al., 2024; Song et al., 2025).

**Dynamical stability analysis of optimizers.** A growing body of work studies optimization methods through the lens of dynamical stability. Recent work analyzes *linear stability* by examining the behavior of the linearized dynamics

under a local quadratic approximation, extending beyond GD to momentum methods (Muehlebach & Jordan, 2021) and stochastic optimization. For SGD, prior analyses have characterized linear stability condition in the mean-square sense (Wu et al., 2018; Granziol et al., 2022; Velikanov et al., 2023), for higher moments (Ma & Ying, 2021), and in probability (Ziyin et al., 2023). Most closely related to our setting, Mulayoff & Michaeli (2024) derive the exact mean-square stability threshold of mini-batch SGD and show that it is monotonically non-decreasing in the batch size. Our work also studies mean-square linear stability, but for ZO methods, where stochasticity arises from the estimator directions even under full-batch training.

## 3. Preliminaries

We consider the optimization problem

$$\min_{\boldsymbol{x}\in\mathbb{R}^d} \ f(\boldsymbol{x}) \ ,$$

for a loss function $f : \mathbb{R}^d \to \mathbb{R}$ and a model parameter $\boldsymbol{x}$, and study the dynamics of zeroth-order (ZO) optimization methods. ZO methods iteratively update a sequence of iterates $\{\boldsymbol{x}_t\}_{t\geqslant 0}$ based solely on function evaluations without evaluating any gradients. We analyze the three most popular types of ZO methods: ZO-GD, ZO-GDM, and ZO-Adam.

**ZO Gradient Descent (ZO-GD)** replaces the gradient $\nabla f(\boldsymbol{x}_t)$ in the GD update with a gradient estimate $\widehat{\nabla} f(\boldsymbol{x}_t)$:

$$\boldsymbol{x}_{t+1} \ = \ \boldsymbol{x}_t - \eta\, \widehat{\nabla} f(\boldsymbol{x}_t) \ ,$$

for a step size $\eta > 0$. We consider the standard two-point estimator (Nesterov & Spokoiny, 2017), defined as

$$\widehat{\nabla} f(\boldsymbol{x}_t) \ := \ \frac{f(\boldsymbol{x}_t + \mu\boldsymbol{u}_t) - f(\boldsymbol{x}_t - \mu\boldsymbol{u}_t)}{2\mu} \cdot \boldsymbol{u}_t \ , \quad (1)$$

where $\boldsymbol{u}_t \overset{\text{i.i.d.}}{\sim} \mathcal{N}(\boldsymbol{0}, \boldsymbol{I})$ and $\mu > 0$ is a smoothing parameter. In general, $\widehat{\nabla} f(\boldsymbol{x}_t)$ is a biased estimator of $\nabla f(\boldsymbol{x}_t)$, with bias that vanishes as $\mu \to 0$.

**ZO Gradient Descent with Momentum (ZO-GDM)** combines Polyak momentum terms with gradient estimates:

$$\boldsymbol{m}_{t+1} = \beta\boldsymbol{m}_t + \widehat{\nabla} f(\boldsymbol{x}_t) \ ,$$
$$\boldsymbol{x}_{t+1} = \boldsymbol{x}_t - \eta\boldsymbol{m}_{t+1} \ ,$$

for a step size $\eta > 0$ and a momentum parameter $\beta \in [0,1)$.

**ZO-Adam** combines adaptive moment estimation (Adam) with gradient estimates:

$$\boldsymbol{m}_{t+1} = \beta_1\boldsymbol{m}_t + (1-\beta_1)\widehat{\nabla} f(\boldsymbol{x}_t) \ ,$$
$$\boldsymbol{x}_{t+1} = \boldsymbol{x}_t - \eta\boldsymbol{P}_{t+1}^{-1}\boldsymbol{m}_{t+1} \ ,$$

for step size $\eta > 0$, momentum parameters $\beta_1, \beta_2 \in [0, 1)$, and $\epsilon > 0$, where $\boldsymbol{P}_{t+1}$ is a preconditioner defined using an exponential moving average (EMA) of squared (estimated) gradients:

$$\boldsymbol{\nu}_{t+1} = \beta_2 \boldsymbol{\nu}_t + (1 - \beta_2) \widehat{\nabla} f(\boldsymbol{x}_t) \odot \widehat{\nabla} f(\boldsymbol{x}_t) \,,$$

$$\boldsymbol{P}_{t+1} = (1 - \beta_1^{t+1}) \left[ \mathrm{diag}\left( \sqrt{\frac{\boldsymbol{\nu}_{t+1}}{1 - \beta_2^{t+1}}} \right) + \epsilon \boldsymbol{I} \right] \,,$$

and $\odot$ denotes element-wise multiplication. This form is equivalent to the standard bias-corrected Adam update, written as a preconditioned momentum step.

In practical neural network training, the short-term stability behavior of Adam is often well approximated by a *frozen-preconditioner* variant, in which the preconditioner is held fixed at its current value (Cohen et al., 2022). Motivated by this, we also consider the corresponding ZO analogue.

**Frozen ZO-Adam** uses a fixed preconditioner $\boldsymbol{P} > 0$ with a step size $\eta > 0$:

$$\boldsymbol{m}_{t+1} = \beta_1 \boldsymbol{m}_t + (1 - \beta_1) \widehat{\nabla} f(\boldsymbol{x}_t) \,,$$

$$\boldsymbol{x}_{t+1} = \boldsymbol{x}_t - \eta \boldsymbol{P}^{-1} \boldsymbol{m}_{t+1} \,,$$

and a momentum parameter $\beta_1 \in [0, 1)$. This optimizer is introduced purely for theoretical analysis and serves as the ZO analogue of *Frozen Adam*, which was used to study the *adaptive edge of stability* of Adam (Cohen et al., 2022).

## 4. Linear stability analysis

Directly analyzing the full dynamics of optimizers in deep learning is typically intractable. Instead, we adopt the standard dynamical systems approach of studying stability near a local minimizer via the *linearized dynamics*. Exponential stability of the linearized dynamics implies local stability of the corresponding nonlinear dynamics near the equilibrium, which justifies linear stability analysis as a principled tool (Khalil, 2002).

**Definition 1** (Linearized dynamics). Let $\boldsymbol{x}^\star$ be a local minimizer of $f$, and assume that $f$ is twice differentiable in a neighborhood of $\boldsymbol{x}^\star$. Let $\boldsymbol{H} := \nabla^2 f(\boldsymbol{x}^\star)$ be the Hessian at $\boldsymbol{x}^\star$. The *linearized dynamics* of an optimizer around $\boldsymbol{x}^\star$ are the dynamics obtained by applying the optimizer to the quadratic Taylor approximation of $f$ at $\boldsymbol{x}^\star$,

$$f_{\mathrm{quad}}(\boldsymbol{x}) := f(\boldsymbol{x}^\star) + \frac{1}{2}(\boldsymbol{x} - \boldsymbol{x}^\star)^\top \boldsymbol{H}(\boldsymbol{x} - \boldsymbol{x}^\star) \,.$$

Throughout, we assume $\boldsymbol{H} \succeq 0$ and $\boldsymbol{H} \neq \boldsymbol{0}$. Let $\lambda_1 \geqslant \cdots \geqslant \lambda_d \geqslant 0$ denote the eigenvalues of $\boldsymbol{H}$, and define $\lambda_{\max}(\boldsymbol{H}) := \lambda_1$ and $\mathrm{Tr}(\boldsymbol{H}) := \sum_{i=1}^d \lambda_i$.

We next formalize the notion of linear stability for the resulting (possibly stochastic) optimizer dynamics.

**Definition 2** (Linear stability). Let $\boldsymbol{x}^\star$ be as in Definition 1, and let $\{\boldsymbol{x}_t\}_{t \geqslant 0}$ denote the iterates generated by an optimizer applied to $f_{\mathrm{quad}}$.

We say the optimizer is *linearly stable in the mean* if

$$\sup_{t \geqslant 0} \|\mathbb{E}[\boldsymbol{x}_t - \boldsymbol{x}^\star]\| < \infty \quad \text{for every initialization } \boldsymbol{x}_0 \in \mathbb{R}^d.$$

We say the optimizer is *mean-square linearly stable* if

$$\sup_{t \geqslant 0} \mathbb{E}\big[\|\boldsymbol{x}_t - \boldsymbol{x}^\star\|^2\big] < \infty \quad \text{for every initialization } \boldsymbol{x}_0 \in \mathbb{R}^d.$$

Mean-square linear stability implies linear stability in the mean by Jensen's inequality. For deterministic optimizers, the expectation is redundant, and the two notions coincide.

Our goal is to theoretically characterize the *critical step size* that guarantees linear stability of ZO methods and empirically connect it to the edge of stability phenomena.

**Definition 3** (Critical step size). Consider an optimizer with step size $\eta > 0$ applied to $f_{\mathrm{quad}}$, producing iterates $\{\boldsymbol{x}_t\}_{t \geqslant 0}$. The *critical step size in the mean* is defined as

$$\eta_{\mathrm{mean}}^\star := \sup \left\{ \eta > 0 : \forall \boldsymbol{x}_0 \in \mathbb{R}^d, \sup_{t \geqslant 0} \|\mathbb{E}[\boldsymbol{x}_t - \boldsymbol{x}^\star]\| < \infty \right\},$$

and the *critical step size in the mean-square* is defined as

$$\eta_{\mathrm{ms}}^\star := \sup \left\{ \eta > 0 : \forall \boldsymbol{x}_0 \in \mathbb{R}^d, \sup_{t \geqslant 0} \mathbb{E}\big[\|\boldsymbol{x}_t - \boldsymbol{x}^\star\|^2\big] < \infty \right\}.$$

Next, we review known stability thresholds for FO methods (Section 4.1) and present our main results for ZO methods (Section 4.2). All proofs are deferred to Appendix A.

### 4.1. Linear stability of first-order methods

The first-order (FO) counterparts of the ZO optimizers in Section 3, namely GD, GDM, and Frozen Adam, are deterministic, so their mean and mean-square linear stability conditions coincide. A key takeaway is that FO linear stability is governed solely by the top eigenvalue of the (preconditioned) Hessian, as summarized below.

**Proposition 1** (Stability of GD). *The critical step size of GD is* $\eta_{\mathrm{mean}}^\star = \eta_{\mathrm{ms}}^\star = 2/\lambda_{\max}(\boldsymbol{H})$.

The stability condition of GD with Momentum (GDM) was established in Theorem 2 of Cohen et al. (2021).

**Proposition 2** (Stability of GDM). *The critical step size of GDM is* $\eta_{\mathrm{mean}}^\star = \eta_{\mathrm{ms}}^\star = 2(1 + \beta)/\lambda_{\max}(\boldsymbol{H})$.

The stability condition of Frozen Adam was established in Lemma 2 and Proposition 1 of Cohen et al. (2022).

**Proposition 3** (Stability of Frozen Adam). *The critical step size of Frozen Adam is*

$$\eta^\star_{\text{mean}} = \eta^\star_{\text{ms}} = \frac{2(1 + \beta_1)}{(1 - \beta_1)\lambda_{\max}(\boldsymbol{P}^{-1}\boldsymbol{H})} \ .$$

### 4.2. Linear stability of zeroth-order methods

The inherent stochasticity in ZO updates fundamentally changes the linear stability as shown in Figure 1. However, the *mean* dynamics of a ZO method matches that of its FO counterpart, and $\eta^\star_{\text{mean}}$ coincides with the FO critical step size presented in Section 4.1. This is due to the fact that under the quadratic model $f_{\text{quad}}$ in Definition 1, the two-point estimator is *unbiased*: $\mathbb{E}\big[\hat{\nabla} f_{\text{quad}}(\boldsymbol{x}_t)\big] = \nabla f_{\text{quad}}(\boldsymbol{x}_t)$. The intricacy of ZO linear stability is only captured by the *mean-square* stability. In particular, we show that $\eta^\star_{\text{ms}}$ for ZO-GD, ZO-GDM, and Frozen ZO-Adam, depends on the entire eigen spectrum of the (preconditioned) Hessian and is dominated by its trace value.

**Theorem 1** (Stability of ZO-GD). *For ZO-GD, $\eta^\star_{\text{mean}} = 2/\lambda_{\max}(\boldsymbol{H})$, and the mean-square critical step size $\eta^\star_{\text{ms}}$ is the unique $\eta > 0$ satisfying*

$$\eta\lambda_{\max}(\boldsymbol{H}) < 1 \quad \text{and} \quad \sum_{i=1}^{d} \frac{\eta\lambda_i}{2(1 - \eta\lambda_i)} = 1 \ ,$$

*which admits the bounds*

$$\frac{2}{\text{Tr}(\boldsymbol{H}) + 2\lambda_{\max}(\boldsymbol{H})} \leqslant \eta^\star_{\text{ms}} \leqslant \frac{2}{\text{Tr}(\boldsymbol{H})} \ . \tag{2}$$

*Remark* 1 (Computing $\eta^\star_{\text{ms}}$). If the full spectrum $\{\lambda_i\}_{i=1}^{d}$ were available, $\eta^\star_{\text{ms}}$ in Theorem 1 can be computed by solving $\sum_{i=1}^{d}(\eta\lambda_i/2(1 - \eta\lambda_i)) = 1$ over $\eta \in (0, 1/\lambda_{\max}(\boldsymbol{H}))$. In practice, we instead track the bounds (2), which depend only on $\text{Tr}(\boldsymbol{H})$ and $\lambda_{\max}(\boldsymbol{H})$ and can be estimated efficiently during training, which is critical for large models.

**Theorem 2** (Stability of ZO-GDM). *For ZO-GDM, $\eta^\star_{\text{mean}} = 2(1 + \beta)/\lambda_{\max}(\boldsymbol{H})$, and the mean-square critical step size $\eta^\star_{\text{ms}}$ is the unique $\eta > 0$ satisfying*

$$\eta\lambda_{\max}(\boldsymbol{H}) < 1 - \beta^2 \text{ and } \sum_{i=1}^{d} \frac{\eta\lambda_i}{2(1 - \beta)\big(1 - \frac{\eta\lambda_i}{1-\beta^2}\big)} = 1 \ ,$$

*which admits the bounds*

$$\frac{2(1 - \beta)}{\text{Tr}(\boldsymbol{H}) + \frac{2\lambda_{\max}(\boldsymbol{H})}{1+\beta}} \leqslant \eta^\star_{\text{ms}} \leqslant \frac{2(1 - \beta)}{\text{Tr}(\boldsymbol{H})} \ . \tag{3}$$

*Proof sketch.* We analyze the recursions of the second-moment matrices $\mathbb{E}[\boldsymbol{x}_t\boldsymbol{x}_t^\top]$, $\mathbb{E}[\boldsymbol{x}_t\boldsymbol{m}_t^\top]$, and $\mathbb{E}[\boldsymbol{m}_t\boldsymbol{m}_t^\top]$. Using the Isserlis' Theorem to evaluate Gaussian fourth moments, we obtain a linear recursion that can be expressed as

a cone-preserving linear operator on a product cone. Mean-square stability is then equivalent to this operator having spectral radius smaller than one. We characterize this spectral radius using Theorem 4 in Appendix A.2, which leverages the Krein–Rutman Theorem and leads to the explicit mean-square stability condition. □

**Theorem 3** (Stability of Frozen ZO-Adam). *Let $\tilde{\lambda}_1 \geqslant \tilde{\lambda}_2 \geqslant \cdots \geqslant \tilde{\lambda}_d \geqslant 0$ denote the eigenvalues of the preconditioned Hessian $\boldsymbol{P}^{-1}\boldsymbol{H}$. For Frozen ZO-Adam, $\eta^\star_{\text{mean}} = 2(1 + \beta_1)/\big((1 - \beta_1)\lambda_{\max}(\boldsymbol{P}^{-1}\boldsymbol{H})\big)$. Assuming $\boldsymbol{PH} = \boldsymbol{HP}$, the mean-square critical step size $\eta^\star_{\text{ms}}$ is the unique $\eta > 0$ satisfying*

$$\eta\lambda_{\max}(\boldsymbol{P}^{-1}\boldsymbol{H}) < 1 + \beta_1 \text{ and } \sum_{i=1}^{d} \frac{\eta\tilde{\lambda}_i}{2\big(1 - \frac{\eta\tilde{\lambda}_i}{1+\beta_1}\big)} = 1 \ ,$$

*and it admits the bounds*

$$\frac{2}{\text{Tr}(\boldsymbol{P}^{-1}\boldsymbol{H}) + \frac{2\lambda_{\max}(\boldsymbol{P}^{-1}\boldsymbol{H})}{1+\beta_1}} \leqslant \eta^\star_{\text{ms}} \leqslant \frac{2}{\text{Tr}(\boldsymbol{P}^{-1}\boldsymbol{H})}. \tag{4}$$

*Remark* 2 (Commutativity assumption). The condition $\boldsymbol{PH} = \boldsymbol{HP}$ ensures that $\boldsymbol{P}^{-1}\boldsymbol{H}$ is diagonalizable in the same eigenbasis as $\boldsymbol{H}$, which allows the mean-square dynamics to decouple across eigendirections and enables a tractable spectral analysis. Without commutativity, the second-moment recursion generally couples different eigenspaces, and obtaining an explicit stability characterization becomes substantially less tractable. Empirically, in neural network training with ZO-Adam, we observe that $\boldsymbol{P}_t$ and $\boldsymbol{H}_t$ are nearly commuting: the relative commutator Frobenius norm $\|\boldsymbol{P}_t\boldsymbol{H}_t - \boldsymbol{H}_t\boldsymbol{P}_t\|_F/\|\boldsymbol{P}_t\boldsymbol{H}_t\|_F$ decreases from 0.8–0.9 at initialization to below 0.05 and remains below 0.05 throughout training (see Appendix D.3).

Taken together, Theorems 1 to 3 provide exact mean-square stability characterizations under the linearized dynamics. In the next section, we use the corresponding upper and lower bounds in (2), (3), and (4) to empirically test whether ZO methods operate near the mean-square edge of stability during neural network training.

Our main theory focuses on the Gaussian symmetric two-point estimator in (1), which is the standard estimator used in MeZO-style LLM fine-tuning (Malladi et al., 2023). Appendix B shows that the same covariance-operator framework extends naturally to other estimator choices, including forward finite differences, non-Gaussian directions, and multi-query averages.

## 5. Zeroth-order optimization operates at the mean-square edge of stability

Building on the mean-square linear stability theory in Section 4.2, we empirically show that full-batch ZO methods on

neural networks operate at the *mean-square* edge of stability (EoS): the training dynamics stabilizes near the predicted mean-square linear stability boundary.

## 5.1. Tracking mean-square stability during training

Let $\boldsymbol{H}_t := \nabla^2 f(\boldsymbol{x}_t)$ denote the loss Hessian along the training trajectory. For each ZO optimizer, Section 4.2 provides explicit mean-square stability conditions under the linearized dynamics, together with computable lower and upper bounds on the corresponding stability threshold that depend only on the trace and top eigenvalue of the relevant curvature matrix. In large-scale neural network training, however, evaluating the *exact* mean-square stability condition is typically infeasible, since it requires the full spectrum of the Hessian (or the preconditioned Hessian for Adam-style methods). We therefore estimate only the trace and top eigenvalue during training, and use them to form tractable lower and upper bounds. Concretely, for the purpose of visualizing these dynamic conditions between $\boldsymbol{H}_t$ and the step size $\eta$, we present our results in the following format:

$$\text{(lower term)} \leqslant \text{(stability threshold)} \leqslant \text{(upper term)},$$

where the stability threshold depends on the step size $\eta$ and does not change over iterations and the upper and lower terms depend on the spectrum of $\boldsymbol{H}_t$ (e.g., Figure 2). We say the training operates near the mean-square EoS when the stability threshold remains within, or very close to, this interval for a sustained portion of training.

**ZO-GD.** From (2), we track mean-square stability via

$$\text{Tr}(\boldsymbol{H}_t) \leqslant \frac{2}{\eta} \leqslant \text{Tr}(\boldsymbol{H}_t) + 2\lambda_{\max}(\boldsymbol{H}_t). \quad (5)$$

**ZO-GDM.** From (3), we track mean-square stability via

$$\text{Tr}(\boldsymbol{H}_t) \leqslant \frac{2(1-\beta)}{\eta} \leqslant \text{Tr}(\boldsymbol{H}_t) + \frac{2\lambda_{\max}(\boldsymbol{H}_t)}{1+\beta}. \quad (6)$$

**ZO-Adam.** For ZO-Adam, the bounds depend on the preconditioner $\boldsymbol{P}_t$ at iteration $t$. From (4) in Theorem 3, we track mean-square stability via

$$\text{Tr}(\boldsymbol{P}_t^{-1}\boldsymbol{H}_t) \leqslant \frac{2}{\eta} \leqslant \text{Tr}(\boldsymbol{P}_t^{-1}\boldsymbol{H}_t) + \frac{2\lambda_{\max}(\boldsymbol{P}_t^{-1}\boldsymbol{H}_t)}{1+\beta_1}. \quad (7)$$

## 5.2. Experimental setup

We consider an image classification task on a subset of CIFAR-10 and train ZO methods using the squared loss. We evaluate three representative vision architectures: a CNN, a ResNet, and a Vision Transformer (ViT). Unless stated otherwise, we use full-batch training and a constant step size to match the linearized stability theory in Section 4.

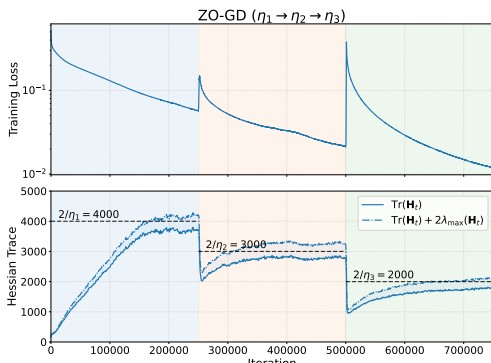

*Figure 3.* **Catapult dynamics in ZO-GD.** We train ZO-GD on CNN and increase the step size midway through training (from $\eta_1$ to $\eta_2$ and then to $\eta_3$). *Top:* the training loss exhibits a pronounced spike after each step size increase, consistent with catapult dynamics. *Bottom:* the Hessian trace $\text{Tr}(\boldsymbol{H}_t)$ drops sharply during the catapult phase and then rises again, re-equilibrating near the new stability threshold $2/\eta$.

**Optimizers and hyperparameters.** We study ZO-GD, ZO-GDM, and ZO-Adam, all using the standard two-point estimator with default smoothing parameter $\mu = 10^{-3}$. During training, we log curvature statistics every $1,000$ iterations by estimating the top Hessian eigenvalue using power iteration and the Hessian trace using Hutchinson's estimator. Additional experimental details are provided in Appendix C.

## 5.3. Main experiments

In Figure 2, we train ZO methods on the CNN and track the stability intervals in (5), (6), and (7). Across all three optimizers, we observe a consistent mean-square EoS pattern: after an initial phase of progressive sharpening, the tracked curvature terms adjust and stabilize so that the stability threshold remains within, or very close to, the corresponding intervals. As shown in Figure 6 and Figure 7, we observe the same behavior when training a ResNet and a ViT.

To test whether this behavior is specific to vision, we also train LSTM and Mamba sequence models on the synthetic sorting task described in Karpathy (2020), following the experimental setup used by Cohen et al. (2025, Appendix B.3). These experiments exhibit the same qualitative mean-square EoS behavior; full plots are provided in Appendix D.1.

Specifically, ($i$) for ZO-GD, the threshold $2/\eta$ remains close to the interval $[\text{Tr}(\boldsymbol{H}_t), \text{Tr}(\boldsymbol{H}_t) + 2\lambda_{\max}(\boldsymbol{H}_t)]$ across step sizes; ($ii$) for ZO-GDM, the threshold $2(1-\beta)/\eta$ remains close to $[\text{Tr}(\boldsymbol{H}_t), \text{Tr}(\boldsymbol{H}_t) + \frac{2}{1+\beta}\lambda_{\max}(\boldsymbol{H}_t)]$ across momentum values; and ($iii$) for ZO-Adam, the threshold $2/\eta$ remains close to the corresponding preconditioned interval in (7). In all cases, the *trace* of the (preconditioned) Hessian provides the dominant stability signal throughout training.

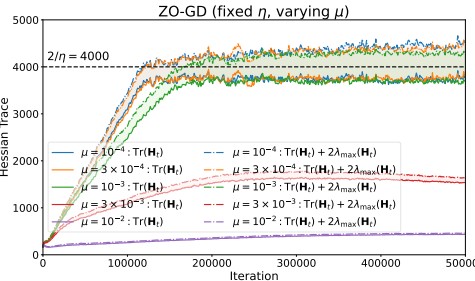

Figure 4. **Effect of the smoothing parameter $\mu$.** We train ZO-GD on a CNN with a fixed step size and vary the smoothing parameter $\mu$ in the two-point estimator. For moderate and small smoothing ($\mu \leqslant 10^{-3}$), ZO-GD operates at the mean-square EoS. For larger smoothing ($\mu \geqslant 3 \times 10^{-3}$), ZO-GD no longer reaches the EoS threshold and instead trains in a lower-curvature regime with a smaller Hessian trace.

### 5.4. Additional experiments

**Catapult dynamics.** In Figure 3, we train ZO-GD and increase the step size midway through training (from $\eta_1$ to $\eta_2$ and then to $\eta_3$). We observe a pronounced spike in the training loss after each increase, consistent with the *catapult* dynamics (Lewkowycz et al., 2020). Immediately after the step size increase, the new step size temporarily exceeds the mean-square stability critical step size at the current iterate, so the dynamics become locally unstable and the loss increases sharply. At the same time, the Hessian trace drops rapidly below the new threshold and then rises again, re-equilibrating near the new threshold.

**Effect of the smoothing parameter $\mu$.** In Figure 4, we train ZO-GD with fixed step size and vary the smoothing parameter $\mu$ in the two-point estimator. For moderate and small $\mu$, the tracked stability terms increase early in training and then stabilize near the threshold $2/\eta$, consistent with mean-square EoS behavior. For larger $\mu$, both $\mathrm{Tr}(\boldsymbol{H}_t)$ and $\mathrm{Tr}(\boldsymbol{H}_t) + 2\lambda_{\max}(\boldsymbol{H}_t)$ saturate at substantially smaller values and remain far below $2/\eta$, indicating that training does not approach the predicted mean-square stability boundary. Overall, mean-square EoS persists across a broad range of practically relevant smoothing levels, while overly large smoothing suppresses curvature growth. We attribute this phenomenon to implicit bias in Section 6.

**Beyond full-batch: mini-batch ZO-SGD.** Although our main results focus on full-batch ZO methods, we include a preliminary mini-batch experiment in Figure 5. Compared to full-batch ZO-GD, mini-batch ZO-SGD converges to significantly flatter regimes.

## 6. Discussion

In this section, we discuss implications of our mean-square stability theory and empirical mean-square EoS results, and

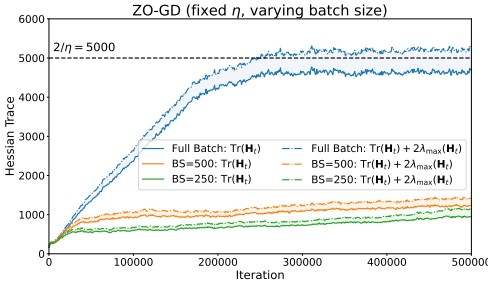

Figure 5. **Effect of batch size in mini-batch ZO-SGD.** We train mini-batch ZO-SGD on a CNN with a fixed step size and vary the batch size. Compared to full-batch ZO-GD, mini-batch ZO-SGD trains in a lower-curvature regime with a smaller Hessian trace.

highlight several directions for future work.

**Why mean-square stability is the relevant notion for ZO dynamics.** In ZO optimization, randomness persists even in full-batch training due to random perturbation directions. As a result, stability cannot be assessed solely through the mean trajectory; $\mathbb{E}[\boldsymbol{x}_t]$ may remain bounded even when fluctuations grow and dominate the behavior of the iterates. Mean-square stability captures this effect by directly controlling the second moment $\mathbb{E}\|\boldsymbol{x}_t - \boldsymbol{x}^\star\|^2$, while remaining analyzable under the linearized dynamics and yielding explicit step size conditions. Other notions of stability are also meaningful, such as stability of higher moments or tail-probability bounds. Nevertheless, our experiments suggest that the curvature quantities appearing in the mean-square stability conditions closely track the stability behavior observed during ZO neural network training.

**Curvature quantities that govern ZO stability.** For FO methods under the linearized dynamics, stability depends only on the largest eigenvalue of the (preconditioned) Hessian. In contrast, our results show that mean-square stability of ZO methods depends on the full Hessian spectrum. This dependence appears explicitly in the exact stability conditions, and it is reflected in the computable bounds through both trace and top-eigenvalue terms. A practically important regime is when $\mathrm{Tr}(\boldsymbol{H})$ dominates $\lambda_{\max}(\boldsymbol{H})$, in which case these bounds become tight and the trace term largely sets the stability scale. Empirically, this matches our empirical observations: the Hessian trace (or the preconditioned trace for ZO-Adam) closely tracks the relevant stability threshold throughout training, while $\lambda_{\max}(\boldsymbol{H}_t)$ can be less informative for ZO dynamics.

**Momentum reshapes ZO stability differently from FO stability.** Momentum provides a concrete example showing that ZO stability is not a direct analogue of FO stability. For GD with momentum (GDM), the linearized stability threshold increases with $\beta$, corresponding to stable training at sharper curvature levels, with $\lambda_{\max}(\boldsymbol{H}_t) \approx 2(1 + \beta)/\eta$ at the EoS. For ZO-GDM, our mean-square conditions im-

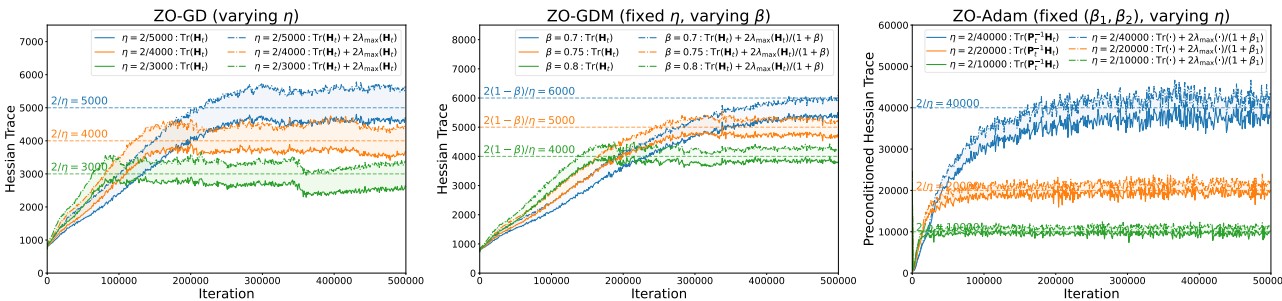

*Figure 6.* **Mean-square EoS for full-batch ZO methods on ResNet.** We train full-batch ZO-GD, ZO-GDM, and ZO-Adam on ResNet20 for CIFAR-10 and track the corresponding mean-square stability bounds and threshold in Section 5.1.

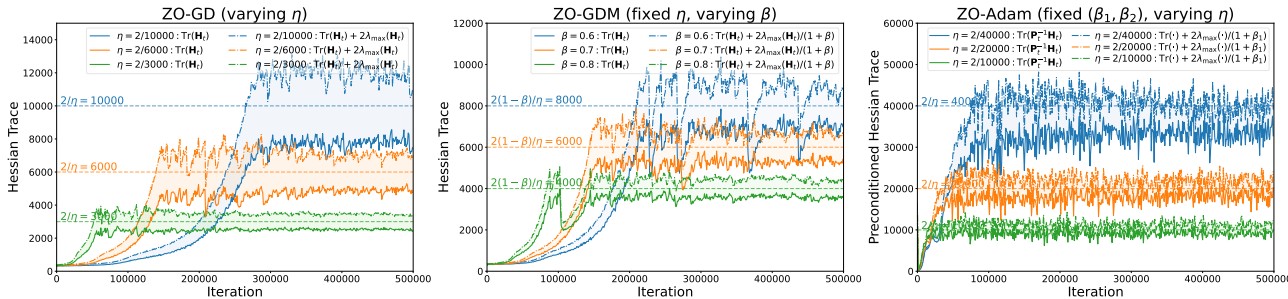

*Figure 7.* **Mean-square EoS for full-batch ZO methods on Vision Transformer.** We train full-batch ZO-GD, ZO-GDM, and ZO-Adam on a Vision Transformer for CIFAR-10 and track the corresponding mean-square stability bounds and threshold in Section 5.1.

ply the opposite dependence: increasing $\beta$ shrinks the stable regime and reduces the corresponding stability scale. Empirically, this is consistent with ZO-GDM operating in lower-curvature regimes as $\beta$ increases, with $\mathrm{Tr}(\boldsymbol{H}_t) \approx 2(1-\beta)/\eta$ at the mean-square EoS.

Intuitively, in FO-GDM, momentum damps deterministic oscillations along sharp directions, which enlarges the stable step-size region. In ZO-GDM, by contrast, momentum accumulates not only the gradient signal but also the random-direction estimator noise. Since ZO stability is governed by second moments, this extra accumulated noise makes the dynamics less stable and causes increasing $\beta$ to shrink the stable region.

A similar contrast appears for adaptive methods. For (frozen) Adam, increasing $\beta_1$ increases the stability threshold, with $\lambda_{\max}(\boldsymbol{P}_t^{-1}\boldsymbol{H}_t) \approx 2(1+\beta_1)/((1-\beta_1)\eta)$. For ZO-Adam, by comparison, our experiments suggest that $\mathrm{Tr}(\boldsymbol{P}_t^{-1}\boldsymbol{H}_t) \approx 2/\eta$ at the mean-square EoS, which is independent of $\beta_1$ (see Figure 10 and Appendix D.2). Understanding how these effects translate into practical benefits (or tradeoffs) of momentum in ZO training is an interesting open question.

**Effect of the smoothing parameter $\mu$ and trace-related implicit bias.** The two-point estimator introduces smoothing, controlled by $\mu$, that changes both the bias and the noise structure of the ZO update. Zhang et al. (2025a) connect

such ZO optimization to an implicit preference for small-trace regions, formalized as approximately minimizing

$$f_\mu(\boldsymbol{x}) \; := \; f(\boldsymbol{x}) + \frac{\mu^2}{2}\,\mathrm{Tr}\big(\nabla^2 f(\boldsymbol{x})\big),$$

up to higher-order terms. Under this perspective, $\mu$ directly modulates a curvature-dependent bias in the effective objective, and large $\mu$ can prevent the dynamics from approaching the mean-square stability threshold predicted by the unsmoothed linearized model. A systematic theory that jointly captures $(i)$ the mean-square stability constraint and $(ii)$ the effect of smoothing bias on the effective landscape is an interesting direction for future work.

**Beyond full-batch: mini-batch ZO methods.** Our analysis focuses on full-batch ZO dynamics, where the only randomness comes from the estimator directions. In practical settings, mini-batching introduces an additional noise source through stochastic sampling of data points. A complete stability theory for mini-batch ZO training would need to incorporate both estimator noise and sampling noise, and quantify how these two sources interact in the second-moment recursion. Recent work gives sharp mean-square stability thresholds for mini-batch SGD (Mulayoff & Michaeli, 2024). Deriving analogous results for mini-batch ZO methods would clarify whether mean-square EoS persists under data subsampling, and which curvature quantities control stability in that regime.

# 7. Conclusion

We developed a mean-square linear stability theory for zeroth-order (ZO) optimization methods based on the standard two-point estimator, including ZO-GD, ZO-GDM, and Adam-style preconditioned variants. We derived exact step size characterizations for mean-square stability via linearization, showing that ZO stability depends on the full spectrum of the (preconditioned) Hessian and admits computable bounds in terms of the trace and top eigenvalue.

Guided by these results, we empirically studied full-batch ZO training on standard neural network architectures (CNN, ResNet, and ViT) and found consistent evidence that ZO methods operate at the mean-square edge of stability: the curvature quantities governing the theoretical stability threshold adapt during training and stabilize near the predicted boundary. Across methods, the stability behavior is driven primarily by trace-based curvature terms, providing a concrete mechanism through which large step sizes implicitly regularize ZO training dynamics.

Our results position mean-square stability as a principled framework for analyzing and predicting ZO optimization behavior in deep learning. A natural next step is to use this mean-square EoS perspective to extend the central-flow framework of Cohen et al. (2025) to zeroth-order methods, and to empirically verify whether the resulting flow description captures ZO training dynamics across architectures and tasks. It is also important to extend the theory to mini-batch ZO methods. Another important direction is to understand how stability constraints interact with optimization efficiency and generalization in practice.

## Acknowledgements

MS thanks Alex Damian and Jeremy Cohen for insightful discussion. LZ gratefully acknowledges funding by the Max Planck ETH Center for Learning Systems. BL and NH are supported by Swiss National Science Foundation (SNSF) Starting Grant. MM thanks the German Research Foundation for the support. SO acknowledges funding by NSF grants 2112471, 2229876, and 2505865.

## Impact Statement

This paper presents work whose goal is to advance the field of machine learning. There are many potential societal consequences of our work, none which we feel must be specifically highlighted here.

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

# A. Proofs for the mean-square stability analysis of zeroth-order methods

In this section, we provide the proofs of the mean-square linear stability results in Section 4.2. Our analysis treats zeroth-order gradient descent with momentum (ZO-GDM) as the canonical zeroth-order method. The remaining algorithms studied in the paper are handled as special cases or extensions: ZO-GD corresponds to the specialization $\beta = 0$, while Frozen ZO-Adam is analyzed by modifying the covariance recursion under an additional commutativity assumption.

The core of the analysis proceeds in two steps. First, we derive an explicit linear recursion for the second-moment quantities of the ZO-GDM iterates under the linearized dynamics (Appendix A.1). This recursion induces a linear operator acting on a product space of covariance matrices. Second, we characterize mean-square linear stability by analyzing the spectral radius of this operator (Appendix A.2). A key technical ingredient is that the operator preserves a natural cone and includes a rank-one global coupling term, which allows its spectral radius to be characterized via the Krein–Rutman Theorem.

We first derive the second-moment recursion for ZO-GDM and express it in operator form. We then develop the spectral analysis of the resulting covariance operator and use it to prove Theorem 2. Theorem 1 follows immediately as the special case $\beta = 0$, and Theorem 3 is proved by adapting the analogous argument to the frozen preconditioned setting.

The proof of Theorem 2 is provided in Appendix A.3, and the proof of Theorem 3 is provided in Appendix A.4.

## A.1. Second-moment recursion and covariance operator for ZO-GDM

In this subsection, we derive the linear recursion governing the second-moment dynamics of ZO-GDM under the linearized dynamics. Following Definition 1, we consider applying ZO-GDM to the quadratic objective

$$f_{\text{quad}}(\boldsymbol{x}) = \tfrac{1}{2}\boldsymbol{x}^\top \boldsymbol{H}\boldsymbol{x},$$

where $\boldsymbol{H} \geq 0$ and $\boldsymbol{H} \neq \boldsymbol{0}$.

Recall that the ZO-GDM updates are given by

$$\boldsymbol{m}_{t+1} = \beta\boldsymbol{m}_t + \widehat{\nabla} f_{\text{quad}}(\boldsymbol{x}_t),$$
$$\boldsymbol{x}_{t+1} = \boldsymbol{x}_t - \eta\boldsymbol{m}_{t+1},$$

where $\beta \in [0, 1)$ with initialization $\boldsymbol{m}_0 = \boldsymbol{0}$, and $\widehat{\nabla} f_{\text{quad}}(\boldsymbol{x}_t)$ denotes the standard two-point estimator

$$\widehat{\nabla} f_{\text{quad}}(\boldsymbol{x}_t) = \frac{f_{\text{quad}}(\boldsymbol{x}_t + \mu\boldsymbol{u}_t) - f_{\text{quad}}(\boldsymbol{x}_t - \mu\boldsymbol{u}_t)}{2\mu}\,\boldsymbol{u}_t, \qquad \boldsymbol{u}_t \sim \mathcal{N}(0, I_d).$$

Under the quadratic model, this estimator admits the explicit form

$$\widehat{\nabla} f_{\text{quad}}(\boldsymbol{x}_t) = (\boldsymbol{u}_t\boldsymbol{u}_t^\top)\boldsymbol{H}\boldsymbol{x}_t,$$

and is unbiased, i.e., $\mathbb{E}[\widehat{\nabla} f_{\text{quad}}(\boldsymbol{x}_t)] = \boldsymbol{H}\boldsymbol{x}_t$.

Let $\boldsymbol{H} = \boldsymbol{U}\boldsymbol{\Lambda}\boldsymbol{U}^\top$ with $\boldsymbol{\Lambda} = \text{diag}(\lambda_1, \ldots, \lambda_d)$ and orthogonal $\boldsymbol{U}$. Defining rotated variables $\bar{\boldsymbol{x}}_t = \boldsymbol{U}^\top\boldsymbol{x}_t$, $\bar{\boldsymbol{m}}_t = \boldsymbol{U}^\top\boldsymbol{m}_t$, and $\bar{\boldsymbol{u}}_t = \boldsymbol{U}^\top\boldsymbol{u}_t$, rotational invariance implies $\bar{\boldsymbol{u}}_t \sim \mathcal{N}(\boldsymbol{0}, \boldsymbol{I}_d)$ i.i.d., and the dynamics in the rotated coordinates take the same form with $\boldsymbol{H}$ replaced by $\boldsymbol{\Lambda}$. Hence, without loss of generality, we assume $\boldsymbol{H} = \boldsymbol{\Lambda}$ in what follows.

We now state a lemma that reduces the ZO-GDM second-moment dynamics to a linear covariance operator, which will be the basis for the spectral analysis in subsequent subsections.

**Lemma 1** (Second-moment recursion and covariance operator for ZO-GDM). *Consider ZO-GDM applied to the quadratic model $f_{\text{quad}}(\boldsymbol{x}) = \tfrac{1}{2}\boldsymbol{x}^\top\boldsymbol{\Lambda}\boldsymbol{x}$ with $\boldsymbol{\Lambda} = \text{diag}(\lambda_1, \ldots, \lambda_d) \neq \boldsymbol{0}$, step size $\eta > 0$, and momentum $\beta \in [0, 1)$. For each iteration $t$ and coordinate $i = 1, \ldots, d$, define the $2 \times 2$ covariance block*

$$\boldsymbol{W}_{i,t} := \begin{bmatrix} \mathbb{E}[x_{i,t}^2] & \mathbb{E}[\eta x_{i,t}m_{i,t}] \\ \mathbb{E}[\eta x_{i,t}m_{i,t}] & \mathbb{E}[\eta^2 m_{i,t}^2] \end{bmatrix},$$

*where $\boldsymbol{x}_t = (x_{1,t}, \ldots, x_{d,t}) \in \mathbb{R}^d$ and $\boldsymbol{m}_t = (m_{1,t}, \ldots, m_{d,t}) \in \mathbb{R}^d$. Then the covariance blocks satisfy the recursion*

$$\boldsymbol{W}_{i,t+1} = \boldsymbol{A}_i\boldsymbol{W}_{i,t}\boldsymbol{A}_i^\top + \eta^2\left(\lambda_i^2(\boldsymbol{W}_{i,t})_{11} + \sum_{j=1}^{d}\lambda_j^2(\boldsymbol{W}_{j,t})_{11}\right)\boldsymbol{Q},$$

*where*

$$A_i := \begin{bmatrix} 1 - \eta\lambda_i & -\beta \\ \eta\lambda_i & \beta \end{bmatrix}, \qquad Q := \begin{bmatrix} 1 & -1 \\ -1 & 1 \end{bmatrix}.$$

*Proof of Lemma 1.* We consider with the augmented state $(x_t, \eta m_t) \in \mathbb{R}^{2d}$, for which the ZO-GDM update can be written as

$$\begin{bmatrix} x_{t+1} \\ \eta m_{t+1} \end{bmatrix} = \begin{bmatrix} I - \eta u_t u_t^\top \Lambda & -\beta I \\ \eta u_t u_t^\top \Lambda & \beta I \end{bmatrix} \begin{bmatrix} x_t \\ \eta m_t \end{bmatrix}.$$

To track second moments, define

$$X_t := \mathbb{E}[x_t x_t^\top], \qquad Y_t := \mathbb{E}[\eta^2 m_t m_t^\top], \qquad C_t := \mathbb{E}[\eta x_t m_t^\top].$$

Using independence of $u_t$ from $(x_t, m_t)$ and applying Lemma 4 (Isserlis's theorem) to evaluate the Gaussian fourth moments, we obtain the following closed recursion:

$$X_{t+1} = X_t - \eta(\Lambda X_t + X_t \Lambda) + \eta^2 \big(2\Lambda X_t \Lambda + \mathrm{Tr}(\Lambda X_t \Lambda)I\big) - \beta(C_t + C_t^\top) + \beta\eta(\Lambda C_t + C_t^\top \Lambda) + \beta^2 Y_t,$$
$$Y_{t+1} = \eta^2 \big(2\Lambda X_t \Lambda + \mathrm{Tr}(\Lambda X_t \Lambda)I\big) + \eta\beta(\Lambda C_t + C_t^\top \Lambda) + \beta^2 Y_t,$$
$$C_{t+1} = \eta X_t \Lambda - \eta^2 \big(2\Lambda X_t \Lambda + \mathrm{Tr}(\Lambda X_t \Lambda)I\big) + \beta C_t - \beta\eta(\Lambda C_t + C_t^\top \Lambda) - \beta^2 Y_t.$$

Since $\Lambda$ is diagonal, the coordinates decouple except through the scalar coupling term

$$\mathrm{Tr}(\Lambda X_t \Lambda) = \sum_{j=1}^d \lambda_j^2 (X_t)_{jj}.$$

Taking diagonal entries, for each coordinate $i = 1, \ldots, d$, we obtain

$$(X_{t+1})_{ii} = (1 - 2\eta\lambda_i + 2\eta^2\lambda_i^2)(X_t)_{ii} + \beta^2 (Y_t)_{ii} + 2\beta(-1 + \eta\lambda_i)(C_t)_{ii} + \eta^2 \sum_{j=1}^d \lambda_j^2 (X_t)_{jj},$$

$$(Y_{t+1})_{ii} = 2\eta^2\lambda_i^2(X_t)_{ii} + \beta^2 (Y_t)_{ii} + 2\beta\eta\lambda_i(C_t)_{ii} + \eta^2 \sum_{j=1}^d \lambda_j^2 (X_t)_{jj},$$

$$(C_{t+1})_{ii} = (\eta\lambda_i - 2\eta^2\lambda_i^2)(X_t)_{ii} - \beta^2 (Y_t)_{ii} + \beta(1 - 2\eta\lambda_i)(C_t)_{ii} - \eta^2 \sum_{j=1}^d \lambda_j^2 (X_t)_{jj}.$$

Recalling that

$$(X_t)_{ii} = \mathbb{E}[x_{i,t}^2], \qquad (Y_t)_{ii} = \mathbb{E}[\eta^2 m_{i,t}^2], \qquad (C_t)_{ii} = \mathbb{E}[\eta x_{i,t} m_{i,t}],$$

the above relations can be grouped into the $2 \times 2$ covariance block $W_{i,t}$ defined in the statement of the lemma. A direct calculation then gives

$$W_{i,t+1} = A_i W_{i,t} A_i^\top + \eta^2 \left( \lambda_i^2 (W_{i,t})_{11} + \sum_{j=1}^d \lambda_j^2 (W_{j,t})_{11} \right) Q,$$

with

$$A_i := \begin{bmatrix} 1 - \eta\lambda_i & -\beta \\ \eta\lambda_i & \beta \end{bmatrix}, \qquad Q := \begin{bmatrix} 1 & -1 \\ -1 & 1 \end{bmatrix}.$$

This completes the proof. □

Lemma 1 shows that the ZO-GDM second-moment dynamics are governed by a closed linear recursion over the covariance blocks $\{W_{i,t}\}_{i=1}^d$. In particular,

$$\mathbb{E}\|x_t\|^2 = \sum_{i=1}^d (W_{i,t})_{11},$$

so mean-square linear stability is equivalent to uniform boundedness of the quantity $\sum_{i=1}^{d}(\boldsymbol{W}_{i,t})_{11}$ over iterations.

Collecting the blocks into the product space of $2 \times 2$ symmetric matrices $(\mathbb{S}^2)^d$, the mapping $\{\boldsymbol{W}_{i,t}\}_{i=1}^{d} \mapsto \{\boldsymbol{W}_{i,t+1}\}_{i=1}^{d}$ can be viewed as the action of a linear operator $\mathcal{T} : (\mathbb{S}^2)^d \to (\mathbb{S}^2)^d$, parameterized by the step size $\eta$, momentum $\beta$, and the eigenvalues $\lambda_1, \ldots, \lambda_d$. In the next subsection, we analyze the spectral properties of this covariance operator and derive explicit conditions under which its spectral radius is smaller than one.

## A.2. Spectral analysis of the covariance operator

In this subsection, we analyze the spectral properties of the covariance operator induced by the ZO-GDM second-moment recursion derived in Lemma 1 (Appendix A.1). That recursion is governed by a linear operator $\mathcal{T} : (\mathbb{S}^2)^d \to (\mathbb{S}^2)^d$ acting on the covariance blocks $\{\boldsymbol{W}_{i,t}\}_{i=1}^{d}$, and in particular controls $\mathbb{E}\|\boldsymbol{x}_t\|^2 = \sum_{i=1}^{d}(\boldsymbol{W}_{i,t})_{11}$. We therefore analyze the spectral radius of $\mathcal{T}$.

The operator $\mathcal{T}$ preserves the cone $(\mathbb{S}^2_+)^d$ and contains a rank-one coupling term across coordinates, allowing us to invoke the Krein–Rutman theorem. The following theorem gives an exact characterization of the spectral radius of $\mathcal{T}$, which forms the technical core of the mean-square stability analysis for ZO-GDM.

**Theorem 4** (Spectral characterization of ZO-GDM covariance operator). *Let $\mathcal{X} := (\mathbb{S}^2)^d$ and $\mathcal{K} := (\mathbb{S}^2_+)^d$. Fix $\eta > 0$, $\beta \in [0, 1)$, and $\lambda_{\max} = \lambda_1 \geqslant \lambda_2 \geqslant \cdots \geqslant \lambda_d > 0$. Define the linear operator $\mathcal{T} : \mathcal{X} \to \mathcal{X}$ blockwise by*

$$(\mathcal{T}(\boldsymbol{W}))_i = \boldsymbol{A}_i \boldsymbol{W}_i \boldsymbol{A}_i^\top + \eta^2 \left( \lambda_i^2 (\boldsymbol{W}_i)_{11} + \sum_{j=1}^{d} \lambda_j^2 (\boldsymbol{W}_j)_{11} \right) \boldsymbol{Q}, \qquad i = 1, \ldots, d,$$

*where*

$$\boldsymbol{A}_i := \begin{bmatrix} 1 - \eta\lambda_i & -\beta \\ \eta\lambda_i & \beta \end{bmatrix}, \qquad \boldsymbol{Q} := \begin{bmatrix} 1 & -1 \\ -1 & 1 \end{bmatrix}.$$

*Define the scalar function*

$$S(\eta, \beta) := \sum_{i=1}^{d} \frac{\eta\lambda_i}{2(1 - \beta)\left(1 - \frac{\eta\lambda_i}{1-\beta^2}\right)}.$$

*Then the operator $\mathcal{T}$ satisfies the following properties.*

(a) *(**Leading eigenvalue in the cone**) The operator $\mathcal{T}$ preserves the cone $\mathcal{K}$, i.e., $\mathcal{T}(\mathcal{K}) \subseteq \mathcal{K}$. Moreover, $\mathcal{T}$ has an eigenvalue equal to its spectral radius $\rho(\mathcal{T})$, with an associated eigenvector $\boldsymbol{W}^\star = (\boldsymbol{W}_1^\star, \ldots, \boldsymbol{W}_d^\star) \in \mathcal{K}\backslash\{\boldsymbol{0}\}$ satisfying*

$$\rho(\mathcal{T}) \geqslant \eta^2 \sum_{i=1}^{d} \lambda_i^2 > 0 \qquad \text{and} \qquad \sum_{i=1}^{d}(\boldsymbol{W}_i^\star)_{11} > 0.$$

(b) *(**Critical case**)*
$$\rho(\mathcal{T}) = 1 \quad \Longleftrightarrow \quad \eta\lambda_{\max} < 1 - \beta^2 \ \text{ and } \ S(\eta, \beta) = 1.$$

(c) *(**Subcritical case**)*
$$\rho(\mathcal{T}) < 1 \quad \Longleftrightarrow \quad \eta\lambda_{\max} < 1 - \beta^2 \ \text{ and } \ S(\eta, \beta) < 1.$$

*Proof of Theorem 4.* We work on the product space $\mathcal{X} := (\mathbb{S}^2)^d$ equipped with the product cone $\mathcal{K} := (\mathbb{S}^2_+)^d$ and the induced partial order $\preceq$.

**Proof of (a).** Let $\boldsymbol{W} = (\boldsymbol{W}_1, \ldots, \boldsymbol{W}_d) \in \mathcal{K}$, i.e., $\boldsymbol{W}_i \succeq 0$ for all $i = 1, \ldots, d$. Then $\boldsymbol{A}_i \boldsymbol{W}_i \boldsymbol{A}_i^\top \succeq 0$ and $(\boldsymbol{W}_i)_{11} \geqslant 0$, so $\sum_{j=1}^{d} \lambda_j^2 (\boldsymbol{W}_j)_{11} \geqslant 0$. Hence $(\mathcal{T}(\boldsymbol{W}))_i \succeq 0$ for all $i$, and therefore $\mathcal{T}(\mathcal{K}) \subseteq \mathcal{K}$.

Since $\mathcal{X}$ is finite dimensional and $\mathcal{K}$ is a closed, convex, pointed cone with nonempty interior, the Krein–Rutman theorem (Lemma 6) implies that $\mathcal{T}$ has an eigenvalue equal to its spectral radius $\rho(\mathcal{T})$ with a corresponding eigenvector $\boldsymbol{W}^\star \in \mathcal{K}\backslash\{\boldsymbol{0}\}$.

To lower bound $\rho(\mathcal{T})$, define $\bar{\boldsymbol{Q}} := (\boldsymbol{Q}, \ldots, \boldsymbol{Q}) \in \mathcal{K} \backslash \{\boldsymbol{0}\}$. For each $i$,

$$(\mathcal{T}(\bar{\boldsymbol{Q}}))_i = \boldsymbol{A}_i \boldsymbol{Q} \boldsymbol{A}_i^\top + \eta^2 \left( \lambda_i^2 + \sum_{j=1}^d \lambda_j^2 \right) \boldsymbol{Q} \succeq \eta^2 \left( \sum_{j=1}^d \lambda_j^2 \right) \boldsymbol{Q}.$$

Thus $\mathcal{T}(\bar{\boldsymbol{Q}}) \succeq c\,\bar{\boldsymbol{Q}}$ with $c := \eta^2 \sum_{j=1}^d \lambda_j^2 > 0$. Iterating gives $\mathcal{T}^t(\bar{\boldsymbol{Q}}) \succeq c^t \bar{\boldsymbol{Q}}$ for all $t \geqslant 1$. Fixing any norm on $\mathcal{X}$ and applying Gelfand's formula, we obtain

$$\rho(\mathcal{T}) \geqslant c = \eta^2 \sum_{j=1}^d \lambda_j^2.$$

Finally, we show $\sum_i (\boldsymbol{W}_i^\star)_{11} > 0$. If $(\boldsymbol{W}_i^\star)_{11} = 0$ for all $i$, then $\boldsymbol{W}_i^\star = \left[ \begin{smallmatrix} 0 & 0 \\ 0 & y_i \end{smallmatrix} \right]$ with $y_i \geqslant 0$. But then

$$\rho(\mathcal{T}) \boldsymbol{W}_i^\star = (\mathcal{T}(\boldsymbol{W}^\star))_i = \boldsymbol{A}_i \boldsymbol{W}_i^\star \boldsymbol{A}_i^\top = \beta^2 y_i \, \boldsymbol{Q} \ .$$

Note that $\beta^2 y_i \, \boldsymbol{Q}$ has the same value at the $(1,1)$-entry and $(2,2)$-entry. Consequently, $\rho(\mathcal{T})\boldsymbol{W}_i^\star$ should have the same value at the $(1,1)$-entry and $(2,2)$-entry. Thus $\rho(\mathcal{T})y_i = 0$, forcing $y_i = 0$ since $\rho(\mathcal{T}) > 0$. Hence $\boldsymbol{W}^\star = 0$, a contradiction. This concludes the proof of (a).

**Local-global decomposition.**    For each $i = 1, \ldots, d$, we define the local linear map $\mathcal{M}_i : \mathbb{S}^2 \to \mathbb{S}^2$ by

$$\mathcal{M}_i(\boldsymbol{X}) := \boldsymbol{A}_i \boldsymbol{X} \boldsymbol{A}_i^\top + \eta^2 \lambda_i^2 (\boldsymbol{X})_{11} \boldsymbol{Q}.$$

Define the global coupling scalar function $s : \mathcal{X} \to \mathbb{R}$ by

$$s(\boldsymbol{W}) := \sum_{j=1}^d \eta^2 \lambda_j^2 (\boldsymbol{W}_j)_{11}.$$

Then, the linear operator $\mathcal{T} : \mathcal{X} \to \mathcal{X}$ can be decomposed as below:

$$(\mathcal{T}(\boldsymbol{W}))_i = \mathcal{M}_i(\boldsymbol{W}_i) + s(\boldsymbol{W})\boldsymbol{Q}, \qquad i = 1, \ldots, d. \tag{8}$$

Note that each linear map $\mathcal{M}_i$ is $\mathbb{S}_+^2$-preserving, i.e., $\mathcal{M}_i(\mathbb{S}_+^2) \subseteq \mathbb{S}_+^2$. Define the block-diagonal operator $\mathcal{M} : \mathcal{X} \to \mathcal{X}$ and $\bar{\boldsymbol{Q}} \in \mathcal{K}$ by

$$(\mathcal{M}(\boldsymbol{W}))_i := \mathcal{M}_i(\boldsymbol{W}_i) \ , \quad \text{and} \quad \bar{\boldsymbol{Q}} := (\boldsymbol{Q}, \ldots, \boldsymbol{Q}) \in \mathcal{K} \ . \tag{9}$$

Then (8) can be written as

$$\mathcal{T}(\boldsymbol{W}) = \mathcal{M}(\boldsymbol{W}) + s(\boldsymbol{W})\, \bar{\boldsymbol{Q}} \ .$$

**Positivity of $s(\boldsymbol{W}^\star)$.**    Let $\boldsymbol{W}^\star \in \mathcal{K} \backslash \{\boldsymbol{0}\}$ denote the leading eigenvector of $\mathcal{T}$ satisfying $\mathcal{T}(\boldsymbol{W}^\star) = \rho(\mathcal{T})\boldsymbol{W}^\star$ and $\sum_{i=1}^d (\boldsymbol{W}_i^\star)_{11} > 0$, which exists by Theorem 4(a). Then, $s(\boldsymbol{W}^\star) = \sum_{j=1}^d \eta^2 \lambda_j^2 (\boldsymbol{W}_j^\star)_{11} > 0$.

**Key lemmas.**    We use the following key lemmas to prove (b) and (c): Lemma 2 and Lemma 3.

Lemma 2 shows that $\rho(\mathcal{M}_i) < 1$ if and only if $\eta \lambda_i < 1 - \beta^2$. Moreover, if $\eta \lambda_i < 1 - \beta^2$, then $\text{Id} - \mathcal{M}_i$ is invertible, $(\text{Id} - \mathcal{M}_i)^{-1}(\mathbb{S}_+^2) \subseteq \mathbb{S}_+^2$, and $\boldsymbol{Y}_i := (\text{Id} - \mathcal{M}_i)^{-1}(\boldsymbol{Q}) \in \mathbb{S}_+^2$ satisfies

$$\gamma_i := (\boldsymbol{Y}_i)_{11} = \frac{1 + \beta}{2\eta \lambda_i (1 - \beta^2 - \eta \lambda_i)}.$$

Lemma 3 gives a scaled infeasibility statement: for any $c > 0$, if $\rho(c\mathcal{M}_i) \geqslant 1$, then for every $\alpha > 0$, there does not exist $\boldsymbol{W} \succeq 0$ such that

$$(\text{Id} - c\mathcal{M}_i)(\boldsymbol{W}) \succeq \alpha \boldsymbol{Q}.$$

**Proof of (b).** We prove
$$\rho(\mathcal{T}) = 1 \quad \Longleftrightarrow \quad \eta\lambda_{\max} < 1 - \beta^2 \ \text{ and } \ S(\eta, \beta) = 1 .$$

( $\Longrightarrow$ ) Assume $\rho(\mathcal{T}) = 1$. By Theorem 4(a), there exists $\boldsymbol{W}^\star \in \mathcal{K}\backslash\{\boldsymbol{0}\}$ with $\mathcal{T}(\boldsymbol{W}^\star) = \boldsymbol{W}^\star$. Let $s^\star := s(\boldsymbol{W}^\star)$. Then, $s^\star > 0$ by the positivity of $s(\boldsymbol{W}^\star)$. From (8), for each $i$,
$$\boldsymbol{W}_i^\star = (\mathcal{T}(\boldsymbol{W}^\star))_i = \mathcal{M}_i(\boldsymbol{W}_i^\star) + s^\star \boldsymbol{Q},$$

or equivalently,
$$(\mathrm{Id} - \mathcal{M}_i)(\boldsymbol{W}_i^\star) = s^\star \boldsymbol{Q}.$$

Since $s^\star > 0$, Lemma 3 implies $\rho(\mathcal{M}_i) < 1$ for every $i$, and by Lemma 2(a),
$$\eta\lambda_i < 1 - \beta^2 \quad \text{for all } i, \qquad \text{hence} \qquad \eta\lambda_{\max} < 1 - \beta^2.$$

By Lemma 2(b), $\mathrm{Id} - \mathcal{M}_i$ is invertible and $(\mathrm{Id} - \mathcal{M}_i)^{-1}$ is $\mathbb{S}_+^2$-preserving, so
$$\boldsymbol{W}_i^\star = s^\star(\mathrm{Id} - \mathcal{M}_i)^{-1}\boldsymbol{Q} = s^\star \boldsymbol{Y}_i,$$

where $\boldsymbol{Y}_i \geq 0$ and $\gamma_i := (\boldsymbol{Y}_i)_{11}$ is given by Lemma 2(c). Taking $(1,1)$-entries and plugging into the definition of $s^\star$, we get
$$s^\star = \sum_{i=1}^d \eta^2\lambda_i^2(\boldsymbol{W}_i^\star)_{11} = \sum_{i=1}^d \eta^2\lambda_i^2\gamma_i s^\star.$$

Cancelling $s^\star > 0$ gives
$$1 = \sum_{i=1}^d \eta^2\lambda_i^2\gamma_i.$$

Using Lemma 2(c),
$$\eta^2\lambda_i^2\gamma_i = \eta^2\lambda_i^2 \cdot \frac{1 + \beta}{2\eta\lambda_i(1 - \beta^2 - \eta\lambda_i)} = \frac{\eta\lambda_i}{2(1 - \beta)\left(1 - \frac{\eta\lambda_i}{1 - \beta^2}\right)}.$$

Therefore $\sum_{i=1}^d \eta^2\lambda_i^2\gamma_i = S(\eta, \beta)$, and thus we conclude that $S(\eta, \beta) = 1$.

( $\Longleftarrow$ ) Assume $\eta\lambda_{\max} < 1 - \beta^2$ and $S(\eta, \beta) = 1$. Then, $\eta\lambda_i < 1 - \beta^2$ for all $i$.

By Lemma 2(b), the matrices $\boldsymbol{Y}_i := (\mathrm{Id} - \mathcal{M}_i)^{-1}\boldsymbol{Q}$ are well-defined and satisfy $\boldsymbol{Y}_i \geq 0$. Set $\boldsymbol{Y} := (\boldsymbol{Y}_1, \ldots, \boldsymbol{Y}_d) \in \mathcal{K}\backslash\{\boldsymbol{0}\}$. For each $i$, $(\mathrm{Id} - \mathcal{M}_i)(\boldsymbol{Y}_i) = \boldsymbol{Q}$, i.e. $\mathcal{M}_i(\boldsymbol{Y}_i) = \boldsymbol{Y}_i - \boldsymbol{Q}$. Moreover,
$$s(\boldsymbol{Y}) = \sum_{i=1}^d \eta^2\lambda_i^2(\boldsymbol{Y}_i)_{11} = \sum_{i=1}^d \eta^2\lambda_i^2\gamma_i = S(\eta, \beta) = 1.$$

Thus (8) gives, for each $i$,
$$(\mathcal{T}(\boldsymbol{Y}))_i = \mathcal{M}_i(\boldsymbol{Y}_i) + s(\boldsymbol{Y})\boldsymbol{Q} = (\boldsymbol{Y}_i - \boldsymbol{Q}) + \boldsymbol{Q} = \boldsymbol{Y}_i,$$

so $\mathcal{T}(\boldsymbol{Y}) = \boldsymbol{Y}$. Hence, 1 is an eigenvalue of $\mathcal{T}$ with eigenvector $\boldsymbol{Y}$, and therefore $\rho(\mathcal{T}) \geqslant 1$. Set $r := \rho(\mathcal{T}) \geqslant 1$.

By Theorem 4(a), there exists $\boldsymbol{W}^\star \in \mathcal{K}\backslash\{\boldsymbol{0}\}$ with $\mathcal{T}(\boldsymbol{W}^\star) = \rho(\mathcal{T})\boldsymbol{W}^\star$. Set $s^\star := s(\boldsymbol{W}^\star)$. By the positivity of $s(\boldsymbol{W}^\star)$, we have $s^\star > 0$. From $\mathcal{T}(\boldsymbol{W}^\star) = r\boldsymbol{W}^\star$ and the decomposition (9), we obtain
$$r\boldsymbol{W}^\star = \mathcal{M}(\boldsymbol{W}^\star) + s^\star\bar{\boldsymbol{Q}}, \qquad \text{so that} \qquad \left(\mathrm{Id} - \tfrac{1}{r}\mathcal{M}\right)\boldsymbol{W}^\star = \tfrac{s^\star}{r}\bar{\boldsymbol{Q}}.$$

Since $\eta\lambda_{\max} < 1 - \beta^2$, Lemma 2(a) implies that $\rho(\mathcal{M}_i) < 1$ for all $i$. Consequently, $\rho(\mathcal{M}) = \max_i \rho(\mathcal{M}_i) < 1$. Since $r \geqslant 1$, we have $\rho(\frac{1}{r}\mathcal{M}) \leqslant \rho(\mathcal{M}) < 1$ and hence $\mathrm{Id} - \frac{1}{r}\mathcal{M}$ is invertible with
$$\left(\mathrm{Id} - \tfrac{1}{r}\mathcal{M}\right)^{-1} = \sum_{k=0}^\infty \left(\tfrac{1}{r}\mathcal{M}\right)^k,$$

where the series converges in operator norm. Applying this inverse gives

$$\boldsymbol{W}^\star = \tfrac{s^\star}{r}\left(\mathrm{Id} - \tfrac{1}{r}\mathcal{M}\right)^{-1}\bar{\boldsymbol{Q}}.$$

Applying the nonnegative functional $s(\cdot)$ to both sides and cancelling $s^\star > 0$, we obtain

$$r = s\left(\left(\mathrm{Id} - \tfrac{1}{r}\mathcal{M}\right)^{-1}\bar{\boldsymbol{Q}}\right). \tag{10}$$

Using the Neumann series, we have

$$s\left(\left(\mathrm{Id} - \tfrac{1}{r}\mathcal{M}\right)^{-1}\bar{\boldsymbol{Q}}\right) = \sum_{k=0}^{\infty} r^{-k}\, s(\mathcal{M}^k(\bar{\boldsymbol{Q}})) \leqslant \sum_{k=0}^{\infty} s(\mathcal{M}^k(\bar{\boldsymbol{Q}})) = s\left((\mathrm{Id} - \mathcal{M})^{-1}\bar{\boldsymbol{Q}}\right).$$

Because $\mathcal{M}$ is block-diagonal, $(\mathrm{Id} - \mathcal{M})^{-1}$ is block-diagonal with blocks $(\mathrm{Id} - \mathcal{M}_i)^{-1}$, and thus

$$(\mathrm{Id} - \mathcal{M})^{-1}\bar{\boldsymbol{Q}} = \left((\mathrm{Id} - \mathcal{M}_1)^{-1}\boldsymbol{Q}, \ldots, (\mathrm{Id} - \mathcal{M}_d)^{-1}\boldsymbol{Q}\right) = (\boldsymbol{Y}_1, \ldots, \boldsymbol{Y}_d) = \boldsymbol{Y}.$$

Therefore,

$$s\left((\mathrm{Id} - \mathcal{M})^{-1}\bar{\boldsymbol{Q}}\right) = s(\boldsymbol{Y}) = S(\eta, \beta) = 1.$$

Combining with (10) shows that $r \leqslant 1$, and hence $r = 1$. Therefore, we conclude $\rho(\mathcal{T}) = 1$. This finishes the proof of (b).

**Proof of (c).**    We prove

$$\rho(\mathcal{T}) < 1 \quad \Longleftrightarrow \quad \eta\lambda_{\max} < 1 - \beta^2 \text{ and } S(\eta, \beta) < 1.$$

($\Longrightarrow$) Assume $\rho(\mathcal{T}) < 1$. Set $r := \rho(\mathcal{T}) < 1$. By Theorem 4(a), there exists $\boldsymbol{W}^\star \in \mathcal{K}\setminus\{\boldsymbol{0}\}$ with $\mathcal{T}(\boldsymbol{W}^\star) = r\boldsymbol{W}^\star$. Let $s^\star := s(\boldsymbol{W}^\star)$. Then, $s^\star > 0$ by the positivity of $s(\boldsymbol{W}^\star)$. From (8), for each $i$,

$$r\boldsymbol{W}_i^\star = (\mathcal{T}(\boldsymbol{W}^\star))_i = \mathcal{M}_i(\boldsymbol{W}_i^\star) + s^\star\boldsymbol{Q},$$

or equivalently,

$$(\mathrm{Id} - \tfrac{1}{r}\mathcal{M}_i)(\boldsymbol{W}_i^\star) = \tfrac{s^\star}{r}\boldsymbol{Q}.$$

Since $\tfrac{s^\star}{r} > 0$ and $\boldsymbol{W}_i^\star \geq 0$, Lemma 3 with $c = \tfrac{1}{r}$ implies $\rho(\tfrac{1}{r}\mathcal{M}_i) < 1$, and thus $\rho(\mathcal{M}_i) < r < 1$. By Lemma 2(a),

$$\eta\lambda_i < 1 - \beta^2 \quad \text{for all } i, \qquad \text{hence} \qquad \eta\lambda_{\max} < 1 - \beta^2.$$

Moreover, since $\rho(\tfrac{1}{r}\mathcal{M}_i) < 1$ and $\mathcal{M}_i$ is $\mathbb{S}_+^2$-preserving,

$$\left(\mathrm{Id} - \tfrac{1}{r}\mathcal{M}_i\right)^{-1} = \sum_{k=0}^{\infty}\left(\tfrac{1}{r}\mathcal{M}_i\right)^k$$

is well defined and $\mathbb{S}_+^2$-preserving. Therefore

$$\boldsymbol{W}_i^\star = \tfrac{s^\star}{r}\left(\mathrm{Id} - \tfrac{1}{r}\mathcal{M}_i\right)^{-1}\boldsymbol{Q} \geq 0.$$

Since this holds for every block, $\rho(\tfrac{1}{r}\mathcal{M}) < 1$, so the block inverse $\left(\mathrm{Id} - \tfrac{1}{r}\mathcal{M}\right)^{-1}$ is well defined. Applying $s(\cdot)$ to the identity $\boldsymbol{W}^\star = \tfrac{s^\star}{r}(\mathrm{Id} - \tfrac{1}{r}\mathcal{M})^{-1}\bar{\boldsymbol{Q}}$ gives, after cancelling $s^\star > 0$,

$$r = s\left(\left(\mathrm{Id} - \tfrac{1}{r}\mathcal{M}\right)^{-1}\bar{\boldsymbol{Q}}\right). \tag{11}$$

Using the Neumann series and $r < 1$, we have

$$s\left(\left(\mathrm{Id} - \tfrac{1}{r}\mathcal{M}\right)^{-1}\bar{\boldsymbol{Q}}\right) = \sum_{k=0}^{\infty} r^{-k} s(\mathcal{M}^k(\bar{\boldsymbol{Q}})) \geqslant \sum_{k=0}^{\infty} s(\mathcal{M}^k(\bar{\boldsymbol{Q}})) = s\left((\mathrm{Id} - \mathcal{M})^{-1}\bar{\boldsymbol{Q}}\right).$$

Then from (11), we obtain

$$r \geqslant s\left((\mathrm{Id} - \mathcal{M})^{-1}\bar{\boldsymbol{Q}}\right) = s(\boldsymbol{Y}) = S(\eta, \beta).$$

Since $r < 1$, we conclude that $S(\eta, \beta) < 1$.

( $\Longleftarrow$ ) Assume $\eta\lambda_{\max} < 1 - \beta^2$ and $S(\eta, \beta) < 1$. We prove $\rho(\mathcal{T}) < 1$ by contradiction. Assume $\rho(\mathcal{T}) \geqslant 1$, and let $\boldsymbol{W}^\star \in \mathcal{K}\backslash\{\boldsymbol{0}\}$ satisfy $\mathcal{T}(\boldsymbol{W}^\star) = r\boldsymbol{W}^\star$ with $r := \rho(\mathcal{T}) \geqslant 1$. Set $s^\star := s(\boldsymbol{W}^\star) > 0$. Recall from (9) that

$$r\boldsymbol{W}^\star = \mathcal{M}(\boldsymbol{W}^\star) + s^\star\bar{\boldsymbol{Q}}, \qquad \text{so} \qquad \left(\mathrm{Id} - \tfrac{1}{r}\mathcal{M}\right)\boldsymbol{W}^\star = \tfrac{s^\star}{r}\bar{\boldsymbol{Q}}.$$

Since $\eta\lambda_{\max} < 1 - \beta^2$, Lemma 2(a) gives $\rho(\mathcal{M}_i) < 1$ for all $i$, hence $\rho(\mathcal{M}) < 1$. Since $r \geqslant 1$, we have $\rho(\tfrac{1}{r}\mathcal{M}) \leqslant \rho(\mathcal{M}) < 1$, so $\mathrm{Id} - \tfrac{1}{r}\mathcal{M}$ is invertible and

$$\boldsymbol{W}^\star = \tfrac{s^\star}{r}\left(\mathrm{Id} - \tfrac{1}{r}\mathcal{M}\right)^{-1}\bar{\boldsymbol{Q}}.$$

Applying $s(\cdot)$ and cancelling $s^\star > 0$, we obtain

$$r = s\left(\left(\mathrm{Id} - \tfrac{1}{r}\mathcal{M}\right)^{-1}\bar{\boldsymbol{Q}}\right). \tag{12}$$

Using the Neumann series, we have

$$s\left(\left(\mathrm{Id} - \tfrac{1}{r}\mathcal{M}\right)^{-1}\bar{\boldsymbol{Q}}\right) = \sum_{k=0}^{\infty} r^{-k} s(\mathcal{M}^k(\bar{\boldsymbol{Q}})) \leqslant \sum_{k=0}^{\infty} s(\mathcal{M}^k(\bar{\boldsymbol{Q}})) = s\left((\mathrm{Id} - \mathcal{M})^{-1}\bar{\boldsymbol{Q}}\right).$$

As in (b), $(\mathrm{Id} - \mathcal{M})^{-1}\bar{\boldsymbol{Q}} = \boldsymbol{Y} = (\boldsymbol{Y}_1, \ldots, \boldsymbol{Y}_d)$, hence

$$s\left((\mathrm{Id} - \mathcal{M})^{-1}\bar{\boldsymbol{Q}}\right) = s(\boldsymbol{Y}) = S(\eta, \beta).$$

Combining with (12) yields

$$r \leqslant S(\eta, \beta) < 1,$$

contradicting $r \geqslant 1$. Therefore $\rho(\mathcal{T}) < 1$. This completes the proof of (c). $\qquad \square$

**Lemma 2.** *For each $i = 1, \ldots, d$, define $\mathcal{M}_i : \mathbb{S}^2 \to \mathbb{S}^2$ by*

$$\mathcal{M}_i(\boldsymbol{X}) := \boldsymbol{A}_i\boldsymbol{X}\boldsymbol{A}_i^\top + \eta^2\lambda_i^2(\boldsymbol{X})_{11}\boldsymbol{Q} ,$$

*where $\boldsymbol{A}_i$ and $\boldsymbol{Q}$ are defined in Theorem 4, and $\eta\lambda_i > 0$. It holds that*

*(a) $\rho(\mathcal{M}_i) < 1$ if and only if $\eta\lambda_i < 1 - \beta^2$*

*(b) If $\eta\lambda_i < 1 - \beta^2$, then $\mathrm{Id} - \mathcal{M}_i$ is invertible and $(\mathrm{Id} - \mathcal{M}_i)^{-1}$ is $\mathbb{S}_+^2$-preserving.*

*(c) If $\eta\lambda_i < 1 - \beta^2$, define $\boldsymbol{Y}_i := (\mathrm{Id} - \mathcal{M}_i)^{-1}\boldsymbol{Q} \in \mathbb{S}_+^2$ and $\gamma_i := (\boldsymbol{Y}_i)_{11}$. Then,*

$$\gamma_i = \frac{1 + \beta}{2\eta\lambda_i(1 - \beta^2 - \eta\lambda_i)}.$$

*Proof of Lemma 2.* Let $\Phi : \mathbb{S}^2 \to \mathbb{R}^3$ be the isomorphism defined by

$$\Phi\left(\begin{bmatrix} x_{11} & x_{12} \\ x_{12} & x_{22} \end{bmatrix}\right) = \begin{bmatrix} x_{11} \\ x_{12} \\ x_{22} \end{bmatrix} \in \mathbb{R}^3.$$

Then, for each $i = 1, \ldots, d$, it holds that

$$\forall \boldsymbol{X} \in \mathbb{S}^2 \quad \Phi(\mathcal{M}_i(\boldsymbol{X})) = \boldsymbol{K}_i \Phi(\boldsymbol{X}), \qquad \boldsymbol{K}_i = \begin{bmatrix} 1 - 2\eta\lambda_i + 2\eta^2\lambda_i^2 & 2\beta(-1 + \eta\lambda_i) & \beta^2 \\ \eta\lambda_i - 2\eta^2\lambda_i^2 & \beta(1 - 2\eta\lambda_i) & -\beta^2 \\ 2\eta^2\lambda_i^2 & 2\beta\eta\lambda_i & \beta^2 \end{bmatrix}.$$

Since $\Phi$ is an isomorphism, $\rho(\mathcal{M}_i) = \rho(\boldsymbol{K}_i)$.

**Proof of (a): Exact condition for $\rho(\boldsymbol{K}_i) < 1$.** Fix $i$ and write $x := \eta\lambda_i > 0$ and $b := \beta \in [0, 1)$. Under the identification $\Phi : \mathbb{S}^2 \to \mathbb{R}^3$ in the proof, $\rho(\mathcal{M}_i) = \rho(\boldsymbol{K}_i)$. Let $p(r) := \det(r\boldsymbol{I} - \boldsymbol{K}_i) = r^3 + c_1 r^2 + c_2 r + c_3$. A direct expansion gives

$$c_1 = -b^2 + 2bx - b - 2x^2 + 2x - 1, \tag{13}$$
$$c_2 = b(b^2 + b + 1 - 2(b+1)x), \tag{14}$$
$$c_3 = -b^3. \tag{15}$$

We use the strict Jury criterion for a monic cubic with real coefficients: all roots of $p(r) = r^3 + c_1 r^2 + c_2 r + c_3$ lie in $\{|r| < 1\}$ if and only if

$$|c_3| < 1, \qquad p(1) > 0, \qquad 1 - c_1 + c_2 - c_3 > 0, \qquad 1 - c_3^2 > |c_2 - c_1 c_3|. \tag{16}$$

We verify (16) and show that they hold if and only if $0 < x < 1 - b^2$.

First, from (15), $|c_3| = b^3 < 1$, so the first condition in (16) holds.

A direct substitution of (13),(14), and (15) gives

$$p(1) = 1 + c_1 + c_2 + c_3 = 2x(1 - b^2 - x).$$

Hence,

$$p(1) > 0 \quad \Longleftrightarrow \quad 0 < x < 1 - b^2.$$

Next, substitution of (13),(14), and (15) gives

$$1 - c_1 + c_2 - c_3 = 2\phi(x), \qquad \phi(x) := x^2 - (b+1)^2 x + (b^3 + b^2 + b + 1).$$

The discriminant of $\phi$ is

$$\Delta_\phi = (b+1)^4 - 4(b^3 + b^2 + b + 1) = b^4 + 2b^2 - 3 = (b^2 - 1)(b^2 + 3) < 0 \qquad (b \in [0, 1)),$$

and $\phi$ has leading coefficient $1 > 0$, so $\phi(x) > 0$ for all $x \in \mathbb{R}$. Therefore $1 - c_1 + c_2 - c_3 > 0$ holds automatically for all $x \geqslant 0$ and $b \in [0, 1)$.

It remains to check the last condition in (16): $1 - c_3^2 > |c_2 - c_1 c_3|$. Again, by substituting (13),(14), and (15) gives

$$(1 - c_3^2) - (c_2 - c_1 c_3) = 2b^3 x^2 + 2b(1 + b)^2(1 - b)x + (1 + b)(1 - b)^3(1 + b + b^2) > 0$$

for any $x \geqslant 0$ and $b \in [0, 1)$. Moreover, we have

$$(1 - c_3^2) + (c_2 - c_1 c_3) = -2b^3 x^2 - 2b(1 + b)^2(1 - b)x + (1 + b)(1 - b)(1 + b^2)(1 + b + b^2) := \psi(x).$$

If $b = 0$, then $\psi(x) = 1 > 0$. Otherwise, $b \in (0, 1)$, and $\psi$ is a concave quadratic and has a global maximum at $x < 0$. Hence, if $0 < x < 1 - b^2$, then $\psi(x) > \min\{\psi(0), \psi(1 - b^2)\}$. Note that

$$\psi(0) = (1 + b)(1 - b)(1 + b^2)(1 + b + b^2) > 0 \, ,$$
$$\psi(1 - b^2) = (1 + b)(1 - b)(1 + b + b^2)\big[(1 - b)^2 + 2b^3\big] > 0 \, ,$$

for all $b \in (0, 1)$. Therefore we conclude that all the strict Jury conditions (16) hold if and only if $0 < x < 1 - b^2$. Putting $x = \eta\lambda_i$ and $b = \beta$, we conclude

$$\rho(\mathcal{M}_i) = \rho(\boldsymbol{K}_i) < 1 \quad \Longleftrightarrow \quad 0 < \eta\lambda_i < 1 - \beta^2,$$

which proves part (a).

**Proof of (b).** Assume that $\eta\lambda_i < 1 - \beta^2$. Note that

$$\det(\boldsymbol{I} - \boldsymbol{K}_i) = 2\eta\lambda_i(1 - \beta^2 - \eta\lambda_i) \neq 0,$$

so $\boldsymbol{I} - \boldsymbol{K}_i$ is invertible, and hence $\mathrm{Id} - \mathcal{M}_i$ is also invertible. According to (a), we have $\rho(\mathcal{M}_i) < 1$, and

$$(\mathrm{Id} - \mathcal{M}_i)^{-1} = \sum_{k=0}^{\infty} \mathcal{M}_i^k$$

converges in operator norm (finite-dimensional Neumann series). Since $\mathcal{M}_i(\mathbb{S}_+^2) \subseteq \mathbb{S}_+^2$, we have $\mathcal{M}_i^k(\mathbb{S}_+^2) \subseteq \mathbb{S}_+^2$ for any $k \geq 0$, and thus

$$(\mathrm{Id} - \mathcal{M}_i)^{-1}(\mathbb{S}_+^2) \subseteq \mathbb{S}_+^2.$$

This concludes the proof of (b).

**Proof of (c).** Assume that $\eta\lambda_i < 1 - \beta^2$ and define $\boldsymbol{Y}_i := (\mathrm{Id} - \mathcal{M}_i)^{-1}\boldsymbol{Q} \in \mathbb{S}_+^2$. In $\mathbb{R}^3$, we have

$$\begin{bmatrix} (\boldsymbol{Y}_i)_{11} \\ (\boldsymbol{Y}_i)_{12} \\ (\boldsymbol{Y}_i)_{22} \end{bmatrix} = \Phi(\boldsymbol{Y}_i) = (\boldsymbol{I} - \boldsymbol{K}_i)^{-1}\Phi(\boldsymbol{Q}) = \begin{bmatrix} 2\eta\lambda_i - 2\eta^2\lambda_i^2 & 2\beta(1 - \eta\lambda_i) & -\beta^2 \\ -\eta\lambda_i + 2\eta^2\lambda_i^2 & 1 - \beta(1 - 2\eta\lambda_i) & \beta^2 \\ -2\eta^2\lambda_i^2 & -2\beta\eta\lambda_i & 1 - \beta^2 \end{bmatrix}^{-1} \begin{bmatrix} 1 \\ -1 \\ 1 \end{bmatrix}.$$

Solving this $3 \times 3$ system and extracting the first coordinate gives

$$\gamma_i := (\boldsymbol{Y}_i)_{11} = \frac{1 + \beta}{2\eta\lambda_i(1 - \beta^2 - \eta\lambda_i)}.$$

This concludes the proof of (c). □

**Lemma 3.** *Let $\mathcal{M}_i$ denote a linear map defined as in Lemma 2. For any $c > 0$, if $\rho(c\mathcal{M}_i) \geq 1$, then for every $\alpha > 0$, there does not exist $\boldsymbol{W} \succeq 0$ such that*

$$(\mathrm{Id} - c\mathcal{M}_i)(\boldsymbol{W}) \succeq \alpha\boldsymbol{Q} \, ,$$

*where $\boldsymbol{Q}$ is defined in Theorem 4.*

*Proof of Lemma 3.* Define $\mathcal{N}_i := c\mathcal{M}_i$. Since $\mathcal{M}_i$ is $\mathbb{S}_+^2$-preserving, so is $\mathcal{N}_i$. Hence its adjoint $\mathcal{N}_i^*$ preserves the dual cone, which is again $\mathbb{S}_+^2$. Since $\mathbb{S}_+^2 \subset \mathbb{S}^2$ is a closed, convex, pointed cone with nonempty interior, the Krein–Rutman Theorem (see also Lemma 6) implies that there exists $\boldsymbol{Y} \succeq 0$ with $\boldsymbol{Y} \neq \boldsymbol{0}$ such that

$$\mathcal{N}_i^*(\boldsymbol{Y}) = \rho(\mathcal{N}_i)\boldsymbol{Y}.$$

Define $\mathcal{L}_i := \mathrm{Id} - \mathcal{N}_i$. Then, its adjoint $\mathcal{L}_i^* = \mathrm{Id} - \mathcal{N}_i^*$ satisfies

$$\mathcal{L}_i^*(\boldsymbol{Y}) = (1 - \rho(\mathcal{N}_i))\boldsymbol{Y} \preceq 0,$$

since $\rho(\mathcal{N}_i) \geqslant 1$. Since $\boldsymbol{Y} \succeq 0$ and $\boldsymbol{Q} \succeq 0$, we have $\langle \boldsymbol{Y}, \boldsymbol{Q} \rangle \geqslant 0$.

Now, we show that $\langle \boldsymbol{Y}, \boldsymbol{Q} \rangle > 0$. Assume the contrary that $\langle \boldsymbol{Y}, \boldsymbol{Q} \rangle = 0$. Then,

$$\langle \boldsymbol{Y}, \boldsymbol{Q} \rangle = \mathrm{Tr}(\boldsymbol{Y}\boldsymbol{Q}) = \mathrm{Tr}(\boldsymbol{Y}^{1/2}\boldsymbol{Q}\boldsymbol{Y}^{1/2}) = 0.$$

Since $\boldsymbol{Y}^{1/2}\boldsymbol{Q}\boldsymbol{Y}^{1/2} \succeq 0$ and $\mathrm{Tr}(\boldsymbol{Y}^{1/2}\boldsymbol{Q}\boldsymbol{Y}^{1/2}) = 0$, we have $\boldsymbol{Y}^{1/2}\boldsymbol{Q}\boldsymbol{Y}^{1/2} = \boldsymbol{0}$ and thus $\boldsymbol{Q}\boldsymbol{Y} = \boldsymbol{0}$. Note that $\boldsymbol{Q} = \begin{bmatrix} 1 & -1 \end{bmatrix}^\top \begin{bmatrix} 1 & -1 \end{bmatrix}$ has null space $\mathrm{Null}(\boldsymbol{Q}) = \mathrm{span}(\begin{bmatrix} 1 & 1 \end{bmatrix}^\top)$. Since $\boldsymbol{Q}\boldsymbol{Y} = \boldsymbol{0}$ and $\boldsymbol{Y} \neq \boldsymbol{0}$, we have

$$\mathrm{Range}(\boldsymbol{Y}) \subseteq \mathrm{Null}(\boldsymbol{Q}) = \mathrm{span}\left(\begin{bmatrix} 1 \\ 1 \end{bmatrix}\right), \quad \text{and hence} \quad \boldsymbol{Y} = a \begin{bmatrix} 1 \\ 1 \end{bmatrix} \begin{bmatrix} 1 & 1 \end{bmatrix} = a \begin{bmatrix} 1 & 1 \\ 1 & 1 \end{bmatrix} \text{ for some } a > 0.$$

Then, it holds that

$$\begin{aligned}
\mathcal{N}_i^*(\boldsymbol{Y}) &= c\boldsymbol{A}_i^\top \boldsymbol{Y}\boldsymbol{A}_i + c\eta^2\lambda_i^2 \langle \boldsymbol{Y}, \boldsymbol{Q} \rangle \begin{bmatrix} 1 & 0 \\ 0 & 0 \end{bmatrix} \\
&= c\boldsymbol{A}_i^\top \boldsymbol{Y}\boldsymbol{A}_i \\
&= ca \left( \boldsymbol{A}_i^\top \begin{bmatrix} 1 \\ 1 \end{bmatrix} \right) \left( \boldsymbol{A}_i^\top \begin{bmatrix} 1 \\ 1 \end{bmatrix} \right)^\top \\
&= ca \begin{bmatrix} 1 & 0 \\ 0 & 0 \end{bmatrix}
\end{aligned}$$

However,

$$ca \begin{bmatrix} 1 & 0 \\ 0 & 0 \end{bmatrix} = \mathcal{N}_i^*(\boldsymbol{Y}) = \rho(\mathcal{N}_i)\boldsymbol{Y} = \rho(\mathcal{N}_i)a \begin{bmatrix} 1 & 1 \\ 1 & 1 \end{bmatrix}$$

gives a contradiction. Hence, $\langle \boldsymbol{Y}, \boldsymbol{Q} \rangle > 0$.

Finally, assume for contradiction that there exists $\boldsymbol{W} \succeq 0$ such that

$$(\mathrm{Id} - c\mathcal{M}_i)(\boldsymbol{W}) \succeq \alpha \boldsymbol{Q}.$$

Then,

$$\langle \boldsymbol{Y}, (\mathrm{Id} - c\mathcal{M}_i)(\boldsymbol{W}) \rangle \geqslant \alpha \langle \boldsymbol{Y}, \boldsymbol{Q} \rangle > 0.$$

However, since $\boldsymbol{W} \succeq 0$ and $\mathcal{L}_i^*(\boldsymbol{Y}) \preceq 0$, it holds that

$$\langle \boldsymbol{Y}, (\mathrm{Id} - c\mathcal{M}_i)(\boldsymbol{W}) \rangle = \langle (\mathrm{Id} - c\mathcal{M}_i^*)(\boldsymbol{Y}), \boldsymbol{W} \rangle = \langle \mathcal{L}_i^*(\boldsymbol{Y}), \boldsymbol{W} \rangle \leqslant 0,$$

a contradiction. This concludes the proof of Lemma 3. $\qquad\square$

### A.3. Proof of Theorem 2

We prove Theorem 2 by characterizing mean-square linear stability of the linearized ZO-GDM dynamics. Specifically, if

$$\eta\lambda_{\max}(\boldsymbol{H}) < 1 - \beta^2 \quad \text{and} \quad \sum_{i=1}^d \frac{\eta\lambda_i}{2(1-\beta)\left(1 - \frac{\eta\lambda_i}{1-\beta^2}\right)} < 1, \tag{17}$$

then the linearized dynamics is mean-square stable. Conversely, mean-square linear stability implies

$$\eta\lambda_{\max}(\boldsymbol{H}) < 1 - \beta^2 \quad \text{and} \quad \sum_{i=1}^d \frac{\eta\lambda_i}{2(1-\beta)\left(1 - \frac{\eta\lambda_i}{1-\beta^2}\right)} \leqslant 1. \tag{18}$$

By definition, the mean-square critical step size $\eta_{\mathrm{ms}}^{\star}$ is therefore the unique $\eta > 0$ satisfying

$$\eta\lambda_{\max}(\boldsymbol{H}) < 1 - \beta^2 \quad \text{and} \quad \sum_{i=1}^{d} \frac{\eta\lambda_i}{2(1-\beta)\left(1 - \frac{\eta\lambda_i}{1-\beta^2}\right)} = 1. \tag{19}$$

Moreover, $\eta_{\mathrm{ms}}^{\star}$ obeys the bounds

$$\frac{2(1-\beta)}{\mathrm{Tr}(\boldsymbol{H}) + \frac{2\lambda_{\max}(\boldsymbol{H})}{1+\beta}} \leqslant \eta_{\mathrm{ms}}^{\star} \leqslant \frac{2(1-\beta)}{\mathrm{Tr}(\boldsymbol{H})}. \tag{20}$$

*Proof of Theorem 2.* Without loss of generality, we assume $\boldsymbol{x}^{\star} = \boldsymbol{0}$ and diagonalize the Hessian as $\boldsymbol{H} = \boldsymbol{\Lambda} = \mathrm{diag}(\lambda_1, \ldots, \lambda_d)$ with $\lambda_1 \geqslant \cdots \geqslant \lambda_d \geqslant 0$.

By Lemma 1, the covariance blocks evolve according to $\mathcal{T}(\boldsymbol{W}_{1,t}, \ldots, \boldsymbol{W}_{d,t}) = (\boldsymbol{W}_{1,t+1}, \ldots, \boldsymbol{W}_{d,t+1})$, and

$$\mathbb{E}\|\boldsymbol{x}_t\|^2 = \sum_{i=1}^{d}(\boldsymbol{W}_{i,t})_{11}.$$

Thus mean-square linear stability is equivalent to uniform boundedness of $\sum_{i=1}^{d}(\boldsymbol{W}_{i,t})_{11}$ over time, for every initialization.

**Reduction to strictly positive eigenvalues.** Let $P := \{i : \lambda_i > 0\}$ and $Z := \{i : \lambda_i = 0\}$. The coordinates indexed by $P$ form an autonomous subsystem. Define the restricted operator $\mathcal{T}_P : (\mathbb{S}^2)^{|P|} \to (\mathbb{S}^2)^{|P|}$ by

$$(\mathcal{T}_P(\boldsymbol{W}))_i = \boldsymbol{A}_i \boldsymbol{W}_i \boldsymbol{A}_i^{\top} + \eta^2\left(\lambda_i^2(\boldsymbol{W}_i)_{11} + \sum_{j\in P}\lambda_j^2(\boldsymbol{W}_j)_{11}\right)\boldsymbol{Q}, \qquad i \in P. \tag{21}$$

Since all $\lambda_i > 0$ for $i \in P$, Theorem 4 applies directly to $\mathcal{T}_P$. In particular,

$$\rho(\mathcal{T}_P) < 1 \iff \eta\lambda_{\max}(\boldsymbol{H}) < 1 - \beta^2 \text{ and } \sum_{i\in P}\frac{\eta\lambda_i}{2(1-\beta)\left(1 - \frac{\eta\lambda_i}{1-\beta^2}\right)} < 1,$$

with the corresponding statement for $\rho(\mathcal{T}_P) \leqslant 1$.

It therefore suffices to relate $\rho(\mathcal{T}_P)$ to uniform boundedness of $\sum_{i=1}^{d}(\boldsymbol{W}_{i,t})_{11}$ over time, for every initialization. In particular, it suffices to prove that:

$$\rho(\mathcal{T}_P) < 1 \implies \sup_{t\geqslant 0}\sum_{i=1}^{d}(\boldsymbol{W}_{i,t})_{11} < \infty \text{ for every initialization,}$$

and

$$\sup_{t\geqslant 0}\sum_{i=1}^{d}(\boldsymbol{W}_{i,t})_{11} < \infty \text{ for every initialization} \implies \rho(\mathcal{T}_P) \leqslant 1.$$

**(i)** Proof of $(\sup_{t\geqslant 0}\sum_{i=1}^{d}(\boldsymbol{W}_{i,t})_{11} < \infty$ for every initialization $\Rightarrow \rho(\mathcal{T}_P) \leqslant 1)$. We prove the contrapositive: if $\rho(\mathcal{T}_P) > 1$, then there exists an initialization for which $\sum_{i=1}^{d}(\boldsymbol{W}_{i,t})_{11}$ is unbounded. Assume $\rho(\mathcal{T}_P) > 1$ and set $r := \rho(\mathcal{T}_P)$. Recall that by Theorem 4(a), there exists $\boldsymbol{W}^{\star} \in \mathcal{K}_P\backslash\{\boldsymbol{0}\}$ such that

$$\mathcal{T}_P(\boldsymbol{W}^{\star}) = r\boldsymbol{W}^{\star} \quad \text{and} \quad \sum_{i\in P}(\boldsymbol{W}_i^{\star})_{11} > 0. \tag{22}$$

Extend $\boldsymbol{W}^{\star}$ to $\overline{\boldsymbol{W}}^{\star} \in \mathcal{K}$ by setting $(\overline{\boldsymbol{W}}^{\star})_i = \boldsymbol{W}_i^{\star}$ for $i \in P$ and $(\overline{\boldsymbol{W}}^{\star})_i = \boldsymbol{0}$ for $i \in Z$. Since the $P$-subsystem is autonomous, by (22) it holds that

$$(\mathcal{T}^t(\overline{\boldsymbol{W}}^{\star}))_i = r^t\,\boldsymbol{W}_i^{\star}$$

for all $t \geqslant 0$ and coordinate $i \in P$. Consequently, we have

$$\mathcal{T}^t(\overline{W}^\star) \geq r^t \, \overline{W}^\star \tag{23}$$

for all $t \geqslant 0$.

Initialize $m_0 = 0$ and $x_0 = (x_{0,0}, x_{1,0}, \ldots, x_{d,0}) \in \mathbb{R}^d$ such that

$$W_{i,0} = \begin{bmatrix} x_{i,0}^2 & 0 \\ 0 & 0 \end{bmatrix}, \qquad x_{i,0}^2 := (\overline{W}_i^\star)_{11}.$$

Denote $s_0 := \sum_{i=1}^d \eta^2 \lambda_i^2 (W_{i,0})_{11} = \sum_{i \in P} \eta^2 \lambda_i^2 (W_i^\star)_{11}$. Note that $s_0 > 0$ by (22). Moreover, it holds that

$$W_{i,1} = A_i W_{i,0} A_i^\top + \eta^2 \lambda_i^2 (W_{i,0})_{11} Q + s_0 Q \geq s_0 Q, \qquad i = 1, \ldots, d.$$

Equivalently,

$$W_1 \geq s_0 \, \bar{Q}, \qquad \bar{Q} := (Q, \ldots, Q) \in \mathcal{K}. \tag{24}$$

Since $\overline{W}^\star \in \mathcal{K}$ and each block $Q$ is positive definite on its support, there exists $\alpha > 0$ such that

$$\alpha \, \overline{W}^\star \leq \bar{Q}. \tag{25}$$

Combining (24) and (25), we have

$$W_1 \geq s_0 \alpha \, \overline{W}^\star.$$

Applying $\mathcal{T}^{t-1}$ and using positivity and linearity of $\mathcal{T}$ with (23), we obtain for all $t \geqslant 1$,

$$W_t = \mathcal{T}^{t-1}(W_1) \geq s_0 \alpha \, \mathcal{T}^{t-1}(\overline{W}^\star) \geq s_0 \alpha \, r^{t-1} \overline{W}^\star.$$

Define $\ell(W) := \sum_{i=1}^d (W_i)_{11}$, which is monotone on $\mathcal{K}$. Applying $\ell$ gives

$$\sum_{i=1}^d (W_{i,t})_{11} \geqslant s_0 \alpha \, r^{t-1} \sum_{i=1}^d (\overline{W}_i^\star)_{11} \xrightarrow[t \to \infty]{} \infty,$$

since $r > 1$. Thus, when $\rho(\mathcal{T}_P) > 1$, boundedness of $\sum_{i=1}^d (W_{i,t})_{11}$ fails for the constructed initialization. This proves the contrapositive, and therefore $\rho(\mathcal{T}_P) \leqslant 1$ whenever $\sum_{i=1}^d (W_{i,t})_{11}$ is bounded for all initializations.

**(ii)** Proof of $(\rho(\mathcal{T}_P) < 1 \Rightarrow \sup_{t \geqslant 0} \sum_{i=1}^d (W_{i,t})_{11} < \infty$ for every initialization). Assume $\rho(\mathcal{T}_P) < 1$. Fix any initial condition $(W_{i,0})_{i \in [d]} \in \mathcal{K}$ and let $(W_{i,t})_{i \in [d]} := \mathcal{T}^t(W_{1,0}, \ldots, W_{d,0})$. Define the coupling scalar

$$s_t := \sum_{j=1}^d \lambda_j^2 (W_{j,t})_{11} = \sum_{j \in P} \lambda_j^2 (W_{j,t})_{11}.$$

Then the $P$-coordinates evolve autonomously according to (21) with initial condition $(W_{i,0})_{i \in P}$, hence

$$(W_{i,t})_{i \in P} = \mathcal{T}_P^t \big( (W_{i,0})_{i \in P} \big).$$

Since $\rho(\mathcal{T}_P) < 1$, there exist $C < \infty$ and $\rho \in (0, 1)$ such that

$$\sum_{i \in P} \|W_{i,t}\| \leqslant C \rho^t \sum_{i \in P} \|W_{i,0}\|, \qquad \forall \, t \geqslant 0, \tag{26}$$

and in particular $s_t \leqslant \lambda_{\max}^2 \sum_{i \in P} \|W_{i,t}\|$ is summable:

$$\sum_{t=0}^\infty s_t < \infty. \tag{27}$$

For $i \in Z$, we have $\lambda_i = 0$ and $\boldsymbol{A}_i = \boldsymbol{A}_0 := \begin{bmatrix} 1 & -\beta \\ 0 & \beta \end{bmatrix}$, so

$$\boldsymbol{W}_{i,t+1} = \boldsymbol{A}_0 \boldsymbol{W}_{i,t} \boldsymbol{A}_0^\top + \eta^2 s_t \boldsymbol{Q}.$$

Unrolling gives

$$\boldsymbol{W}_{i,t} = \boldsymbol{A}_0^t \boldsymbol{W}_{i,0} (\boldsymbol{A}_0^\top)^t + \eta^2 \sum_{k=0}^{t-1} \boldsymbol{A}_0^{t-1-k} (s_k \boldsymbol{Q}) (\boldsymbol{A}_0^\top)^{t-1-k}. \tag{28}$$

Since $\beta \in [0, 1)$, $\sup_{t \geqslant 0} \|\boldsymbol{A}_0^t\| < \infty$ and $\sup_{r \geqslant 0} \|\boldsymbol{A}_0^r \boldsymbol{Q} (\boldsymbol{A}_0^\top)^r\| < \infty$. Combining these bounds with (27) and (28) yields $\sup_{t \geqslant 0} \|\boldsymbol{W}_{i,t}\| < \infty$ for each $i \in Z$. Together with (26) and finiteness of $Z$, we obtain $\sup_{t \geqslant 0} \|\mathcal{T}^t(\boldsymbol{W}_{1,0}, \ldots, \boldsymbol{W}_{d,0})\| < \infty$ for every initial condition. Therefore, $\sup_{t \geqslant 0} \sum_{i=1}^d (\boldsymbol{W}_{i,t})_{11}$ is bounded for all initializations.

This proves the explicit mean-square stability conditions (17) and (18).

**Critical step size and bounds.** Define

$$S(\eta, \beta) := \sum_{i=1}^d \frac{\eta \lambda_i}{2(1 - \beta)\left(1 - \frac{\eta \lambda_i}{1 - \beta^2}\right)}.$$

On $\eta \in \left(0, \frac{1 - \beta^2}{\lambda_{\max}(\boldsymbol{H})}\right)$, each summand is strictly increasing in $\eta$, hence so is $S(\eta, \beta)$. Therefore $\eta_{\text{ms}}^\star$ is the unique $\eta > 0$ satisfying (19).

For the upper bound in (20), using $1 - \frac{\eta_{\text{ms}}^\star \lambda_i}{1 - \beta^2} \leqslant 1$ gives

$$1 = \sum_{i=1}^d \frac{\eta_{\text{ms}}^\star \lambda_i}{2(1 - \beta)\left(1 - \frac{\eta_{\text{ms}}^\star \lambda_i}{1 - \beta^2}\right)} \geqslant \sum_{i=1}^d \frac{\eta_{\text{ms}}^\star \lambda_i}{2(1 - \beta)} = \frac{\eta_{\text{ms}}^\star \operatorname{Tr}(\boldsymbol{H})}{2(1 - \beta)},$$

so $\eta_{\text{ms}}^\star \leqslant \frac{2(1 - \beta)}{\operatorname{Tr}(\boldsymbol{H})}$.

For the lower bound, since $\lambda_i \leqslant \lambda_{\max}(\boldsymbol{H})$,

$$1 - \frac{\eta_{\text{ms}}^\star \lambda_i}{1 - \beta^2} \geqslant 1 - \frac{\eta_{\text{ms}}^\star \lambda_{\max}(\boldsymbol{H})}{1 - \beta^2},$$

and hence

$$1 = \sum_{i=1}^d \frac{\eta_{\text{ms}}^\star \lambda_i}{2(1 - \beta)\left(1 - \frac{\eta_{\text{ms}}^\star \lambda_i}{1 - \beta^2}\right)} \leqslant \frac{\eta_{\text{ms}}^\star \operatorname{Tr}(\boldsymbol{H})}{2(1 - \beta)\left(1 - \frac{\eta_{\text{ms}}^\star \lambda_{\max}(\boldsymbol{H})}{1 - \beta^2}\right)}.$$

Rearranging, we obtain

$$\eta_{\text{ms}}^\star \geqslant \frac{2(1 - \beta)}{\operatorname{Tr}(\boldsymbol{H}) + \frac{2\lambda_{\max}(\boldsymbol{H})}{1 + \beta}},$$

which completes the proof. $\qquad\square$

### A.4. Proof of Theorem 3

We prove Theorem 3 by characterizing mean-square linear stability of the linearized Frozen ZO-Adam dynamics. Specifically, if We prove the following implication for mean-square stability: if

$$\eta \lambda_{\max}(\boldsymbol{P}^{-1}\boldsymbol{H}) < 1 + \beta_1 \quad \text{and} \quad \sum_{i=1}^d \frac{\eta \tilde{\lambda}_i}{2\left(1 - \frac{\eta \tilde{\lambda}_i}{1 + \beta_1}\right)} < 1,$$

then the linearized dynamics of frozen ZO-Adam is mean-square linearly stable. Conversely, if the linearized dynamics is mean-square linearly stable, then

$$\eta\lambda_{\max}(\boldsymbol{P}^{-1}\boldsymbol{H}) < 1 + \beta_1 \quad \text{and} \quad \sum_{i=1}^{d} \frac{\eta\tilde{\lambda}_i}{2\left(1 - \frac{\eta\tilde{\lambda}_i}{1+\beta_1}\right)} \leqslant 1.$$

By definition of the mean-square critical step size $\eta^\star_{\mathrm{ms}}$, it follows that $\eta^\star_{\mathrm{ms}}$ is the unique $\eta > 0$ satisfying

$$\eta\lambda_{\max}(\boldsymbol{P}^{-1}\boldsymbol{H}) < 1 + \beta_1 \quad \text{and} \quad \sum_{i=1}^{d} \frac{\eta\tilde{\lambda}_i}{2\left(1 - \frac{\eta\tilde{\lambda}_i}{1+\beta_1}\right)} = 1.$$

Moreover, $\eta^\star_{\mathrm{ms}}$ satisfies

$$\frac{2}{\mathrm{Tr}(\boldsymbol{P}^{-1}\boldsymbol{H}) + \frac{2\lambda_{\max}(\boldsymbol{P}^{-1}\boldsymbol{H})}{1+\beta_1}} \leqslant \eta^\star_{\mathrm{ms}} \leqslant \frac{2}{\mathrm{Tr}(\boldsymbol{P}^{-1}\boldsymbol{H})}.$$

**Why the commutativity assumption $\boldsymbol{P}\boldsymbol{H} = \boldsymbol{H}\boldsymbol{P}$ is needed.** The mean-square analysis for ZO methods closes on the coordinatewise $2 \times 2$ covariance blocks only after rotating to a basis where the quadratic is diagonal: the global coupling term is $\mathrm{Tr}(\boldsymbol{H}\boldsymbol{X}_t\boldsymbol{H}) = \sum_i \lambda_i^2(\boldsymbol{X}_t)_{ii}$, and this structure is what yields the rank-one coupling operator in Theorem 4. For frozen ZO-Adam, the update involves both $\boldsymbol{H}$ and the fixed preconditioner $\boldsymbol{P}^{-1}$; to reduce the dynamics to independent eigencoordinates with a single scalar coupling, we need a basis that diagonalizes $\boldsymbol{H}$ and $\boldsymbol{P}$ simultaneously. The condition $\boldsymbol{P}\boldsymbol{H} = \boldsymbol{H}\boldsymbol{P}$ (with $\boldsymbol{P} > 0$ and $\boldsymbol{H} \geq 0$) guarantees such a simultaneous orthogonal diagonalization, so that the recursion depends only on the eigenvalues $\{\tilde{\lambda}_i\}$ of $\boldsymbol{P}^{-1}\boldsymbol{H}$ and matches the ZO-GDM covariance-operator form after a change of variables. Without commutativity, $\boldsymbol{P}^{-1/2}\boldsymbol{H}\boldsymbol{P}^{-1/2}$ need not be diagonalizable in the same basis as the random rank-one factor, and the diagonal-block closure used in Theorem 4 generally fails.

*Proof of Theorem 3.* We assume $\boldsymbol{x}^\star = \boldsymbol{0}$ without loss of generality, so $f(\boldsymbol{x}) = \frac{1}{2}\boldsymbol{x}^\top\boldsymbol{H}\boldsymbol{x}$ and $\widehat{\nabla}f(\boldsymbol{x}_t) = \boldsymbol{u}_t\boldsymbol{u}_t^\top\boldsymbol{H}\boldsymbol{x}_t$ with $\boldsymbol{u}_t \sim \mathcal{N}(\boldsymbol{0}, \boldsymbol{I})$ i.i.d. Frozen ZO-Adam is

$$\boldsymbol{m}_{t+1} = \beta_1\boldsymbol{m}_t + (1 - \beta_1)\,\boldsymbol{u}_t\boldsymbol{u}_t^\top\boldsymbol{H}\boldsymbol{x}_t, \tag{29}$$

$$\boldsymbol{x}_{t+1} = \boldsymbol{x}_t - \eta\,\boldsymbol{P}^{-1}\boldsymbol{m}_{t+1}. \tag{30}$$

**Preconditioned coordinates.** Define $\tilde{\eta} := (1 - \beta_1)\eta$ and the change of variables

$$\tilde{\boldsymbol{x}}_t := \boldsymbol{P}^{1/2}\boldsymbol{x}_t, \qquad \tilde{\boldsymbol{m}}_t := \boldsymbol{P}^{-1/2}\boldsymbol{m}_t, \qquad \tilde{\boldsymbol{H}} := \boldsymbol{P}^{-1/2}\boldsymbol{H}\boldsymbol{P}^{-1/2}.$$

Multiplying (29) by $\boldsymbol{P}^{-1/2}$ and (30) by $\boldsymbol{P}^{1/2}$ gives

$$\tilde{\boldsymbol{m}}_{t+1} = \beta_1\tilde{\boldsymbol{m}}_t + (1 - \beta_1)\,\boldsymbol{P}^{-1/2}\boldsymbol{u}_t\boldsymbol{u}_t^\top\boldsymbol{P}^{1/2}\,\tilde{\boldsymbol{H}}\,\tilde{\boldsymbol{x}}_t,$$

$$\tilde{\boldsymbol{x}}_{t+1} = \tilde{\boldsymbol{x}}_t - \eta\,\tilde{\boldsymbol{m}}_{t+1} = \tilde{\boldsymbol{x}}_t - \tilde{\eta}\,\hat{\boldsymbol{m}}_{t+1},$$

where we introduced the scaled momentum $\hat{\boldsymbol{m}}_t := \tilde{\boldsymbol{m}}_t/(1 - \beta_1)$, so that

$$\hat{\boldsymbol{m}}_{t+1} = \beta_1\hat{\boldsymbol{m}}_t + \boldsymbol{P}^{-1/2}\boldsymbol{u}_t\boldsymbol{u}_t^\top\boldsymbol{P}^{1/2}\,\tilde{\boldsymbol{H}}\,\tilde{\boldsymbol{x}}_t, \qquad \tilde{\boldsymbol{x}}_{t+1} = \tilde{\boldsymbol{x}}_t - \tilde{\eta}\,\hat{\boldsymbol{m}}_{t+1}. \tag{31}$$

The linear map $(\boldsymbol{x}_t, \boldsymbol{m}_t) \leftrightarrow (\tilde{\boldsymbol{x}}_t, \hat{\boldsymbol{m}}_t)$ is invertible, hence $\sup_t \mathbb{E}\|\boldsymbol{x}_t\|^2 < \infty$ is equivalent to $\sup_t \mathbb{E}\|\tilde{\boldsymbol{x}}_t\|^2 < \infty$.

**Simultaneous diagonalization.** Since $\boldsymbol{P}\boldsymbol{H} = \boldsymbol{H}\boldsymbol{P}$ with $\boldsymbol{P} > 0$ and $\boldsymbol{H} \geq 0$ symmetric, there exists an orthogonal $\boldsymbol{U}$ and diagonal matrices $\boldsymbol{\Sigma} = \mathrm{diag}(\sigma_1, \ldots, \sigma_d)$ with $\sigma_i > 0$ and $\boldsymbol{\Lambda} = \mathrm{diag}(\lambda_1, \ldots, \lambda_d)$ with $\lambda_i \geqslant 0$ such that

$$\boldsymbol{P} = \boldsymbol{U}\boldsymbol{\Sigma}\boldsymbol{U}^\top, \qquad \boldsymbol{H} = \boldsymbol{U}\boldsymbol{\Lambda}\boldsymbol{U}^\top.$$

Consequently,

$$\tilde{\boldsymbol{H}} = \boldsymbol{P}^{-1/2}\boldsymbol{H}\boldsymbol{P}^{-1/2} = \boldsymbol{U}\,\mathrm{diag}\!\left(\tfrac{\lambda_1}{\sigma_1}, \ldots, \tfrac{\lambda_d}{\sigma_d}\right)\boldsymbol{U}^\top =: \boldsymbol{U}\tilde{\boldsymbol{\Lambda}}\boldsymbol{U}^\top,$$

so $\tilde{\lambda}_i := \lambda_i/\sigma_i$ are the eigenvalues of $\boldsymbol{P}^{-1}\boldsymbol{H}$.

Let $\bar{\boldsymbol{x}}_t := \boldsymbol{U}^\top\tilde{\boldsymbol{x}}_t$, $\bar{\boldsymbol{m}}_t := \boldsymbol{U}^\top\hat{\boldsymbol{m}}_t$, and $\bar{\boldsymbol{u}}_t := \boldsymbol{U}^\top\boldsymbol{u}_t$. Then $\bar{\boldsymbol{u}}_t \sim \mathcal{N}(\boldsymbol{0}, \boldsymbol{I})$ i.i.d., and (31) becomes

$$\bar{\boldsymbol{m}}_{t+1} = \beta_1\bar{\boldsymbol{m}}_t + \boldsymbol{\Sigma}^{-1/2}\bar{\boldsymbol{u}}_t\bar{\boldsymbol{u}}_t^\top\boldsymbol{\Sigma}^{1/2}\tilde{\boldsymbol{\Lambda}}\,\bar{\boldsymbol{x}}_t, \qquad \bar{\boldsymbol{x}}_{t+1} = \bar{\boldsymbol{x}}_t - \tilde{\eta}\,\bar{\boldsymbol{m}}_{t+1}.$$

Since $\|\tilde{\boldsymbol{x}}_t\| = \|\bar{\boldsymbol{x}}_t\|$ and $\mathbb{E}\|\boldsymbol{x}_t\|^2 \leqslant \lambda_{\min}(\boldsymbol{P})^{-1}\mathbb{E}\|\tilde{\boldsymbol{x}}_t\|^2$, it suffices to analyze mean-square boundedness of $\bar{\boldsymbol{x}}_t$.

**Second-moment recursion and covariance operator.** Define

$$\boldsymbol{X}_t := \mathbb{E}[\bar{\boldsymbol{x}}_t\bar{\boldsymbol{x}}_t^\top], \qquad \boldsymbol{Y}_t := \mathbb{E}[\tilde{\eta}^2\bar{\boldsymbol{m}}_t\bar{\boldsymbol{m}}_t^\top], \qquad \boldsymbol{C}_t := \mathbb{E}[\tilde{\eta}\,\bar{\boldsymbol{x}}_t\bar{\boldsymbol{m}}_t^\top].$$

Applying Lemma 5 with $\boldsymbol{P} = \boldsymbol{\Sigma}$ and using independence of $\bar{\boldsymbol{u}}_t$ from $(\bar{\boldsymbol{x}}_t, \bar{\boldsymbol{m}}_t)$ yields a closed recursion on the diagonal entries. Equivalently, for each $i$, define the diagonal $2 \times 2$ covariance block

$$\boldsymbol{W}_{i,t} := \begin{bmatrix} (\boldsymbol{X}_t)_{ii} & (\boldsymbol{C}_t)_{ii} \\ (\boldsymbol{C}_t)_{ii} & (\boldsymbol{Y}_t)_{ii} \end{bmatrix} \in \mathbb{S}_+^2.$$

Let $\mathcal{X} := (\mathbb{S}^2)^d$ and $\mathcal{K} := (\mathbb{S}_+^2)^d$. Then there exists a linear operator $\mathcal{T}_{\mathrm{adam}} : \mathcal{X} \to \mathcal{X}$ such that

$$(\boldsymbol{W}_{1,t+1}, \ldots, \boldsymbol{W}_{d,t+1}) = \mathcal{T}_{\mathrm{adam}}(\boldsymbol{W}_{1,t}, \ldots, \boldsymbol{W}_{d,t}),$$

and $\mathcal{T}_{\mathrm{adam}}$ is exactly the operator in Theorem 5, i.e., for $\boldsymbol{W} = (\boldsymbol{W}_1, \ldots, \boldsymbol{W}_d) \in \mathcal{X}$,

$$(\mathcal{T}_{\mathrm{adam}}(\boldsymbol{W}))_i = \boldsymbol{A}_i\boldsymbol{W}_i\boldsymbol{A}_i^\top + \tilde{\eta}^2\left(\tilde{\lambda}_i^2(\boldsymbol{W}_i)_{11} + \sigma_i^{-1}\sum_{j=1}^d \sigma_j\tilde{\lambda}_j^2(\boldsymbol{W}_j)_{11}\right)\boldsymbol{Q},$$

where

$$\boldsymbol{A}_i := \begin{bmatrix} 1 - \tilde{\eta}\,\tilde{\lambda}_i & -\beta_1 \\ \tilde{\eta}\,\tilde{\lambda}_i & \beta_1 \end{bmatrix}, \qquad \boldsymbol{Q} := \begin{bmatrix} 1 & -1 \\ -1 & 1 \end{bmatrix}.$$

Moreover,

$$\mathbb{E}\|\bar{\boldsymbol{x}}_t\|^2 = \mathrm{Tr}(\boldsymbol{X}_t) = \sum_{i=1}^d (\boldsymbol{W}_{i,t})_{11},$$

so mean-square stability reduces to boundedness of $\sum_i(\boldsymbol{W}_{i,t})_{11}$.

In Theorem 5, we show that the analogous statement to Theorem 4 also holds for $\mathcal{T}_{\mathrm{adam}}$. Hence, the remainder of the proof follows by the same argument as Theorem 2. After reduction to strictly positive eigenvalues as in as Theorem 2 and applying Theorem 5, we obtain that: if

$$\eta\lambda_{\max}(\boldsymbol{P}^{-1}\boldsymbol{H}) < 1 + \beta_1 \quad \text{and} \quad \sum_{i=1}^d \frac{\eta\tilde{\lambda}_i}{2\left(1 - \frac{\eta\tilde{\lambda}_i}{1+\beta_1}\right)} < 1,$$

then the linearized dynamics of frozen ZO-Adam is mean-square linearly stable. Conversely, if the linearized dynamics is mean-square linearly stable, then

$$\eta\lambda_{\max}(\boldsymbol{P}^{-1}\boldsymbol{H}) < 1 + \beta_1 \quad \text{and} \quad \sum_{i=1}^d \frac{\eta\tilde{\lambda}_i}{2\left(1 - \frac{\eta\tilde{\lambda}_i}{1+\beta_1}\right)} \leqslant 1.$$

Hence, the mean-square critical step size $\eta_{\mathrm{ms}}^\star$ is the unique $\eta > 0$ satisfying

$$\eta\,\tilde{\lambda}_{\max} < 1 + \beta_1, \qquad \sum_{i=1}^d \frac{\eta\tilde{\lambda}_i}{2\left(1 - \frac{\eta\tilde{\lambda}_i}{1+\beta_1}\right)} = 1,$$

since the left-hand side is strictly increasing in $\eta$ on $\left(0, \frac{1+\beta_1}{\tilde{\lambda}_{\max}}\right)$.

**Bounds on $\eta_{\mathrm{ms}}^\star$.** At $\eta = \eta_{\mathrm{ms}}^\star$,

$$1 = \sum_{i=1}^d \frac{\eta_{\mathrm{ms}}^\star\tilde{\lambda}_i}{2\left(1 - \frac{\eta_{\mathrm{ms}}^\star\tilde{\lambda}_i}{1+\beta_1}\right)} \geqslant \sum_{i=1}^d \frac{\eta_{\mathrm{ms}}^\star\tilde{\lambda}_i}{2} = \frac{\eta_{\mathrm{ms}}^\star\,\mathrm{Tr}(\boldsymbol{P}^{-1}\boldsymbol{H})}{2},$$

which gives $\eta^\star_{\mathrm{ms}} \leqslant \frac{2}{\mathrm{Tr}(\boldsymbol{P}^{-1}\boldsymbol{H})}$. For the lower bound, use $\tilde{\lambda}_i \leqslant \tilde{\lambda}_{\max}$ to get

$$1 - \frac{\eta^\star_{\mathrm{ms}}\tilde{\lambda}_i}{1+\beta_1} \geqslant 1 - \frac{\eta^\star_{\mathrm{ms}}\tilde{\lambda}_{\max}}{1+\beta_1},$$

hence

$$1 = \sum_{i=1}^{d} \frac{\eta^\star_{\mathrm{ms}}\tilde{\lambda}_i}{2\left(1 - \frac{\eta^\star_{\mathrm{ms}}\tilde{\lambda}_i}{1+\beta_1}\right)} \leqslant \frac{\eta^\star_{\mathrm{ms}}\,\mathrm{Tr}(\boldsymbol{P}^{-1}\boldsymbol{H})}{2\left(1 - \frac{\eta^\star_{\mathrm{ms}}\tilde{\lambda}_{\max}}{1+\beta_1}\right)}.$$

Rearranging, we obtain

$$\eta^\star_{\mathrm{ms}} \geqslant \frac{2}{\mathrm{Tr}(\boldsymbol{P}^{-1}\boldsymbol{H}) + \frac{2\tilde{\lambda}_{\max}}{1+\beta_1}}.$$

$\square$

## A.5. Spectral analysis of Frozen ZO-Adam covariance operator

**Theorem 5** (Spectral characterization of the frozen ZO-Adam covariance operator)**.** *Assume $\boldsymbol{P} > 0$ and $\boldsymbol{H} \succeq 0$ are symmetric and commute: $\boldsymbol{P}\boldsymbol{H} = \boldsymbol{H}\boldsymbol{P}$. Let $\tilde{\lambda}_1 \geqslant \cdots \geqslant \tilde{\lambda}_d > 0$ be the eigenvalues of $\boldsymbol{P}^{-1}\boldsymbol{H}$. Fix $\eta > 0$ and $\beta_1 \in [0,1)$, and define $\tilde{\eta} := (1-\beta_1)\eta$. Let $\mathcal{X} := (\mathbb{S}^2)^d$ and $\mathcal{K} := (\mathbb{S}^2_+)^d$. In the simultaneous diagonalization basis $\boldsymbol{P} = \mathrm{diag}(\sigma_1,\ldots,\sigma_d)$ and $\tilde{\boldsymbol{H}} = \boldsymbol{P}^{-1/2}\boldsymbol{H}\boldsymbol{P}^{-1/2} = \mathrm{diag}(\tilde{\lambda}_1,\ldots,\tilde{\lambda}_d)$, define $\mathcal{T}_{\mathrm{adam}} : \mathcal{X} \to \mathcal{X}$ blockwise by*

$$(\mathcal{T}_{\mathrm{adam}}(\boldsymbol{W}))_i = \boldsymbol{A}_i \boldsymbol{W}_i \boldsymbol{A}_i^\top + \tilde{\eta}^2 \left(\tilde{\lambda}_i^2 (\boldsymbol{W}_i)_{11} + \sigma_i^{-1} \sum_{j=1}^{d} \sigma_j \tilde{\lambda}_j^2 (\boldsymbol{W}_j)_{11}\right) \boldsymbol{Q}, \qquad i = 1,\ldots,d, \tag{32}$$

$$\boldsymbol{A}_i := \begin{bmatrix} 1 - \tilde{\eta}\,\tilde{\lambda}_i & -\beta_1 \\ \tilde{\eta}\,\tilde{\lambda}_i & \beta_1 \end{bmatrix}, \qquad \boldsymbol{Q} := \begin{bmatrix} 1 & -1 \\ -1 & 1 \end{bmatrix}.$$

*Define*

$$S_{\mathrm{adam}}(\eta, \beta_1) := \sum_{i=1}^{d} \frac{\eta\,\tilde{\lambda}_i}{2\left(1 - \frac{\eta\,\tilde{\lambda}_i}{1+\beta_1}\right)}.$$

*Then $\mathcal{T}_{\mathrm{adam}}$ satisfies:*

*(a)* **(Leading eigenvalue in the cone)** *$\mathcal{T}_{\mathrm{adam}}(\mathcal{K}) \subseteq \mathcal{K}$. Moreover, $\mathcal{T}_{\mathrm{adam}}$ has an eigenvalue equal to its spectral radius $\rho(\mathcal{T}_{\mathrm{adam}})$, with an associated eigenvector $\boldsymbol{W}^\star \in \mathcal{K}\backslash\{\boldsymbol{0}\}$ satisfying*

$$\rho(\mathcal{T}_{\mathrm{adam}}) \geqslant \tilde{\eta}^2 \sum_{i=1}^{d} \tilde{\lambda}_i^2 > 0 \qquad and \qquad \sum_{i=1}^{d} (\boldsymbol{W}_i^\star)_{11} > 0.$$

*(b)* **(Critical case)**
$$\rho(\mathcal{T}_{\mathrm{adam}}) = 1 \quad \Longleftrightarrow \quad \eta\,\tilde{\lambda}_1 < 1 + \beta_1 \ \ and \ \ S_{\mathrm{adam}}(\eta, \beta_1) = 1.$$

*(c)* **(Subcritical case)**
$$\rho(\mathcal{T}_{\mathrm{adam}}) < 1 \quad \Longleftrightarrow \quad \eta\,\tilde{\lambda}_1 < 1 + \beta_1 \ \ and \ \ S_{\mathrm{adam}}(\eta, \beta_1) < 1.$$

*Proof.* We work on $\mathcal{X} = (\mathbb{S}^2)^d$ with cone $\mathcal{K} = (\mathbb{S}^2_+)^d$ and order $\preceq$.

**Proof of (a).** If $\boldsymbol{W} \in \mathcal{K}$, then $\boldsymbol{A}_i \boldsymbol{W}_i \boldsymbol{A}_i^\top \succeq 0$, $(\boldsymbol{W}_i)_{11} \geqslant 0$, and $\sigma_i^{-1} \sum_j \sigma_j \tilde{\lambda}_j^2 (\boldsymbol{W}_j)_{11} \geqslant 0$. Since $\boldsymbol{Q} \succeq 0$, (32) implies $(\mathcal{T}_{\mathrm{adam}}(\boldsymbol{W}))_i \succeq 0$ for all $i$, hence $\mathcal{T}_{\mathrm{adam}}(\mathcal{K}) \subseteq \mathcal{K}$.

Since $\mathcal{X}$ is finite dimensional and $\mathcal{K}$ is a closed, convex, pointed cone with nonempty interior, the Krein–Rutman theorem implies that $\mathcal{T}_{\mathrm{adam}}$ admits an eigenvalue equal to $\rho(\mathcal{T}_{\mathrm{adam}})$ with an eigenvector $\boldsymbol{W}^\star \in \mathcal{K} \backslash \{\boldsymbol{0}\}$.

To lower bound $\rho(\mathcal{T}_{\mathrm{adam}})$, define $\bar{\boldsymbol{Q}}_\Sigma := (\sigma_1^{-1} \boldsymbol{Q}, \ldots, \sigma_d^{-1} \boldsymbol{Q}) \in \mathcal{K} \backslash \{\boldsymbol{0}\}$. For each $i$,

$$
(\mathcal{T}_{\mathrm{adam}}(\bar{\boldsymbol{Q}}_\Sigma))_i = \boldsymbol{A}_i (\sigma_i^{-1} \boldsymbol{Q}) \boldsymbol{A}_i^\top + \tilde{\eta}^2 \left( \tilde{\lambda}_i^2 (\sigma_i^{-1} \boldsymbol{Q})_{11} + \sigma_i^{-1} \sum_{j=1}^d \sigma_j \tilde{\lambda}_j^2 (\sigma_j^{-1} \boldsymbol{Q})_{11} \right) \boldsymbol{Q}
$$

$$
\succeq \tilde{\eta}^2 \left( \sigma_i^{-1} \sum_{j=1}^d \tilde{\lambda}_j^2 \right) \boldsymbol{Q} = c\, (\bar{\boldsymbol{Q}}_\Sigma)_i,
$$

where $c := \tilde{\eta}^2 \sum_{j=1}^d \tilde{\lambda}_j^2$. Thus $\mathcal{T}_{\mathrm{adam}}(\bar{\boldsymbol{Q}}_\Sigma) \succeq c \bar{\boldsymbol{Q}}_\Sigma$, so $\mathcal{T}_{\mathrm{adam}}^t(\bar{\boldsymbol{Q}}_\Sigma) \succeq c^t \bar{\boldsymbol{Q}}_\Sigma$ for all $t \geqslant 1$. By Gelfand's formula, $\rho(\mathcal{T}_{\mathrm{adam}}) \geqslant c$.

Finally, if $(\boldsymbol{W}_i^\star)_{11} = 0$ for all $i$, then each $\boldsymbol{W}_i^\star = \left[ \begin{smallmatrix} 0 & 0 \\ 0 & y_i \end{smallmatrix} \right]$ with $y_i \geqslant 0$. In (32), the coupling term vanishes, and $(\mathcal{T}_{\mathrm{adam}}(\boldsymbol{W}^\star))_i = \boldsymbol{A}_i \boldsymbol{W}_i^\star \boldsymbol{A}_i^\top$ has $(1,1)$-entry equal to $\beta_1^2 y_i$. Since $\rho(\mathcal{T}_{\mathrm{adam}}) > 0$ and $\mathcal{T}_{\mathrm{adam}}(\boldsymbol{W}^\star) = \rho(\mathcal{T}_{\mathrm{adam}})\boldsymbol{W}^\star$, we get $\beta_1^2 y_i = 0$ for all $i$, hence $y_i = 0$ and $\boldsymbol{W}^\star = \boldsymbol{0}$, a contradiction. Therefore $\sum_i (\boldsymbol{W}_i^\star)_{11} > 0$.

**Local–global decomposition.** For each $i$, define $\mathcal{M}_i : \mathbb{S}^2 \to \mathbb{S}^2$ by

$$
\mathcal{M}_i(\boldsymbol{X}) := \boldsymbol{A}_i \boldsymbol{X} \boldsymbol{A}_i^\top + \tilde{\eta}^2 \tilde{\lambda}_i^2 (\boldsymbol{X})_{11} \boldsymbol{Q},
$$

and define the block-diagonal map $\mathcal{M} : \mathcal{X} \to \mathcal{X}$ by $(\mathcal{M}(\boldsymbol{W}))_i := \mathcal{M}_i(\boldsymbol{W}_i)$. Define the nonnegative linear functional $s_\Sigma : \mathcal{X} \to \mathbb{R}$ and the injection direction $\bar{\boldsymbol{Q}}_\Sigma \in \mathcal{K}$ by

$$
s_\Sigma(\boldsymbol{W}) := \sum_{j=1}^d \sigma_j \tilde{\lambda}_j^2 (\boldsymbol{W}_j)_{11}, \qquad \bar{\boldsymbol{Q}}_\Sigma := (\sigma_1^{-1} \boldsymbol{Q}, \ldots, \sigma_d^{-1} \boldsymbol{Q}).
$$

Then (32) is equivalently

$$
\mathcal{T}_{\mathrm{adam}}(\boldsymbol{W}) = \mathcal{M}(\boldsymbol{W}) + \tilde{\eta}^2 \, s_\Sigma(\boldsymbol{W}) \, \bar{\boldsymbol{Q}}_\Sigma. \tag{33}
$$

**Key local facts.** All statements below use the ZO-GDM local lemmas with the substitutions $(\eta, \beta, \lambda_i) \leftarrow (\tilde{\eta}, \beta_1, \tilde{\lambda}_i)$:

- Lemma 2(a): $\rho(\mathcal{M}_i) < 1$ if and only if $\tilde{\eta} \tilde{\lambda}_i < 1 - \beta_1^2$.

- Lemma 2(b),(c): Under $\tilde{\eta} \tilde{\lambda}_i < 1 - \beta_1^2$, $(\mathrm{Id} - \mathcal{M}_i)^{-1}$ exists, is $\mathbb{S}_+^2$-preserving, and $\boldsymbol{Y}_i := (\mathrm{Id} - \mathcal{M}_i)^{-1} \boldsymbol{Q} \succeq 0$ satisfies

$$
\gamma_i := (\boldsymbol{Y}_i)_{11} = \frac{1 + \beta_1}{2 \tilde{\eta} \tilde{\lambda}_i (1 - \beta_1^2 - \tilde{\eta} \tilde{\lambda}_i)}. \tag{34}
$$

- Lemma 3: For any $c > 0$, if $\rho(c\mathcal{M}_i) \geqslant 1$, then for any $\alpha > 0$ there is no $\boldsymbol{W} \succeq 0$ with $(\mathrm{Id} - c\mathcal{M}_i)(\boldsymbol{W}) \succeq \alpha \boldsymbol{Q}$.

**Proof of (b).** ($\Rightarrow$) Assume $\rho(\mathcal{T}_{\mathrm{adam}}) = 1$. Let $\boldsymbol{W}^\star \in \mathcal{K} \backslash \{\boldsymbol{0}\}$ satisfy $\mathcal{T}_{\mathrm{adam}}(\boldsymbol{W}^\star) = \boldsymbol{W}^\star$, and set $s^\star := s_\Sigma(\boldsymbol{W}^\star) > 0$. From (33), for each $i$,

$$
(\mathrm{Id} - \mathcal{M}_i)(\boldsymbol{W}_i^\star) = \tilde{\eta}^2 \frac{s^\star}{\sigma_i} \boldsymbol{Q}.
$$

Since $\tilde{\eta}^2 s^\star / \sigma_i > 0$, Lemma 3 forces $\rho(\mathcal{M}_i) < 1$ for all $i$ and hence $\tilde{\eta} \tilde{\lambda}_1 < 1 - \beta_1^2$, i.e. $\eta \tilde{\lambda}_1 < 1 + \beta_1$.

Moreover, $\boldsymbol{W}_i^\star = \tilde{\eta}^2 (s^\star / \sigma_i) \boldsymbol{Y}_i$, so $(\boldsymbol{W}_i^\star)_{11} = \tilde{\eta}^2 (s^\star / \sigma_i) \gamma_i$. Plugging into $s^\star = s_\Sigma(\boldsymbol{W}^\star)$ yields

$$
s^\star = \sum_{i=1}^d \sigma_i \tilde{\lambda}_i^2 (\boldsymbol{W}_i^\star)_{11} = \tilde{\eta}^2 s^\star \sum_{i=1}^d \tilde{\lambda}_i^2 \gamma_i,
$$

and cancelling $s^\star > 0$ gives

$$1 = \tilde{\eta}^2 \sum_{i=1}^d \tilde{\lambda}_i^2 \gamma_i. \tag{35}$$

Using (34),

$$\tilde{\eta}^2 \tilde{\lambda}_i^2 \gamma_i = \frac{\tilde{\eta}\tilde{\lambda}_i}{2(1-\beta_1)\left(1 - \frac{\tilde{\eta}\tilde{\lambda}_i}{1-\beta_1^2}\right)}.$$

Since $\tilde{\eta} = (1-\beta_1)\eta$ and $1 - \beta_1^2 = (1-\beta_1)(1+\beta_1)$, the right-hand side becomes $\frac{\eta\tilde{\lambda}_i}{2\left(1 - \frac{\eta\tilde{\lambda}_i}{1+\beta_1}\right)}$. Thus (35) is equivalent to $S_{\text{adam}}(\eta, \beta_1) = 1$.

($\Leftarrow$) Assume $\eta\tilde{\lambda}_1 < 1 + \beta_1$ and $S_{\text{adam}}(\eta, \beta_1) = 1$. Equivalently, $\tilde{\eta}\tilde{\lambda}_1 < 1 - \beta_1^2$ and $\sum_i \tilde{\eta}^2 \tilde{\lambda}_i^2 \gamma_i = 1$. Define $Y_i := (\text{Id} - \mathcal{M}_i)^{-1} Q \geq 0$ and set $\bar{Y} := (Y_1/\sigma_1, \ldots, Y_d/\sigma_d) \in \mathcal{K}\backslash\{0\}$. Then

$$(\mathcal{T}_{\text{adam}}(\bar{Y}))_i = \mathcal{M}_i(Y_i/\sigma_i) + \tilde{\eta}^2 s_\Sigma(\bar{Y})\sigma_i^{-1} Q.$$

But

$$s_\Sigma(\bar{Y}) = \sum_{i=1}^d \sigma_i \tilde{\lambda}_i^2 \sigma_i^{-1}(Y_i)_{11} = \sum_{i=1}^d \tilde{\lambda}_i^2 \gamma_i, \qquad \text{and} \qquad \tilde{\eta}^2 \sum_{i=1}^d \tilde{\lambda}_i^2 \gamma_i = 1,$$

so that $\tilde{\eta}^2 s_\Sigma(\bar{Y}) = 1$. Also, by definition, $\mathcal{M}_i(Y_i/\sigma_i) = \sigma_i^{-1}(Y_i - Q)$. Hence $(\mathcal{T}_{\text{adam}}(\bar{Y}))_i = (Y_i - Q)/\sigma_i + \sigma_i^{-1} Q = Y_i/\sigma_i = \bar{Y}_i$, i.e. $\mathcal{T}_{\text{adam}}(\bar{Y}) = \bar{Y}$. Therefore 1 is an eigenvalue, so $\rho(\mathcal{T}_{\text{adam}}) \geq 1$.

To show $\rho(\mathcal{T}_{\text{adam}}) \leq 1$, let $r := \rho(\mathcal{T}_{\text{adam}}) \geq 1$ and take $W^\star \in \mathcal{K}\backslash\{0\}$ with $\mathcal{T}_{\text{adam}}(W^\star) = rW^\star$ and $s^\star := s_\Sigma(W^\star) > 0$. From (33),

$$\left(\text{Id} - \tfrac{1}{r}\mathcal{M}\right)W^\star = \tfrac{\tilde{\eta}^2 s^\star}{r} \bar{Q}_\Sigma.$$

Since $\tilde{\eta}\tilde{\lambda}_1 < 1 - \beta_1^2$, we have $\rho(\mathcal{M}) < 1$, hence $\text{Id} - \tfrac{1}{r}\mathcal{M}$ is invertible and

$$W^\star = \tfrac{\tilde{\eta}^2 s^\star}{r}\left(\text{Id} - \tfrac{1}{r}\mathcal{M}\right)^{-1}\bar{Q}_\Sigma.$$

Applying $s_\Sigma(\cdot)$ and cancelling $s^\star > 0$ gives the fixed-point identity

$$r = \tilde{\eta}^2 s_\Sigma\left(\left(\text{Id} - \tfrac{1}{r}\mathcal{M}\right)^{-1}\bar{Q}_\Sigma\right). \tag{36}$$

Using the Neumann series and $r \geq 1$,

$$s_\Sigma\left(\left(\text{Id} - \tfrac{1}{r}\mathcal{M}\right)^{-1}\bar{Q}_\Sigma\right) = \sum_{k=0}^\infty r^{-k} s_\Sigma(\mathcal{M}^k(\bar{Q}_\Sigma)) \leq \sum_{k=0}^\infty s_\Sigma(\mathcal{M}^k(\bar{Q}_\Sigma)) = s_\Sigma\left((\text{Id} - \mathcal{M})^{-1}\bar{Q}_\Sigma\right).$$

Because $(\text{Id} - \mathcal{M})^{-1}$ is block-diagonal with blocks $(\text{Id} - \mathcal{M}_i)^{-1}$ and $(\bar{Q}_\Sigma)_i = \sigma_i^{-1}Q$, we have

$$\left((\text{Id} - \mathcal{M})^{-1}\bar{Q}_\Sigma\right)_i = \sigma_i^{-1}(\text{Id} - \mathcal{M}_i)^{-1}Q = \sigma_i^{-1}Y_i,$$

and therefore

$$s_\Sigma\left((\text{Id} - \mathcal{M})^{-1}\bar{Q}_\Sigma\right) = \sum_{i=1}^d \sigma_i \tilde{\lambda}_i^2(\sigma_i^{-1}Y_i)_{11} = \sum_{i=1}^d \tilde{\lambda}_i^2 \gamma_i.$$

Plugging into (36) yields

$$r \leq \tilde{\eta}^2 \sum_{i=1}^d \tilde{\lambda}_i^2 \gamma_i = 1,$$

so $r \leq 1$ and hence $\rho(\mathcal{T}_{\text{adam}}) = 1$.

**Proof of (c).** ($\Rightarrow$) Assume $\rho(\mathcal{T}_{\mathrm{adam}}) < 1$ and set $r := \rho(\mathcal{T}_{\mathrm{adam}})$. Let $\boldsymbol{W}^\star \in \mathcal{K}\backslash\{\boldsymbol{0}\}$ satisfy $\mathcal{T}_{\mathrm{adam}}(\boldsymbol{W}^\star) = r\boldsymbol{W}^\star$, and set $s^\star := s_\Sigma(\boldsymbol{W}^\star) > 0$. From (33), for each $i$,

$$\left(\mathrm{Id} - \tfrac{1}{r}\mathcal{M}_i\right)(\boldsymbol{W}_i^\star) = \tfrac{\tilde{\eta}^2 s^\star}{r\sigma_i}\boldsymbol{Q}.$$

Since $\tilde{\eta}^2 s^\star/(r\sigma_i) > 0$ and $\boldsymbol{W}_i^\star \geq 0$, Lemma 3 with $c = \tfrac{1}{r}$ implies $\rho(\tfrac{1}{r}\mathcal{M}_i) < 1$ for all $i$. Thus $\rho(\mathcal{M}_i) < r < 1$, and Lemma 2(a) gives $\tilde{\eta}\,\tilde{\lambda}_1 < 1 - \beta_1^2$. The bounds $\rho(\tfrac{1}{r}\mathcal{M}_i) < 1$ imply $\rho(\tfrac{1}{r}\mathcal{M}) < 1$, so $\mathrm{Id} - \tfrac{1}{r}\mathcal{M}$ is invertible by the Neumann series and

$$\boldsymbol{W}^\star = \tfrac{\tilde{\eta}^2 s^\star}{r}\left(\mathrm{Id} - \tfrac{1}{r}\mathcal{M}\right)^{-1}\bar{\boldsymbol{Q}}_\Sigma.$$

Applying $s_\Sigma(\cdot)$ and cancelling $s^\star > 0$ gives

$$r = \tilde{\eta}^2 \, s_\Sigma\left(\left(\mathrm{Id} - \tfrac{1}{r}\mathcal{M}\right)^{-1}\bar{\boldsymbol{Q}}_\Sigma\right).$$

Using the Neumann series and $r < 1$,

$$\tilde{\eta}^2 \, s_\Sigma\left(\left(\mathrm{Id} - \tfrac{1}{r}\mathcal{M}\right)^{-1}\bar{\boldsymbol{Q}}_\Sigma\right) \geqslant \tilde{\eta}^2 \, s_\Sigma\left((\mathrm{Id} - \mathcal{M})^{-1}\bar{\boldsymbol{Q}}_\Sigma\right) = \tilde{\eta}^2 \sum_{i=1}^d \tilde{\lambda}_i^2 \gamma_i = S_{\mathrm{adam}}(\eta, \beta_1).$$

Hence $S_{\mathrm{adam}}(\eta, \beta_1) \leqslant r < 1$.

($\Leftarrow$) Assume $\tilde{\eta}\,\tilde{\lambda}_1 < 1 - \beta_1^2$ and $S_{\mathrm{adam}}(\eta, \beta_1) < 1$. Suppose for contradiction that $r := \rho(\mathcal{T}_{\mathrm{adam}}) \geqslant 1$, and take $\boldsymbol{W}^\star \in \mathcal{K}\backslash\{\boldsymbol{0}\}$ with $\mathcal{T}_{\mathrm{adam}}(\boldsymbol{W}^\star) = r\boldsymbol{W}^\star$ and $s^\star := s_\Sigma(\boldsymbol{W}^\star) > 0$. Since $\tilde{\eta}\,\tilde{\lambda}_1 < 1 - \beta_1^2$, Lemma 2(a) gives $\rho(\mathcal{M}) < 1$, and therefore $\mathrm{Id} - \tfrac{1}{r}\mathcal{M}$ is invertible. As above,

$$r = \tilde{\eta}^2 \, s_\Sigma\left(\left(\mathrm{Id} - \tfrac{1}{r}\mathcal{M}\right)^{-1}\bar{\boldsymbol{Q}}_\Sigma\right).$$

Using the Neumann series and $r \geqslant 1$,

$$r \leqslant \tilde{\eta}^2 \, s_\Sigma\left((\mathrm{Id} - \mathcal{M})^{-1}\bar{\boldsymbol{Q}}_\Sigma\right) = \tilde{\eta}^2 \sum_{i=1}^d \tilde{\lambda}_i^2 \gamma_i = S_{\mathrm{adam}}(\eta, \beta_1) < 1,$$

which contradicts $r \geqslant 1$. Therefore $\rho(\mathcal{T}_{\mathrm{adam}}) < 1$. This completes (c) and the proof. $\qquad\square$

### A.6. Technical Lemmas

**Lemma 4.** *Let $\boldsymbol{u} \sim \mathcal{N}(\boldsymbol{0}, \boldsymbol{I})$. Then, the following identities hold by the Isserlis' Theorem.*

*(a)* $\mathbb{E}[\boldsymbol{u}\boldsymbol{u}^\top] = \boldsymbol{I}$.

*(b)* $\mathbb{E}[\boldsymbol{u}\boldsymbol{u}^\top \boldsymbol{A}\boldsymbol{u}\boldsymbol{u}^\top] = \boldsymbol{A} + \boldsymbol{A}^\top + \mathrm{Tr}(\boldsymbol{A})\boldsymbol{I}$ *for any matrix* $\boldsymbol{A}$.

*Proof.* Let $\boldsymbol{u} = (u_1, \ldots, u_d)^\top \sim \mathcal{N}(\boldsymbol{0}, \boldsymbol{I})$.

**(a)** Let $[d] := \{1, 2, \ldots, d\}$. For $i, j \in [d]$, $\mathbb{E}[u_i u_j] = \delta_{ij}$ since the coordinates are centered, independent, and $\mathrm{Var}(u_i) = 1$. Thus $\mathbb{E}[\boldsymbol{u}\boldsymbol{u}^\top] = \boldsymbol{I}$.

**(b)** Let $\boldsymbol{A} \in \mathbb{R}^{d\times d}$ be arbitrary. For $p, q \in [d]$, the $(p, q)$-entry of $\mathbb{E}[\boldsymbol{u}\boldsymbol{u}^\top \boldsymbol{A}\boldsymbol{u}\boldsymbol{u}^\top]$ equals

$$\mathbb{E}\left[u_p \left(\boldsymbol{u}^\top \boldsymbol{A}\boldsymbol{u}\right) u_q\right] = \sum_{i,j=1}^d A_{ij} \, \mathbb{E}[u_p u_i u_j u_q].$$

By the Isserlis' Theorem (Wick's formula) for centered Gaussians,

$$\mathbb{E}[u_p u_i u_j u_q] = \mathbb{E}[u_p u_i]\mathbb{E}[u_j u_q] + \mathbb{E}[u_p u_j]\mathbb{E}[u_i u_q] + \mathbb{E}[u_p u_q]\mathbb{E}[u_i u_j] = \delta_{pi}\delta_{jq} + \delta_{pj}\delta_{iq} + \delta_{pq}\delta_{ij}.$$

Substituting back gives

$$\sum_{i,j} A_{ij}\delta_{pi}\delta_{jq} = A_{pq}, \qquad \sum_{i,j} A_{ij}\delta_{pj}\delta_{iq} = A_{qp}, \qquad \sum_{i,j} A_{ij}\delta_{pq}\delta_{ij} = \delta_{pq}\operatorname{Tr}(\boldsymbol{A}).$$

Hence,

$$\left(\mathbb{E}[\boldsymbol{u}\boldsymbol{u}^\top \boldsymbol{A}\boldsymbol{u}\boldsymbol{u}^\top]\right)_{pq} = A_{pq} + A_{qp} + \operatorname{Tr}(\boldsymbol{A})\delta_{pq},$$

which is exactly $\boldsymbol{A} + \boldsymbol{A}^\top + \operatorname{Tr}(\boldsymbol{A})\boldsymbol{I}$. $\qquad\square$

**Lemma 5.** *Let* $\boldsymbol{\Sigma} > 0$ *be a fixed diagonal matrix and let* $\boldsymbol{u} \sim \mathcal{N}(\boldsymbol{0}, \boldsymbol{I})$. *Then, the following identities hold by the Isserlis' Theorem.*

*(a)* $\mathbb{E}[\boldsymbol{\Sigma}^{-1/2}\boldsymbol{u}\boldsymbol{u}^\top\boldsymbol{\Sigma}^{1/2}] = \boldsymbol{I}$.

*(b)* $\mathbb{E}[\boldsymbol{\Sigma}^{-1/2}\boldsymbol{u}\boldsymbol{u}^\top\boldsymbol{\Sigma}^{1/2}\boldsymbol{A}\boldsymbol{\Sigma}^{1/2}\boldsymbol{u}\boldsymbol{u}^\top\boldsymbol{\Sigma}^{-1/2}] = \boldsymbol{A} + \boldsymbol{A}^\top + \operatorname{Tr}(\boldsymbol{\Sigma}\boldsymbol{A})\boldsymbol{\Sigma}^{-1}$ *for any matrix* $\boldsymbol{A}$.

*Proof.* Let $\boldsymbol{u} \sim \mathcal{N}(\boldsymbol{0}, \boldsymbol{I})$ and let $\boldsymbol{\Sigma} > 0$ be diagonal. Write $\boldsymbol{D} := \boldsymbol{\Sigma}^{1/2}$, so $\boldsymbol{D}$ is invertible and $\boldsymbol{\Sigma}^{-1/2} = \boldsymbol{D}^{-1}$.

**(a)** We have

$$\mathbb{E}[\boldsymbol{\Sigma}^{-1/2}\boldsymbol{u}\boldsymbol{u}^\top\boldsymbol{\Sigma}^{1/2}] = \boldsymbol{D}^{-1}\,\mathbb{E}[\boldsymbol{u}\boldsymbol{u}^\top]\,\boldsymbol{D} = \boldsymbol{D}^{-1}\boldsymbol{I}\boldsymbol{D} = \boldsymbol{I},$$

where we used Lemma 4(a).

**(b)** Let $\boldsymbol{A}$ be arbitrary and define $\widetilde{\boldsymbol{A}} := \boldsymbol{D}\boldsymbol{A}\boldsymbol{D}$. Then

$$\boldsymbol{\Sigma}^{-1/2}\boldsymbol{u}\boldsymbol{u}^\top\boldsymbol{\Sigma}^{1/2}\boldsymbol{A}\boldsymbol{\Sigma}^{1/2}\boldsymbol{u}\boldsymbol{u}^\top\boldsymbol{\Sigma}^{-1/2} = \boldsymbol{D}^{-1}\,\boldsymbol{u}\boldsymbol{u}^\top\,\widetilde{\boldsymbol{A}}\,\boldsymbol{u}\boldsymbol{u}^\top\,\boldsymbol{D}^{-1}.$$

Taking expectation and applying Lemma 4(b) to $\widetilde{\boldsymbol{A}}$ yields

$$\mathbb{E}[\boldsymbol{u}\boldsymbol{u}^\top\,\widetilde{\boldsymbol{A}}\,\boldsymbol{u}\boldsymbol{u}^\top] = \widetilde{\boldsymbol{A}} + \widetilde{\boldsymbol{A}}^\top + \operatorname{Tr}(\widetilde{\boldsymbol{A}})\boldsymbol{I}.$$

Therefore,

$$\mathbb{E}[\boldsymbol{D}^{-1}\boldsymbol{u}\boldsymbol{u}^\top\,\widetilde{\boldsymbol{A}}\,\boldsymbol{u}\boldsymbol{u}^\top\boldsymbol{D}^{-1}] = \boldsymbol{D}^{-1}\widetilde{\boldsymbol{A}}\boldsymbol{D}^{-1} + \boldsymbol{D}^{-1}\widetilde{\boldsymbol{A}}^\top\boldsymbol{D}^{-1} + \operatorname{Tr}(\widetilde{\boldsymbol{A}})\,\boldsymbol{D}^{-1}\boldsymbol{I}\boldsymbol{D}^{-1}.$$

Using $\boldsymbol{D}^{-1}\widetilde{\boldsymbol{A}}\boldsymbol{D}^{-1} = \boldsymbol{A}$, $\boldsymbol{D}^{-1}\widetilde{\boldsymbol{A}}^\top\boldsymbol{D}^{-1} = \boldsymbol{A}^\top$, and

$$\operatorname{Tr}(\widetilde{\boldsymbol{A}}) = \operatorname{Tr}(\boldsymbol{D}\boldsymbol{A}\boldsymbol{D}) = \operatorname{Tr}(\boldsymbol{A}\boldsymbol{D}^2) = \operatorname{Tr}(\boldsymbol{A}\boldsymbol{\Sigma}), \qquad \boldsymbol{D}^{-1}\boldsymbol{I}\boldsymbol{D}^{-1} = \boldsymbol{D}^{-2} = \boldsymbol{\Sigma}^{-1},$$

we conclude that

$$\mathbb{E}[\boldsymbol{\Sigma}^{-1/2}\boldsymbol{u}\boldsymbol{u}^\top\boldsymbol{\Sigma}^{1/2}\boldsymbol{A}\boldsymbol{\Sigma}^{1/2}\boldsymbol{u}\boldsymbol{u}^\top\boldsymbol{\Sigma}^{-1/2}] = \boldsymbol{A} + \boldsymbol{A}^\top + \operatorname{Tr}(\boldsymbol{A}\boldsymbol{\Sigma})\,\boldsymbol{\Sigma}^{-1}.$$

$\qquad\square$

**Lemma 6** (The Krein–Rutman Theorem (Krein & Rutman, 1948); see also Theorem 19.2 in Deimling (1985))**.** *Let* $\mathcal{X}$ *be a finite-dimensional vector space and* $\mathcal{K} \subset \mathcal{X}$ *be a closed, convex, pointed cone with nonempty interior. Let* $\mathcal{T} : \mathcal{X} \to \mathcal{X}$ *be a linear opearator with* $\mathcal{T}(\mathcal{K}) \subseteq \mathcal{K}$. *Then, there exists* $\boldsymbol{x} \in \mathcal{K}\backslash\{\boldsymbol{0}\}$ *such that*

$$\mathcal{T}(\boldsymbol{x}) = \rho(\mathcal{T})\,\boldsymbol{x}\,.$$

*This is a standard corollary of the Krein–Rutman Theorem in finite dimension.*

# B. Effect of the gradient estimator

Our main analysis studies the Gaussian symmetric two-point estimator

$$\widehat{\nabla} f(\boldsymbol{x}; \boldsymbol{u}) := \frac{f(\boldsymbol{x} + \mu\boldsymbol{u}) - f(\boldsymbol{x} - \mu\boldsymbol{u})}{2\mu}\, \boldsymbol{u}, \qquad \boldsymbol{u} \sim \mathcal{N}(\boldsymbol{0}, \boldsymbol{I}),$$

which is widely used in practice, including in MeZO-style methods for large language model fine-tuning (Malladi et al., 2023). In this section, we explain how the same mean-square linear-stability framework changes under other estimator choices. For clarity, we focus on ZO-GD applied to the centered quadratic loss

$$f(\boldsymbol{x}) = \frac{1}{2}\boldsymbol{x}^\top \boldsymbol{H}\boldsymbol{x}, \qquad \boldsymbol{H} > 0.$$

For the symmetric two-point estimator, the quadratic update is

$$\boldsymbol{x}_{t+1} = (\boldsymbol{I} - \eta\boldsymbol{u}_t\boldsymbol{u}_t^\top \boldsymbol{H})\boldsymbol{x}_t.$$

Defining the second moment $\boldsymbol{\Sigma}_t := \mathbb{E}[\boldsymbol{x}_t\boldsymbol{x}_t^\top]$, we obtain the linear covariance recursion

$$\boldsymbol{\Sigma}_{t+1} = \mathcal{T}_{\boldsymbol{H}}(\boldsymbol{\Sigma}_t),$$

where

$$\mathcal{T}_{\boldsymbol{H}}(\boldsymbol{\Sigma}) := \mathbb{E}_{\boldsymbol{u}}\big[(\boldsymbol{I} - \eta\boldsymbol{u}\boldsymbol{u}^\top \boldsymbol{H})\boldsymbol{\Sigma}(\boldsymbol{I} - \eta\boldsymbol{H}\boldsymbol{u}\boldsymbol{u}^\top)\big]$$
$$= \boldsymbol{\Sigma} - \eta(\boldsymbol{H}\boldsymbol{\Sigma} + \boldsymbol{\Sigma}\boldsymbol{H}) + \eta^2(2\boldsymbol{H}\boldsymbol{\Sigma}\boldsymbol{H} + \mathrm{Tr}(\boldsymbol{H}\boldsymbol{\Sigma}\boldsymbol{H})\boldsymbol{I})\,.$$

Thus, mean-square linear stability is governed by the spectral radius of the covariance operator: stability holds when $\rho(\mathcal{T}_{\boldsymbol{H}}) < 1$, and the corresponding mean-square stability boundary is $\rho(\mathcal{T}_{\boldsymbol{H}}) = 1$.

## B.1. Forward finite-difference estimator

Consider the one-sided forward estimator

$$\widehat{\nabla} f(\boldsymbol{x}; \boldsymbol{u}) := \frac{f(\boldsymbol{x} + \mu\boldsymbol{u}) - f(\boldsymbol{x})}{\mu}\, \boldsymbol{u}, \qquad \boldsymbol{u} \sim \mathcal{N}(\boldsymbol{0}, \boldsymbol{I}).$$

Under the same quadratic loss,

$$\widehat{\nabla} f(\boldsymbol{x}; \boldsymbol{u}) = \boldsymbol{u}\boldsymbol{u}^\top \boldsymbol{H}\boldsymbol{x} + \frac{\mu}{2}(\boldsymbol{u}^\top \boldsymbol{H}\boldsymbol{u})\boldsymbol{u},$$

and therefore

$$\boldsymbol{x}_{t+1} = (\boldsymbol{I} - \eta\boldsymbol{u}_t\boldsymbol{u}_t^\top \boldsymbol{H})\boldsymbol{x}_t - \frac{\eta\mu}{2}(\boldsymbol{u}_t^\top \boldsymbol{H}\boldsymbol{u}_t)\boldsymbol{u}_t.$$

The second term is independent of $\boldsymbol{x}_t$ and has zero mean. Since the Gaussian direction distribution is symmetric, the cross terms in the second-moment recursion vanish, yielding

$$\boldsymbol{\Sigma}_{t+1} = \mathcal{T}_{\boldsymbol{H}}(\boldsymbol{\Sigma}_t) + \eta^2\mu^2\boldsymbol{Q}_{\boldsymbol{H}}, \qquad \boldsymbol{Q}_{\boldsymbol{H}} := \frac{1}{4}\mathbb{E}_{\boldsymbol{u}}\big[(\boldsymbol{u}^\top \boldsymbol{H}\boldsymbol{u})^2\boldsymbol{u}\boldsymbol{u}^\top\big].$$

Hence the homogeneous stability boundary is unchanged: it is still determined by $\rho(\mathcal{T}_{\boldsymbol{H}}) = 1$. What changes is the behavior inside the stable regime. When $\rho(\mathcal{T}_{\boldsymbol{H}}) < 1$, the covariance converges to a nonzero fixed point

$$\boldsymbol{\Sigma}_\infty = (\mathrm{Id} - \mathcal{T}_{\boldsymbol{H}})^{-1}(\eta^2\mu^2\boldsymbol{Q}_{\boldsymbol{H}}).$$

Thus the forward estimator has the same linear homogeneous stability threshold as the symmetric two-point estimator, but it introduces a finite-$\mu$ covariance floor of order $O(\mu^2)$. The symmetric two-point estimator avoids this effect on exact quadratics because it has no additive term at the minimizer.

## B.2. Non-Gaussian search directions

Next suppose the symmetric two-point estimator uses a search direction $\boldsymbol{u} \sim P$ with finite fourth moments and

$$\mathbb{E}[\boldsymbol{u}\boldsymbol{u}^\top] = \boldsymbol{I}.$$

The quadratic update remains

$$\boldsymbol{x}_{t+1} = (\boldsymbol{I} - \eta \boldsymbol{u}_t \boldsymbol{u}_t^\top \boldsymbol{H})\boldsymbol{x}_t,$$

but the covariance operator becomes

$$\boldsymbol{\Sigma}_{t+1} = \mathcal{T}_{\boldsymbol{H}}^P(\boldsymbol{\Sigma}_t),$$

where

$$\mathcal{T}_{\boldsymbol{H}}^P(\boldsymbol{\Sigma}) := \boldsymbol{\Sigma} - \eta(\boldsymbol{H}\boldsymbol{\Sigma} + \boldsymbol{\Sigma}\boldsymbol{H}) + \eta^2 \mathbb{E}_{\boldsymbol{u}\sim P}[\boldsymbol{u}\boldsymbol{u}^\top \boldsymbol{H}\boldsymbol{\Sigma}\boldsymbol{H}\boldsymbol{u}\boldsymbol{u}^\top].$$

Accordingly, the mean-square stability boundary is $\rho(\mathcal{T}_{\boldsymbol{H}}^P) = 1$. This makes explicit that changing the direction distribution changes the fourth-moment term in the covariance recursion, and hence can change the mean-square critical step size.

As a concrete example, let $P$ be the uniform distribution on the sphere of radius $\sqrt{d}$. Then $\mathbb{E}[\boldsymbol{u}\boldsymbol{u}^\top] = \boldsymbol{I}$, and the fourth-moment identity for the sphere gives

$$\mathcal{T}_{\boldsymbol{H}}^P(\boldsymbol{\Sigma}) = \boldsymbol{\Sigma} - \eta(\boldsymbol{H}\boldsymbol{\Sigma} + \boldsymbol{\Sigma}\boldsymbol{H}) + \eta^2 \frac{d}{d+2}\left(2\boldsymbol{H}\boldsymbol{\Sigma}\boldsymbol{H} + \mathrm{Tr}(\boldsymbol{H}\boldsymbol{\Sigma}\boldsymbol{H})\boldsymbol{I}\right).$$

Compared with the Gaussian case, the stochastic quadratic term is multiplied by $d/(d+2) < 1$, which weakens the mean-square noise amplification and enlarges the corresponding mean-square stable region.

## B.3. Multiple Gaussian queries per iteration

Finally, consider averaging $n$ independent Gaussian two-point estimates:

$$\widehat{\nabla} f_n(\boldsymbol{x}) := \frac{1}{n}\sum_{k=1}^n \frac{f(\boldsymbol{x} + \mu\boldsymbol{u}_k) - f(\boldsymbol{x} - \mu\boldsymbol{u}_k)}{2\mu}\,\boldsymbol{u}_k, \qquad \boldsymbol{u}_k \overset{\text{i.i.d.}}{\sim} \mathcal{N}(\boldsymbol{0}, \boldsymbol{I}).$$

Under the quadratic loss,

$$\widehat{\nabla} f_n(\boldsymbol{x}) = \boldsymbol{S}_n \boldsymbol{H}\boldsymbol{x}, \qquad \boldsymbol{S}_n := \frac{1}{n}\sum_{k=1}^n \boldsymbol{u}_k \boldsymbol{u}_k^\top,$$

so the update is $\boldsymbol{x}_{t+1} = (\boldsymbol{I} - \eta \boldsymbol{S}_n \boldsymbol{H})\boldsymbol{x}_t$ and

$$\boldsymbol{\Sigma}_{t+1} = \mathcal{T}_{\boldsymbol{H}}^{(n)}(\boldsymbol{\Sigma}_t), \qquad \mathcal{T}_{\boldsymbol{H}}^{(n)}(\boldsymbol{\Sigma}) := \mathbb{E}[(\boldsymbol{I} - \eta \boldsymbol{S}_n \boldsymbol{H})\boldsymbol{\Sigma}(\boldsymbol{I} - \eta \boldsymbol{H}\boldsymbol{S}_n)].$$

For Gaussian directions and any symmetric matrix $\boldsymbol{A}$,

$$\mathbb{E}[\boldsymbol{S}_n \boldsymbol{A}\boldsymbol{S}_n] = \left(1 + \frac{1}{n}\right)\boldsymbol{A} + \frac{1}{n}\mathrm{Tr}(\boldsymbol{A})\boldsymbol{I}.$$

Applying this identity with $\boldsymbol{A} = \boldsymbol{H}\boldsymbol{\Sigma}\boldsymbol{H}$ yields

$$\mathcal{T}_{\boldsymbol{H}}^{(n)}(\boldsymbol{\Sigma}) = \boldsymbol{\Sigma} - \eta(\boldsymbol{H}\boldsymbol{\Sigma} + \boldsymbol{\Sigma}\boldsymbol{H}) + \eta^2 \left[\left(1 + \frac{1}{n}\right)\boldsymbol{H}\boldsymbol{\Sigma}\boldsymbol{H} + \frac{1}{n}\mathrm{Tr}(\boldsymbol{H}\boldsymbol{\Sigma}\boldsymbol{H})\boldsymbol{I}\right].$$

Thus the mean-square stability boundary becomes $\rho(\mathcal{T}_{\boldsymbol{H}}^{(n)}) = 1$. Increasing $n$ weakens the stochastic coupling terms in the covariance recursion. In the limit $n \to \infty$,

$$\mathcal{T}_{\boldsymbol{H}}^{(n)}(\boldsymbol{\Sigma}) \to (\boldsymbol{I} - \eta\boldsymbol{H})\boldsymbol{\Sigma}(\boldsymbol{I} - \eta\boldsymbol{H}),$$

which is exactly the deterministic first-order GD covariance operator. Consequently, the mean-square critical step size approaches the first-order threshold $2/\lambda_{\max}(\boldsymbol{H})$ as $n \to \infty$.

# C. Experimental details

In this section, we provide additional experimental details. Our dataset construction, preprocessing, and model architectures follow the setup of Cohen et al. (2025).

**Dataset.** We train on a subset of CIFAR-10 consisting of 1,000 training examples drawn from the first four CIFAR-10 classes. We apply standard preprocessing by subtracting the dataset-wide channel-wise mean and dividing by the dataset-wide channel-wise standard deviation. For the squared-loss objective, we use one-hot targets: the ground-truth class is encoded as 1 and the remaining classes are encoded as 0.

**Architectures.** We evaluate three representative vision architectures:

- **CNN.** A four-layer convolutional network with initial channel width 32 and $3 \times 3$ convolutional kernels. We use GeLU activations, average pooling, and a linear readout layer.
- **ResNet.** A 20-layer ResNet (He et al., 2016) with GeLU activations and GroupNorm (Wu & He, 2018).
- **Vision Transformer (ViT).** A Vision Transformer (Dosovitskiy et al., 2021) with depth 3, embedding dimension 64, 8 attention heads, MLP dimension 256, and patch size 4.

**Top eigenvalue and trace estimation.** During training, we log curvature statistics every 1,000 iterations. Specifically, we compute the largest eigenvalue and trace of the Hessian (or the preconditioned Hessian $P_t^{-1} H_t$ for ZO-Adam) using matrix-free procedures, without explicitly forming the Hessian. We estimate the top eigenvalue via power iteration with 50 iterations, and estimate the trace via Hutchinson's method with 500 probe vectors. Both computations rely on Hessian–vector products.

**Compute.** All experiments are run on a single NVIDIA H100 GPU.

**Sequence sorting experiments.** For the sequence experiments in Appendix D.1, we use the synthetic sorting task described in Karpathy (2020) and follow the setup of Cohen et al. (2025, Appendix B.3). The network is fed a sequence of numbers and trained with a language-modeling loss to return the numbers in sorted order. We use numbers 1 through 4 and sequences of length 8, with 1,000 training examples for LSTM and 250 training examples for Mamba. We train under the same full-batch ZO protocol used in the vision experiments and track the same mean-square stability quantities for ZO-GD, ZO-GDM, and ZO-Adam.

# D. Additional experiments

## D.1. Sequence sorting tasks

To test whether mean-square EoS is specific to vision architectures, we run additional full-batch ZO experiments on the synthetic sorting task described in Karpathy (2020), using the setup adopted in Cohen et al. (2025, Appendix B.3). Figures 8 and 9 show the stability-band curves for LSTM and Mamba sequence models. Across ZO-GD, ZO-GDM, and ZO-Adam, the trace-based curvature terms again stabilize near the predicted mean-square stability thresholds, matching the qualitative behavior observed on the CIFAR-10 vision tasks.

## D.2. Effect of $\beta_1$ in ZO-Adam

Figure 10 reports full-batch ZO-Adam training of a CNN on CIFAR-10 while sweeping both $\beta_1$ and the step size $\eta$ with fixed $\beta_2 = 0.999$. Across $\beta_1 \in \{0.1, 0.5, 0.9\}$, we observe that the preconditioned trace $\mathrm{Tr}(P_t^{-1} H_t)$ stabilizes near the stability threshold $2/\eta$, suggesting that the effective stability boundary is primarily controlled by $\eta$ and is largely insensitive to $\beta_1$.

## D.3. Empirical verification of the commutativity approximation in ZO-Adam

We empirically assess the commutativity approximation $P_t H_t \approx H_t P_t$ used in Theorem 3.

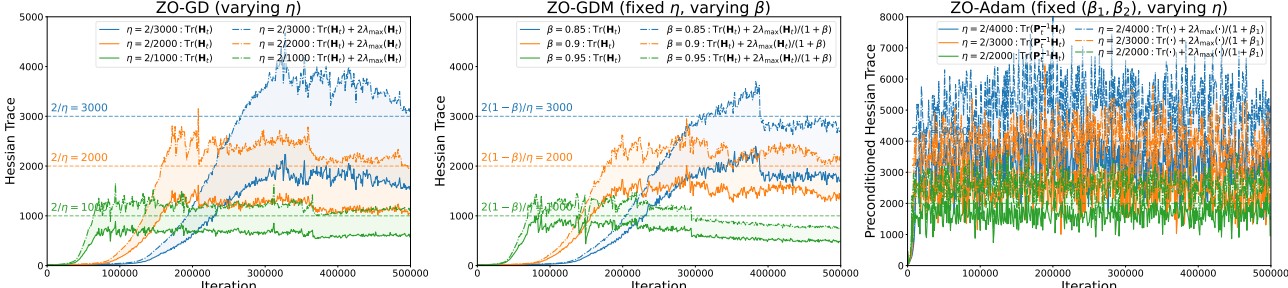

*Figure 8.* **Mean-square EoS on a synthetic sorting task with an LSTM.** On the synthetic sorting task described in Karpathy (2020), using the setup adopted by Cohen et al. (2025, Appendix B.3), we train full-batch ZO-GD, ZO-GDM, and ZO-Adam and track the corresponding mean-square stability quantities. The trace-based curvature terms stabilize near the predicted thresholds, mirroring the behavior observed in the vision experiments.

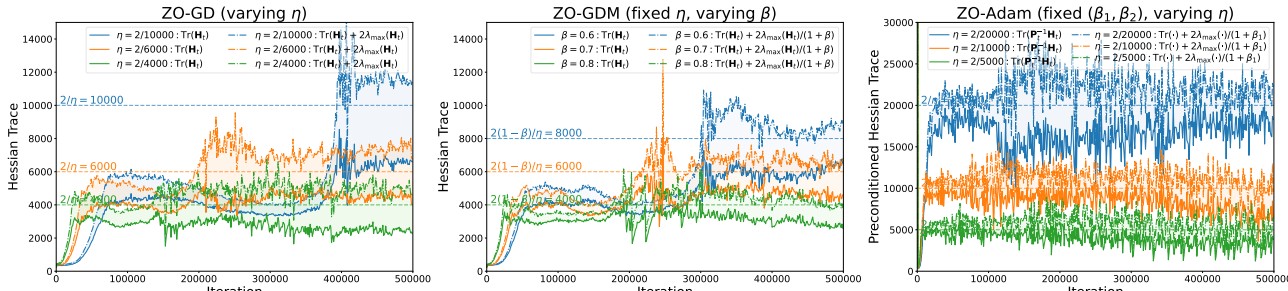

*Figure 9.* **Mean-square EoS on a synthetic sorting task with Mamba.** On the synthetic sorting task described in Karpathy (2020), using the setup adopted by Cohen et al. (2025, Appendix B.3), we train full-batch ZO-GD, ZO-GDM, and ZO-Adam and track the corresponding mean-square stability quantities. The trace-based curvature terms stabilize near the predicted thresholds, mirroring the behavior observed in the vision experiments.

**Metric and estimator.** We measure the *relative commutator Frobenius norm*

$$\mathrm{RelComm}_F(\boldsymbol{P}_t, \boldsymbol{H}_t) := \frac{\|[\boldsymbol{P}_t, \boldsymbol{H}_t]\|_F}{\|\boldsymbol{P}_t \boldsymbol{H}_t\|_F}, \qquad [\boldsymbol{P}_t, \boldsymbol{H}_t] = \boldsymbol{P}_t \boldsymbol{H}_t - \boldsymbol{H}_t \boldsymbol{P}_t.$$

Using the identity $\|\boldsymbol{A}\|_F^2 = \mathbb{E}\|\boldsymbol{A}\boldsymbol{z}\|_2^2$ for $\boldsymbol{z}$ with i.i.d. Rademacher entries, we estimate both $\|[\boldsymbol{P}_t, \boldsymbol{H}_t]\|_F$ and $\|\boldsymbol{P}_t \boldsymbol{H}_t\|_F$ via Hutchinson probes using only Hessian–vector products. Each probe requires two HVPs, namely $\boldsymbol{H}_t \boldsymbol{z}$ and $\boldsymbol{H}_t(\boldsymbol{P}_t \boldsymbol{z})$, together with a diagonal scaling by $\boldsymbol{P}_t$.

**Setup.** We use the same CNN and CIFAR-10 setup as in Figure 10 and log $\mathrm{RelComm}_F(\boldsymbol{P}_t, \boldsymbol{H}_t)$ at the same frequency as the curvature statistics. We use 50 probes per checkpoint.

**Results.** Across all tested $(\beta_1, \eta)$, $\mathrm{RelComm}_F(\boldsymbol{P}_t, \boldsymbol{H}_t)$ decreases rapidly from the value 0.8–0.9 at initialization to below 0.05 and remains below 0.05 throughout training (bottom row of Figure 10), supporting $\boldsymbol{P}_t \boldsymbol{H}_t \approx \boldsymbol{H}_t \boldsymbol{P}_t$ as a reasonable approximation in this setting.

**Why do $\boldsymbol{P}_t$ and $\boldsymbol{H}_t$ approximately commute in deep learning?** For ZO-Adam, $\boldsymbol{P}_t$ is diagonal in the parameter basis. Writing entries explicitly,

$$([\boldsymbol{P}_t, \boldsymbol{H}_t])_{ij} = (p_{t,i} - p_{t,j})(\boldsymbol{H}_t)_{ij},$$

so non-commutativity is driven by off-diagonal Hessian couplings between coordinates that receive different preconditioning. Consequently, $\boldsymbol{P}_t$ and $\boldsymbol{H}_t$ are expected to approximately commute when $(i)$ $\boldsymbol{H}_t$ is close to block-diagonal under a natural parameter partition (e.g., by layer blocks), so cross-block couplings are small, and $(ii)$ the preconditioner varies primarily across such blocks (or is slowly varying within blocks). This heuristic is consistent with empirical observations that neural network Hessians exhibit a near-block-diagonal structure, and with blockwise second-moment approximations studied in recent work (e.g., Zhang et al. (2025b)).

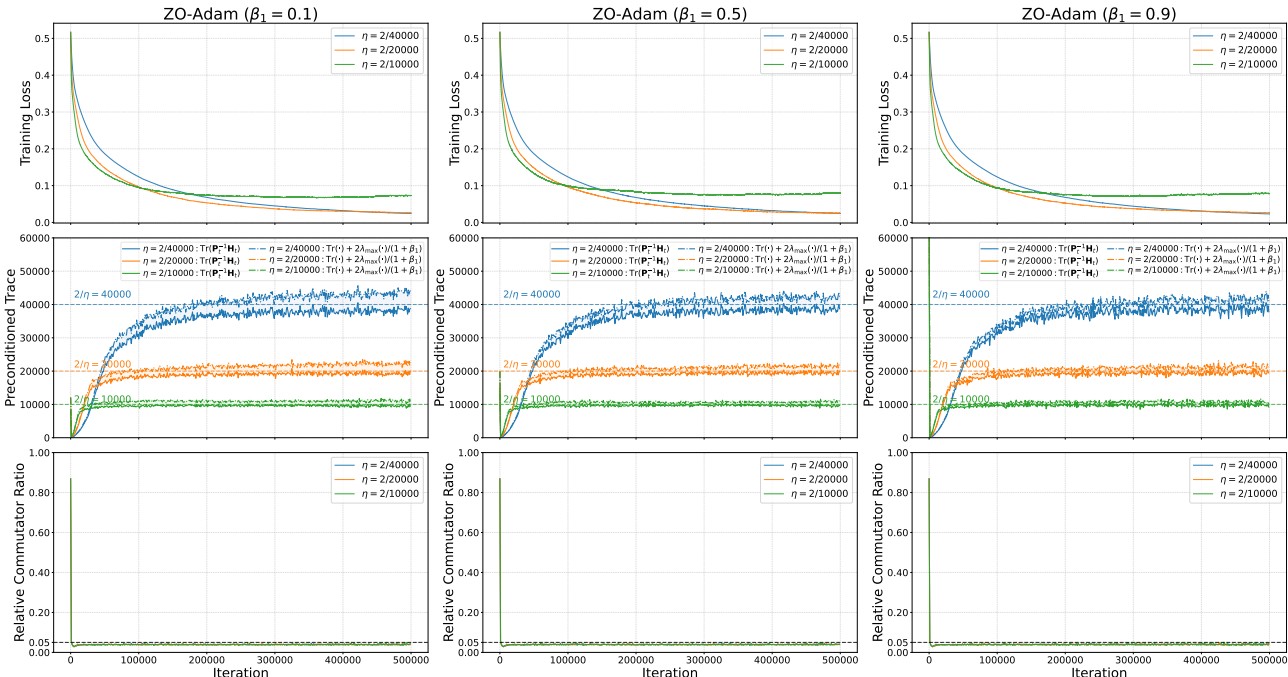

*Figure 10.* **ZO-Adam sweep over $\beta_1$ and $\eta$.** Full-batch ZO-Adam on a CNN trained on CIFAR-10 with $\beta_1 \in \{0.1, 0.5, 0.9\}$ (left to right) and multiple step sizes $\eta$ per setting. *Top:* training loss. *Middle:* preconditioned curvature statistics $\mathrm{Tr}(\boldsymbol{P}_t^{-1}\boldsymbol{H}_t)$ and $\mathrm{Tr}(\boldsymbol{P}_t^{-1}\boldsymbol{H}_t) + \frac{2}{1+\beta_1}\lambda_{\max}(\boldsymbol{P}_t^{-1}\boldsymbol{H}_t)$. *Bottom:* relative commutator ratio $\|[\boldsymbol{P}_t, \boldsymbol{H}_t]\|_F / \|\boldsymbol{P}_t \boldsymbol{H}_t\|_F$ (cf. Appendix D.3).

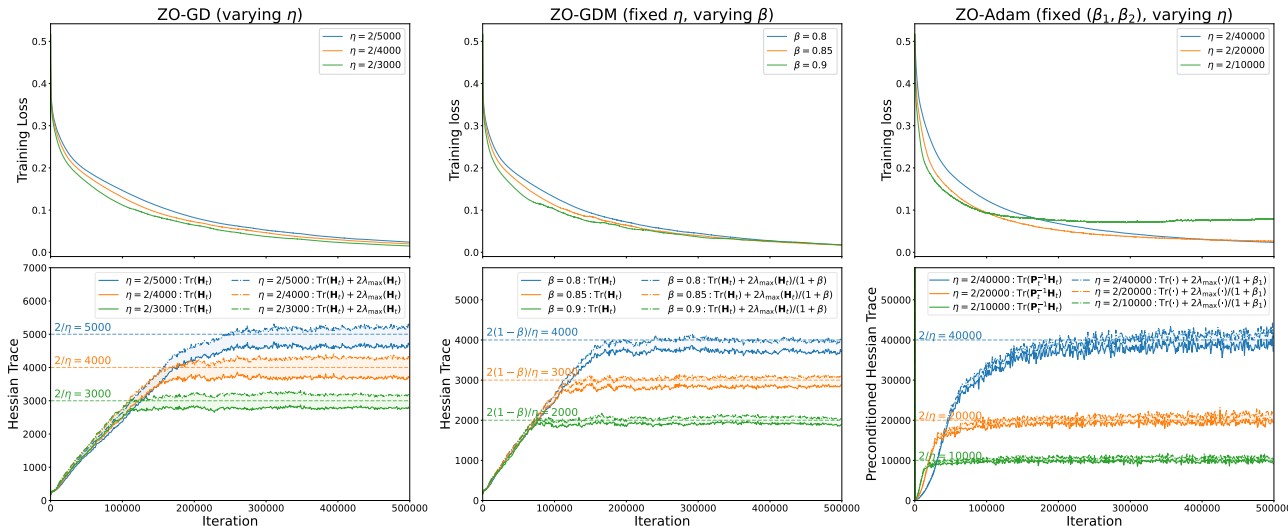

*Figure 11.* **CNN: same experiments as Figure 2, with training loss.** *Top:* training loss. *Bottom:* the stability-band plots from Figure 2 for ZO-GD (left), ZO-GDM (middle), and ZO-Adam (right).

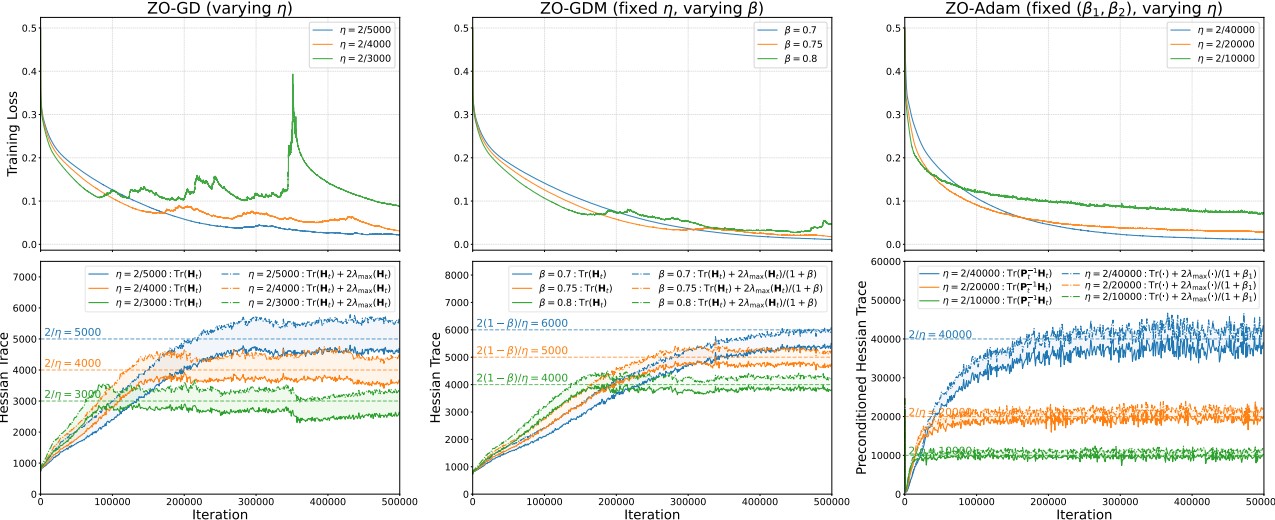

*Figure 12.* **ResNet: same experiments as Figure 6, with training loss.** *Top:* training loss. *Bottom:* the stability-band plots from Figure 6 for ZO-GD (left), ZO-GDM (middle), and ZO-Adam (right).

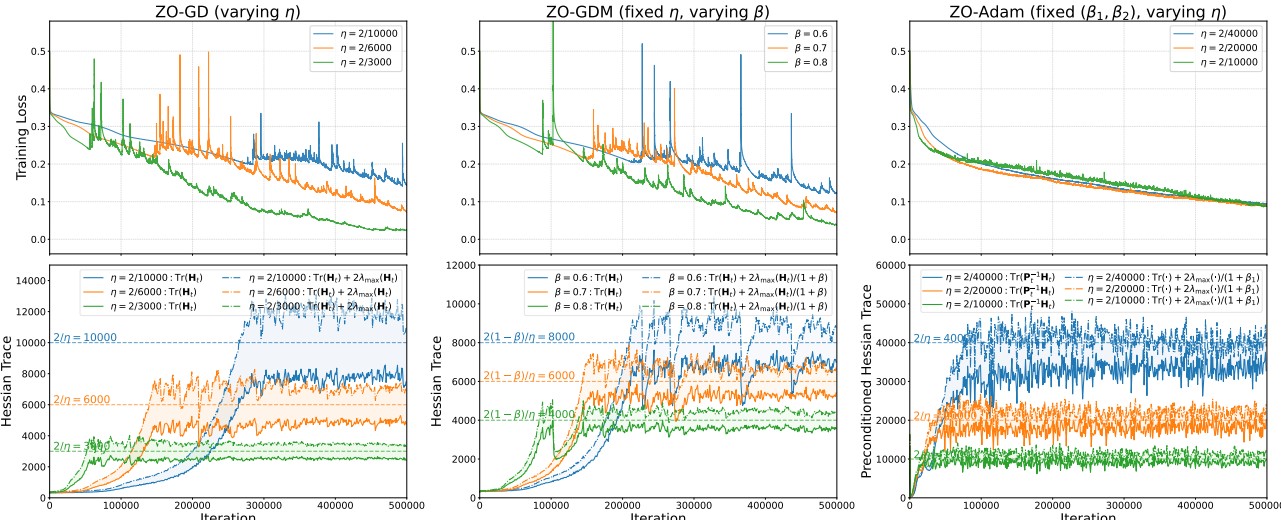

*Figure 13.* **Vision Transformer: same experiments as Figure 7, with training loss.** *Top:* training loss. *Bottom:* the stability-band plots from Figure 7 for ZO-GD (left), ZO-GDM (middle), and ZO-Adam (right).

