# OpenReview forum: "Zeroth-Order Optimization at the Edge of Stability"
_ICML.cc/2026/Conference — ICML 2026 regular_

### Official Review · Reviewer_fRCn · 2026-02-22

**Soundness:** 2
**Presentation:** 3
**Significance:** 2
**Originality:** 3
**Overall Recommendation:** 4
**Confidence:** 3

**Summary:**

The paper investigates the edge of stability (EoS) phenomenon for zeroth-order (ZO) optimization methods. The authors develop a mean-square linear stability theory for ZO methods that are based on the standard two-point gradient estimator. The key finding is that, unlike FO methods where stability depends solely on the largest Hessian eigenvalue, mean-square stability of ZO methods depends on the entire Hessian spectrum dominated by the Hessian trace. The authors derive exact critical step size characterizations and tractable bounds with the trace and top eigenvalue. Empirically, they demonstrate that full-batch ZO methods consistently operate near the predicted mean-square stability boundary across CNNs, ResNets, and Vision Transformers.

**Compliance With Llm Reviewing Policy:**

Affirmed.

**Key Questions For Authors:**

Does the EoS phenomenon persist under cross-entropy loss? Under mini-batch training at moderate batch sizes?

How tight are the upper/lower bounds in practice? Could you report the exact critical step size (computed from the full spectrum) for the smaller CNN model to quantify the gap?

Can you provide intuition for why momentum shrinks the ZO stable region (opposite to FO)?

**Limitations:**

As discussed above, the main limitations of the paper come from the lack of scalable experiments, and comprehensive comparisons with similar methods. While this is a theory dominated paper, including these will be helpful.

**Strengths And Weaknesses:**

Strengths:

The paper asks a natural yet previously unexplored question: does the EoS phenomenon extend to ZO methods, and if so, what governs it? The observation that the Hessian trace (rather than the top eigenvalue) is the controlling quantity is surprising and insightful. Theorems 1–3 provide exact mean-square stability characterizations for ZO-GD, ZO-GDM, and Frozen ZO-Adam. Table 1 provides a summary comparing FO and ZO stability conditions.

The paper surfaces several qualitatively interesting differences, including ZO stability depends on the full spectrum vs. FO depending only on lambda_max; Momentum has opposite effects, where increasing beta enlarges the stable region for FO-GDM but shrinks it for ZO-GDM. The experiments across three architectures (CNN, ResNet, ViT) and three optimizers (ZO-GD, ZO-GDM, ZO-Adam) consistently show the stability threshold staying within or near the predicted interval. The paper is generally well-written, the use of stability intervals rather than exact thresholds for empirical tracking is a well-justified design choice.

Weaknesses

All experiments use a small 1,000-sample subset of CIFAR-10 with full-batch training and squared loss. While this is consistent with prior EoS literature, it raises questions about whether the findings extend to realistic-scale ZO training. Meanthilw, the experiments used the squared loss, which is non-standard for classification, and cross-entropy loss is far more common. Does the phenomenon persist under cross-entropy? The paper also does not report test accuracy or generalization metrics. There is mo comparison with other ZO estimators beyond the standard two-point estimator (e.g., coordinate-wise estimators, multi-point estimators). How specific are the results to the Gaussian two-point estimator? Also, some figures (especially Figures 6, 7) are dense and hard to parse at the resolution provided.

---

> ### Author Rebuttal · Authors · 2026-03-31
>
> Thank you for the positive review! We respond to the remaining concerns below.
>
> **1. Does the EoS phenomenon persist under cross-entropy loss?**
>
> We used MSE loss because it is the standard choice in the EoS literature and usually gives the clearest picture. In the original EoS paper, full-batch GD also shows EoS under cross-entropy, but the behavior is less clean because the sharpness drops in the terminal phase of training (see this [figure](https://shorturl.at/Rxy21)). The explanation there is that, under cross-entropy, the second derivative becomes small once the prediction margins grow large, so the sharpness naturally decreases late in training.
>
> We observe the same qualitative effect for ZO-GD in our additional experiment with cross-entropy ([cross-entropy result](https://shorturl.at/VtKv7)): the mean-square EoS behavior is visible earlier in training, but the trace decreases in the terminal phase for the same geometric reason. We will add this result and discussion in the revision.
>
> **2. Does ZO EoS persist under mini-batch training at moderate batch sizes?**
>
> This is an important question. Our theory focuses on the full-batch setting, where the only source of randomness is the ZO estimator itself. We see this as a natural first step toward understanding ZO EoS. In the mini-batch setting, the dynamics must account for both estimator-direction noise and data-subsampling noise, which makes the second-moment analysis substantially more involved.
>
> That said, the current manuscript already includes a preliminary mini-batch ZO-SGD experiment in Figure 5. In that experiment, mini-batch training converges to a noticeably flatter regime than full-batch ZO-GD. This suggests that the full-batch theory alone is not enough to explain mini-batch ZO EoS, and that a separate analysis is needed for that setting.
>
> **3. How tight are the upper/lower bounds in practice? Could you report the exact critical step size from the full spectrum for the smaller CNN?**
>
> Even our smallest neural-network experiment, the 3-layer CNN, has 30,570 parameters. While this is small by deep-learning standards, computing the exact Hessian spectrum throughout training is still not practical in this setting. This is why we focus on trace- and top-eigenvalue-based upper and lower bounds, which are much more tractable to estimate during training.
>
> Empirically, the tightness of the bounds is already reflected in our figures: we plot both the lower and upper bounds, so the gap between the two curves directly shows the width of the bound. In particular, the bounds become tight when the Hessian trace is much larger than the top eigenvalue.
>
> **4. Why does momentum shrink the ZO stable region, opposite to FO?**
>
> A simple way to think about the difference is as follows. In FO-GDM, momentum damps deterministic oscillations near sharp directions, which allows a larger stable step size. In ZO-GDM, by contrast, momentum accumulates not only the gradient signal but also the random-direction estimator noise. Since ZO stability is governed by second moments, that extra accumulated noise makes the dynamics less stable, so increasing $\beta$ shrinks the stable region. We will add this intuition in the revision.
>
> **5. How specific are the results to the Gaussian symmetric two-point estimator?**
>
> This is a very interesting question! We focus on the symmetric two-point estimator
> $$
> \hat{\nabla}f(x;z):=\frac{f(x+\mu z)-f(x-\mu z)}{2\mu}z,\qquad z\sim N(0,I),
> $$
> mainly because it is widely used in practice (e.g., MeZO-style methods for LLM fine-tuning).
>
> That said, our framework extends naturally beyond this choice.
>
> For the forward estimator $\frac{f(x+\mu z)-f(x)}{\mu}z$, the mean-square EoS threshold does not change. What changes is the stable-regime behavior: the covariance recursion becomes nonhomogeneous, which introduces a nonzero stationary covariance floor of order $O(\mu^2)$.
>
> More generally, if the Gaussian direction is replaced by another isotropic distribution $P$ with $\mathbb E_{z\sim P}[zz^\top]=I$, then the same second-moment framework applies, but the operator changes through the fourth moments of $P$. As a result, the critical step size generally changes as well. For example, for the uniform distribution on the sphere of radius $\sqrt d$, the mean-square critical step size becomes larger than in the Gaussian case.
>
> For a more detailed derivation and discussion, please also see [our response to Reviewer thVy](https://openreview.net/forum?id=s87tQaKAER&noteId=jqOwft4Y0o).
>
> **6. Additional experiments**
>
> We have also run additional experiments on sequence architectures on a synthetic sorting task. We observe the same qualitative mean-square EoS behavior there as well ([LSTM result](https://shorturl.at/sGMZr), [Mamba result](https://shorturl.at/6HX6r)). We will add these experiments in the revision.
>
> ---
>
> Thank you again for your review. Does the above address your concerns? Please let us know if you have any further questions!

---

> > ### Author Rebuttal · Reviewer_fRCn · 2026-04-01
> >
> > I thank the author for addressing my questions, and will keep my score at weak accept.

---

> > > ### Author Response · Authors · 2026-04-02
> > >
> > > We are delighted to hear that your concerns have been fully addressed! We thank the reviewer again for the thoughtful comments and valuable insights.

---

### Official Review · Reviewer_thVy · 2026-02-24

**Soundness:** 3
**Presentation:** 4
**Significance:** 3
**Originality:** 3
**Overall Recommendation:** 5
**Confidence:** 4

**Summary:**

This paper investigate the Edge of Stability (EoS) phenomenom when using full-batch zeroth-order estimator of the gradient (zeroth order gradient descent, polyak's momentum and frozen Adam), precisely using a two point estimator with random gaussian direction. The authors observe empirically such an EoS phenomenon when training several deep learning models using zeroth-order algorithms, but it is significantly different compared with the case where training occurs with gradient-based algorithms: stability is not merely governed by sharpness, but here the whole Hessian spectrum matters. In the line of existing studies investigating the EoS phenomenom, a (mean square) linear stability analysis is performed to obtain the precise relation between algorithms parameters and Hessian spectrum that makes the algorithm diverge when minimizing a (convex) quadratic function, providing a stability criterion. This exact criterion depends on the whole spectrum, so the authors lower and upper bound it with quantity relying only on the trace of the Hessian matrix and the sharpness, to be able to track these quantities in experiments. On the experiments, the training dynamics indeed appear to happen near this stability bound, describing a EoS phenomenom as for gradient-based algorithms, but specific to zeroth order algorithms.

**Compliance With Llm Reviewing Policy:**

Affirmed.

**Final Justification:**

My opinion about this paper is still very positive, as it was in my initial review (5-accept).
I find the paper very well written and the findings are interesting.
I would also like to highlight the commitment shown by the authors during the rebuttal period.
I recommend acceptance of this paper.

**Key Questions For Authors:**

1. I would have appreciated a discussion about your choice of two-point estimator. In particular, another very common choice is the form $\frac{(f(x+\mu u) - f(x)}{\mu}u$. I guess your choice is, at least partly, motivated by the fact that the stability analysis would be more complicated with this estimator, as $\hat \nabla f_{quad}$ would involve a supplementary term $\frac{1}{2}\mu u^T Hu$, introducing a dependence on $\mu$. Do you have any insight about how your analysis and findings are dependent on the specific form of your estimator ?

2. You explain that the full spectrum is intractable in large scale experiment, justifying to bound the theoretical critical $\eta$ value using only the Trace and $\lambda_{max}$, which is relevant. However, I would have appreciated at least one experiment at smaller scale where this is tractable, in order to test the relevance of the exact instability criterion. Did you try such experiment, or do you have any insight ?

3.  The argument between lines 866 and 869 seems incorrect to me, as your lemma 2.b applies to $(I_d - \mathcal{M}_i)^{-1}$ but you apply it to $(I_d - \frac{1}{r}\mathcal{M}_i)^{-1}$. Could you please clarify ?

4.  You highlight that the momentum parameter affect the stability bound in the opposite way compared to gradient-based optimization. I would guess that it is because momentum is more sensitive to error we make on the gradient. Do you have another interpretation or additional insight ?

5. Did you consider other choices of distribution than gaussian distribution, e.g. uniformly distributed on the euclidean sphere (which leads to smaller variance of the estimator [1]) ?  To my understanding, the choice of distribution intervenes only for Lemma 4. Point (a) is rather easily verified by several kind of random direction. (b) would be more difficult. I note that [1] highlighted the important role of the forth moment of the direction vector, which also somewhat intervenes in your Lemma 4 (b).  Do you have any insight ?

[1] Revisiting zeroth-order optimization: Minimum-variance two-point estimators and directionally aligned perturbations, Shaocong Ma and Heng Huang

**Limitations:**

yes

**Strengths And Weaknesses:**

**Strenghts**

Overall, I think this is a very good paper. Specifically:

 *(i) Significance and Originality.* To my knowledge it is the first paper to highlight the specific EoS phenomenon occurring when using (full batch) zeroth-order algorithms, which is an important finding. The fact that it is fundamentally different compared with gradient-based optimization, with the special role of the trace of the hessian matrix, is particularly interesting. The fact that the paper also studies a with momentum version, and an adaptive version, is noticeable.

 *(ii) Presentation.* The paper is very well written, and easy to follow.  The proofs are technical, but the paper effectively helps the reader understand them, by describing the key steps of the proofs before diving in, and when technical lemmas are used, by giving some intuition about these lemmas. Overall, the appendices are well polished with very few typos (see minor remarks below).

*(iii) Soundness.* The claim is solid to me. Specifically is the relevance of the instability criterion; well supported by experiments, and theoretically by the mean-squared stability analysis.  Also is the fact that we are facing an EoS phenomenom, e.g. with catapult dynamics, "progressive-sharpening" like behavior  (even if it does not reduce to sharpness increasing). I checked most of the technical detail of the proof of Theorems 1 and 2 (not the proof of Lemmas 2 and 3), everything I read seems correct (just one probably mild exception, see Question 3.).

I believe this work opens the door to interesting further studies.

**Weaknesses**

Please see the "Question" section, points 1., 2. and 3., for my most important remarks.

*Minor remarks on presentation*

 - Lines 198-202: "Exponential stability... as a principled tool", could you provide a reference ?
 - Lines 202-203: "Let $x^\ast$ be a twice differentiable minimizer", I guess you mean that the function is twice differentiable at this point ? If yes, I think it is not a standard formulation and is slightly ambiguous.
 - In the statement of Theorem 4, I read that all the eigenvalues of the hessian matrix should be strictly above zero. According to the proof and the setting, I think only $\lambda_{max}$ has to be strictly above zero.
 - Line 730 "should the same", the word "have" is missing
- Lines 1335-1377 (Proof of  point (i) in proof of theorem 2), the wording is slightly confusing because you start stating "we prove the contrapositive", and you end arguing that you found a "contradiction", which induce a proof by contradiction.

---

> ### Author Rebuttal · Authors · 2026-03-31
>
> Thank you for the positive review! We respond to the remaining concerns below.
>
> **1. Dependence on the choice of gradient estimator.**
>
> This is a very interesting question! We focus on the symmetric two-point estimator
> $$
> \hat{\nabla}f(x;z):=\frac{f(x+\mu z)-f(x-\mu z)}{2\mu}z,\qquad z\sim N(0,I),
> $$
> mainly because it is widely used in practice (e.g., MeZO-style methods for LLM fine-tuning).
>
> Under the quadratic loss $f(x):=\frac12 x^\top Hx$, ZO-GD updates as
> $$x_{t+1}=(I-\eta zz^\top H)x_t.$$
> Accordingly, the second moment $\Sigma_t:=\mathbb E[x_t x_t^\top]$ evolves as $\Sigma_{t+1}=\mathcal T_H(\Sigma_t)$,
> where
> $$\mathcal T_H(\Sigma):=\mathbb E_z[(I-\eta zz^\top H)\Sigma (I-\eta Hzz^\top)]=\Sigma-\eta(H\Sigma+\Sigma H)+\eta^2(2H\Sigma H+\mathrm{Tr}(H\Sigma H)I).$$
> The mean-square EoS condition is $\rho(\mathcal T_H)=1$.
>
> **(1) Forward estimator:**
>
> Now consider the forward estimator $\frac{f(x+\mu z)-f(x)}{\mu}z$.
> Under the same loss,
> $$x_{t+1}=(I-\eta zz^\top H)x_t-\frac{\eta\mu}{2}(z^\top Hz)z.$$
> Thus, the second-moment recursion becomes nonhomogeneous:
> $$\Sigma_{t+1}=\mathcal T_H(\Sigma_t)+\eta^2\mu^2 Q_H,$$
> where
> $$Q_H:=\frac14\,\mathbb E_z[(z^\top Hz)^2zz^\top].$$
> Therefore, the mean-square EoS condition remains unchanged: $\rho(\mathcal T_H)=1$.
> What changes is the limiting covariance in the stable regime. When $\rho(\mathcal T_H)<1$,
> - for the symmetric two-point estimator, $\Sigma_t\to 0$;
> - for the forward estimator, $\Sigma_t\to \Sigma_\infty:=(I-\mathcal T_H)^{-1}(\eta^2\mu^2 Q_H)$.
>
> **(2) Random direction distribution:**
>
> Next, suppose $z\sim P$ for some distribution $P$ with $\mathbb E_{z\sim P}[zz^\top]=I$. Then the second-moment recursion becomes $\Sigma_{t+1}=\mathcal T_H^P(\Sigma_t)$, where
> $$\mathcal T_H^P(\Sigma):=\Sigma-\eta(H\Sigma+\Sigma H)+\eta^2\mathbb E[zz^\top H\Sigma H zz^\top].$$
> Accordingly, the EoS condition becomes $\rho(\mathcal T_H^P)=1$.
>
> For example, if $P$ is the uniform distribution on the sphere of radius $\sqrt d$, then
> $$\mathcal T_H^{P}(\Sigma)=\Sigma-\eta(H\Sigma+\Sigma H)
> +\eta^2\frac{d}{d+2}(2H\Sigma H+\mathrm{Tr}(H\Sigma H)I).$$
> Compared with the Gaussian case, the stochastic quadratic term is multiplied by $d/(d+2)<1$, so the mean-square critical step size becomes larger.
>
> **2. Exact mean-square critical step size on smaller-scale models**
>
> Thank you for this suggestion. Even in our smallest neural-network experiment, the 3-layer CNN has 30,570 parameters. While this is still small by deep-learning standards, computing the exact Hessian spectrum throughout training is not practical in this setting. It would certainly be interesting to study a smaller synthetic setup where full-spectrum tracking is feasible and ZO EoS is still clearly visible. In this paper, however, we chose to focus on real datasets and practically meaningful neural-network experiments.
>
> **3. Clarification and correction of the proof of Theorem 4(c)**
>
> Thank you for catching this issue! As you pointed out, the current proof of Theorem 4(c) has two issues:
> - In Line 861, Lemma 3 is applied to $\mathcal M_i$, but the lemma is stated for $\frac1r\mathcal M_i$.
> - In Line 866, Lemma 2(b) is applied to for $\mathrm{Id}-\mathcal M_i$, but the lemma is stated for $\mathrm{Id}-\frac1r\mathcal M_i$.
>
> Both issues can be fixed cleanly by using the following scaled version of Lemma 3:
>
> **Lemma 3'** For any $c>0$, if $\rho(c\mathcal M_i)\ge 1$, then for every $\alpha>0$, there does not exist $W\succeq 0$ such that
> $$
> (\mathrm{Id}-c\mathcal M_i)(W)\succeq \alpha Q.
> $$
>
> This follows because the proof of Lemma 3 only uses cone preservation and the spectral-radius condition, both of which remain valid for $c\mathcal M_i$.
>
> Applying Lemma 3' with $c=\frac{1}{r}$,
> $$
> \rho\!\left(\frac{1}{r}\mathcal M_i\right)<1.
> $$
> Therefore,
> $$
> \left(\mathrm{Id}-\frac{1}{r}\mathcal M_i\right)^{-1}
> =\sum_{k=0}^\infty \left(\frac{1}{r}\mathcal M_i\right)^k
> $$
> is well defined by the Neumann series. Since $\mathcal M_i$ is $\mathbb S_+^2$-preserving, so is $\frac{1}{r}\mathcal M_i$, and hence its inverse is also $\mathbb S_+^2$-preserving. This justifies the step in Line 866 without invoking Lemma 2(b). We will revise the proof accordingly.
>
> **4. Intuition on the opposite effect of momentum in FO vs ZO**
>
> We share your intuition here. In FO-GDM, momentum damps deterministic oscillations near sharp directions, which enlarges the stable step-size region. In ZO-GDM, by contrast, momentum accumulates not only the gradient signal but also the random-direction estimator noise. Since ZO stability is governed by second moments, this extra accumulated noise makes the dynamics less stable, so increasing $\beta$ shrinks the stable region. We will add this intuition in the revision.
>
> **5. Minor remarks on presentation**
>
> Thank you very much for all of these detailed comments. We will revise them accordingly.
>
> ---
>
> Thank you again for your review. Does the above address your concerns? Please let us know if you have any further questions!

---

> > ### Author Rebuttal · Reviewer_thVy · 2026-04-01
> >
> > Thank you very much for addressing my questions with such detailed answers. My few (minor) concerns are addressed. I am very happy to maintain my positive score and recommend acceptance of this paper.

---

> > > ### Author Response · Authors · 2026-04-02
> > >
> > > We are delighted to hear that your concerns have been fully addressed! We thank the reviewer again for the thoughtful comments and valuable insights.

---

### Official Review · Reviewer_cNSB · 2026-03-10

**Soundness:** 3
**Presentation:** 3
**Significance:** 3
**Originality:** 3
**Overall Recommendation:** 4
**Confidence:** 4

**Summary:**

The paper studies the linear stability condition for zeroth-order (ZO) optimizers, which relates to the well-known edge of stability (EoS) phenomenon observed in training with first-order (FO) gradient-based algorithms. Using the quadratic approximation of the loss function, the authors derive the EoS conditions for the learning rate for the naive ZO method, ZO methods with momentum, and with fixed preconditioning. Experiments are conducted to verify the stability condition under various classical vision tasks. The phenomenon is similar to the EoS, while the "sharpness" is characterized mostly by the trace of the Hessian due to the ZO method. Moreover, momentum affects the stable regime differently from the FO methods, reducing the stability region rather than expanding it.

**Compliance With Llm Reviewing Policy:**

Affirmed.

**Final Justification:**

Fully addressed. I will keep the score

**Key Questions For Authors:**

1. Is it possible to have some central-flow-like non-rigorous derivation on how self-stabilization effects happen under the zeroth-order EoS?
2. How could the ZO EoS stability condition help design new ZO methods based on the current observation?

**Limitations:**

yes.

**Strengths And Weaknesses:**

Strength: Solid work on analyzing relatively precise linear stability conditions in ZO optimization, and characterizing the edge of stability (EoS) phenomenon along ZO optimization trajectories. The paper is well presented and easy to follow.

Specifically, the authors derive a quadratic approximation of the training trajectory, following the original EoS paper from Cohen et al., and identify the critical learning-rate interval. Compared to the original GD analysis, the calculation is involved and requires careful tracking of the iterations due to the introduction of momentum and the interplay between the $uu^\top$ terms and momentum. It is much more complicated than what one might expect at first glance. The derivations appear sound to me, though I only checked the proof without preconditioner. The experiments are able to verify the theoretical findings.

There are also some differences between the ZO EoS and the first-order EoS, which I believe are new insights. First (maybe a less surprising one, as noted in weakness) is that the zeroth-order method EoS threshold will depend on the trace of the Hessian. A more surprising one to me is the inverse connection of momentum and stability region, which is quite counterintuitive. Those are novel contributions to the best of my knowledge.

Weakness: Although there are some differences between ZO EoS and their first-order counterpart, I am still a bit concerned about the significance of this work. Since ZO optimization is often viewed as an approximation of first-order methods, it would be natural to observe similar phenomena. Though the stability condition here is dependent on the trace of Hessian instead of the top eigenvalue, the intuition behind this can be simply derived using the same techniques in EoS and adaptive EoS paper [1,2]: the trace term basically comes from terms like $uu^\top (u^\top Au)$ when $u$ is Gaussian used in the ZO method, which appears in the loss term after a step of update. Nevertheless, I think the results are still contributions to the community.

Moreover, given the careful and holistic characterization of EoS under GD & RMSProp by previous works (self-stabilization [4], central flow [3], etc.), the paper lacks a dynamical analysis of the mechanism of ZO EoS. I fully understand that it would be too complicated to analyze as in the full-batch settings of Cohen et al. and Damian et al., but I would expect some derivations, or at least discussions, along the lines of how ZO optimization self-stabilizes and operates along the zeroth-order central flow.

The experiments are limited to vision tasks. I believe the phenomenon is general enough in neural network optimization, but it would be nicer to include natural language tasks in the experiments.

[1] https://arxiv.org/abs/2103.00065 Gradient Descent on Neural Networks Typically Occurs at the Edge of Stability
[2] https://arxiv.org/abs/2207.14484 Adaptive Gradient Methods at the Edge of Stability
[3] https://arxiv.org/abs/2410.24206 Understanding Optimization in Deep Learning with Central Flows
[4] https://arxiv.org/abs/2209.15594 Self-Stabilization: The Implicit Bias of Gradient Descent at the Edge of Stability

---

> ### Author Rebuttal · Authors · 2026-03-31
>
> Thank you for your positive review of our work! We respond to the remaining concerns below.
>
> **1. Significance and relation to prior EoS work.**
>
> We respectfully disagree with the view that the main phenomenon is natural to expect simply because ZO is often regarded as an approximation of FO methods, or that it follows from the same techniques used in prior EoS/adaptive-EoS works.
>
> The main reason is that, even in the full-batch setting, ZO updates are still stochastic because of the random search directions. That changes the notion of stability. In the classical EoS literature, stability is studied through deterministic linearized dynamics. In our setting, the relevant notion is **mean-square linear stability**, which controls second moments of the iterates. This is the key difference: for ZO, the mean dynamics agree with the FO counterpart, but the interesting stability behavior only appears at the second-moment level. Our analysis makes this explicit and shows that the resulting stability depends on the **full Hessian spectrum**, not only on the top eigenvalue.
>
> The technical analysis is different for the same reason. Our proofs work through second-moment recursions, Gaussian fourth-moment calculations, and a cone-preserving covariance operator whose spectral radius characterizes mean-square stability. So while the motivation is clearly connected to prior EoS work, the framework and proof techniques are not the same.
>
> We therefore view the paper’s significance as twofold:
>
> (i) it identifies a genuinely different EoS mechanism for ZO, where stability is governed mainly by trace-based curvature quantities; and
>
> (ii) it introduces mean-square stability as a natural framework for studying ZO optimization dynamics.
>
> **2. Is a central-flow-like derivation possible for ZO EoS?**
>
> This is an interesting theoretical question! We think there is a natural central-flow derivation for ZO-GD.
>
> Let the ZO-GD trajectory be decomposed as
> $$w_t = w(t)+\delta_t,\quad\mathbb E[\delta_t]=0,\quad\mathbb E[\delta_t\delta_t^\top]=\Sigma(t).$$
> Then, the center $w(t)$ follows the central flow ODE:
> $$\frac{dw}{dt}=-\eta[\nabla f(w)+\frac12 \nabla_w\langle \nabla^2 f(w),\Sigma(t)\rangle].$$
> which also appears in the central flow for GD (see Eq.(19) in Cohen et al. 2025).
>
> The real difference is how the covariance $\Sigma(t)$ is determined. For ZO-GD, the second-moment dynamics are approximated by
> $$\frac{d\Sigma}{dt}=\mathcal T_w(\Sigma)-\Sigma+\eta^2 Q_w,$$
> where we denote $g:=\nabla f(w)$, $H:=\nabla^2 f(w)$,
> $$\mathcal T_w(\Sigma):=\Sigma-\eta(H\Sigma+\Sigma H)
> +\eta^2(2H\Sigma H+\mathrm{Tr}(H\Sigma H)I),$$
> and
> $$Q_w:=gg^\top+\|g\|^2I.$$
> At EOS, we impose  $\rho(\mathcal T_w)=1$. Assume the leading eigenvalue $1$ is simple, and let $V$ and $Y$ denote the corresponding right and left eigenmatrices:
> $$\mathcal T_w(V)=V,\quad\mathcal T_w^*(Y)=Y,\quad\langle Y,V\rangle=1.$$
> Decompose the covariance as
> $$\Sigma=sV+R,\quad\langle Y,R\rangle=0,$$
> where $s$ is the scalar amplitude of the critical mode and $R$ is the stable-complement component.
>
> Under a quasistationary approximation, $R(w)$ is determined by
> $$\Pi_V^\perp[\mathcal T_w(R)-R+\eta^2 Q_w]=0,$$
> where
> $$\Pi_V^\perp[X]:=X-\langle Y,X\rangle V.$$
> The scalar $s(w)$ is then fixed by the EoS constraint. Let
> $$\psi(w):=\rho(\mathcal T_w).$$
> Since the central flow is constrained to remain on the EOS manifold  $\psi(w)=1$, we impose
> $$\frac{d}{dt}\psi(w(t))=0.$$
> Using the central-flow ODE gives
> $$0=\langle \nabla \psi, g + \frac{1}{2}\nabla \langle H, \Sigma \rangle \rangle .$$
> Substituting $\Sigma=sV+R$ yields
> $$s(w)=-\frac{\langle \nabla \psi, g + \frac{1}{2}\nabla \langle H,R \rangle \rangle}{\langle \nabla \psi,\frac{1}{2}\nabla \langle H,V \rangle \rangle}.$$
> Thus, $R(w)$ and $s(w)$ determine $\Sigma=sV+R$, which in turn determines the central-flow ODE.
>
> **3. Experiments beyond vision.**
>
> Thank you for this suggestion. Since submission, we have also run experiments on sequence architectures on a synthetic sorting task, following the setup of the central flow paper. We observe the same qualitative mean-square EoS behavior there as well ([LSTM result](https://shorturl.at/sGMZr), [Mamba result](https://shorturl.at/6HX6r)). We will add these experiments in the revision.
>
> **4. Can the ZO EoS condition help design new ZO methods?**
>
> We believe so. One immediate implication of our theory is that larger step sizes in ZO-GD drive training toward solutions with smaller Hessian trace, meaning that they implicitly regularize average-direction sharpness, which may in turn improve generalization. This gives a concrete way to think about how the learning rate shapes the loss landscape in ZO training. More broadly, we think this perspective could help guide the design of improved ZO algorithms, and it seems like a promising direction for future work.
>
> ---
>
> Thank you again for your review. Does the above address your concerns? Please let us know if you have any further questions!

---

> > ### Author Rebuttal · Reviewer_cNSB · 2026-03-31
> >
> > Yes, that fully addressed my concern. The central flow analysis will further strengthen the paper. Thanks to the author for the detailed and prompt response.

---

> > > ### Author Response · Authors · 2026-04-02
> > >
> > > We are delighted to hear that your concerns have been fully addressed! We thank the reviewer again for the thoughtful comments and valuable insights.

---

### Official Review · Reviewer_MTHN · 2026-03-11

**Soundness:** 3
**Presentation:** 3
**Significance:** 2
**Originality:** 3
**Overall Recommendation:** 4
**Confidence:** 4

**Summary:**

This paper investigates the EoS (Edge of Stability) phenomenon in zeroth-order (ZO) optimization algorithms, particularly ZO-GD, ZO-GDM, and ZO-Adam. The authors demonstrate that the linear stability of ZO methods differs significantly from that of first-order (FO) methods: the stability of FO methods is determined by the largest eigenvalue of the Hessian matrix, while the mean-square stability of ZO methods is determined by the entire spectrum of the Hessian matrix. Through analysis of linear stability, the authors distinguish between linear stability in the mean and mean-square linear stability, and derive an explicit step-size condition for the mean-square linear stability of zeroth-order optimization algorithms. They then provide tractable upper and lower bounds based on the trace and largest eigenvalue of the Hessian matrix. Empirical studies on CNNs, ResNets, and Vision Transformers show that the full-batch ZO methods remain stable near the trace-dominated mean-square stability boundary.

**Compliance With Llm Reviewing Policy:**

Affirmed.

**Final Justification:**

I have carefully read the authors’ rebuttal, and I am satisfied that most of my concerns have been adequately addressed. I previously suggested including a small-scale experiment to precisely validate the proposed exact mean-square critical step size. The authors explained the practical difficulties of conducting such an experiment at this stage and indicated that they plan to pursue it in future work. Given this clarification, I am willing to maintain my original rating of weak accept.

**Key Questions For Authors:**

1. See weaknesses.

2. Can the intuition that momentum has opposite effects on zeroth-order and first-order methods be further explained through examples or experiments?

3. In large-scale experiments such as those involving Vision Transformers, can the authors report the exact number of model parameters and detail the practical computational overhead of obtaining the trace estimates? This procedure could require considerable computation.

**Limitations:**

yes

**Strengths And Weaknesses:**

Strengths

1. The theoretical section is highly rigorous. To maintain stability, being linearly stable in the mean is insufficient; variance must also be controlled, requiring mean-square linear stability.

2. Empirical verification closely matches theoretical limits. The experimental stability threshold $2/\eta$ consistently lies within the derived upper and lower bounds based on the Hessian trace under various frameworks. This observation provides empirical support for the linearized mean-square stability theory.

3. This paper is well-written and structured. The introduction clearly elucidates the differences between the FO and ZO stability mechanisms. The visualization of dynamic stability limits (lower bound, stability threshold, upper bound) makes complex theoretical results easy to understand and closely links theory with empirical phenomena.

Weaknesses

1. The theoretical analysis of Frozen ZO-Adam heavily relies on the commutativity assumption $PH=HP$. Although the authors verify this hypothesis by demonstrating during training that the relative commutator Frobenius norm is small, it remains a mathematically restrictive assumption because, in practice, the preconditioner $P$ and the Hessian matrix $H$ rarely share an exact eigenbasis.

2. Furthermore, the core theory and main experiments are limited to full-batch training; the impact of stochastic gradient noise in mini-batch training is only briefly explored.

3. The paper points out that increasing $\beta$ shrinks the stability region of Zeroth-Order Gradient Descent with Momentum (ZO-GDM) while expanding the stability region of the first-order model (FO-GDM). However, this viewpoint would be better clarified if it could be confirmed from a deeper dynamical systems perspective or through more careful experiments.

4. All of the authors' experiments are based on upper and lower bound controls, resulting in a certain range of approximation. The authors explain that this is because accurately solving for the mean-square critical step size is difficult; therefore, traces are used to provide estimates of the upper and lower bounds. It is encouraged that the authors rigorously solve for the exact mean-square critical step size on a smaller-scale experiment to confirm whether this quantity truly matches the experimental stability precisely.

---

> ### Author Rebuttal · Authors · 2026-03-31
>
> Thank you for your positive review of our work! We respond to the remaining concerns below.
>
> 1. **Commutativity assumption in Frozen ZO-Adam.**
>
> We agree that the assumption $PH=HP$ is restrictive and should be viewed as an approximation, not as an exact property of practical training. We use it because it makes the mean-square stability condition explicit and interpretable. Under commutativity, $P^{-1}H$ and $H$ share an eigenbasis, the second-moment dynamics decouple across eigendirections, and this leads to a closed-form criterion together with tractable trace/top-eigenvalue bounds in Theorem 3. Without commutativity, one can still characterize stability through the spectral radius of the induced covariance operator, but the recursion mixes different eigenspaces, and no similarly transparent formula seems available.
>
> We also want to note that this approximation is reasonably well supported in our experiments. In Appendix C.2, the relative commutator Frobenius norm drops quickly from about $0.8$-$0.9$ at initialization to below $0.05$, and then stays below $0.05$ during training. Empirically, ZO-Adam also tracks the predicted mean-square EoS boundary well across CNN, ResNet, and ViT experiments (Figures 2, 6, 7, and 8), so the approximation still appears to capture the relevant behavior.
>
> 2. **Limited to full-batch training.**
>
> We agree that including the mini-batch setting would make the paper more complete. In the current manuscript, we focus on the full-batch case as a clean starting point, where the only source of randomness is the ZO estimator itself. The mini-batch setting is substantially more complicated, since the second-moment dynamics would need to account for both estimator noise and data-subsampling noise. Even for first-order methods, the EoS picture for SGD is still much less understood than the full-batch case. To give some initial evidence beyond the scope of the theory, we include a preliminary mini-batch ZO-SGD experiment in Figure 5, which already shows behavior that is qualitatively different from full-batch ZO-GD. We will clarify this limitation and motivation more explicitly in the revision.
>
> 3. **Intuition behind the opposite effect of momentum in FO vs ZO.**
>
> We agree that this point would benefit from more intuition. A simple way to think about it is the following. In FO-GDM, momentum smooths deterministic oscillations near sharp directions, which allows a larger stable step size. In ZO-GDM, however, momentum accumulates not only the gradient signal but also the random-direction estimator noise. Since ZO stability is governed by second moments, that extra accumulated noise makes the dynamics less stable, so increasing $\beta$ shrinks the stable region. We will add this intuition in the revision.
>
> This picture is also consistent with both our theory and our experiments. On the theory side, Theorem 2 shows that for ZO-GDM the mean-square stability scale decreases with $\beta$, while for FO-GDM the classical stability threshold increases with $\beta$. On the empirical side, Figures 2, 6, and 7 show the same trend: as $\beta$ increases, ZO-GDM stabilizes at a lower trace level.
>
> 4. **Exact mean-square critical step size on smaller-scale models.**
>
> We agree that directly comparing the exact mean-square critical step size with experiments would make the paper stronger. Our theory does characterize this exact quantity once the full Hessian spectrum is available. The practical difficulty is that doing this during training requires repeatedly tracking the full spectrum, which is why we focus instead on trace- and top-eigenvalue-based bounds.
>
> To make this concrete, even our smallest neural-network experiment, the 3-layer CNN, has 30,570 parameters. While this is still small by deep-learning standards, tracking the exact Hessian spectrum throughout training is not practical under our current experimental protocol. It would certainly be interesting to study a smaller synthetic setting where full-spectrum tracking is feasible, and ZO EoS still appears clearly. In this paper, though, we chose to focus on real datasets and practically meaningful neural-network experiments.
>
> 5. **Computational overhead / experimental scale.**
>
> Thank you for raising this point. As described in Appendix B, we estimate the top eigenvalue using 50 steps of power iteration and the trace using 500 Hutchinson probes. All experiments were run on a single NVIDIA H100 GPU. The model sizes are: CNN: 30,570 parameters; ResNet: 294,842; ViT: 661,802. In our implementation, a ResNet run takes about 1-2 days, while a ViT run takes about 3-4 days. We will add these details in the revision as guidance for other researchers who might follow our experimental setup.
>
> ---
>
> Thank you again for your review. Does the above address your concerns? Please let us know if you have any further questions!

---

> > ### Author Rebuttal · Reviewer_MTHN · 2026-04-02
> >
> > Thank you for the detailed response. I understand the computational challenges of tracking the full Hessian spectrum, even for a 3-layer CNN with 30k parameters.
> >
> > However, my suggestion was aimed at rigorously verifying the theory's exactness, not evaluating a practically meaningful network. In a small toy setting — e.g., a tiny fully-connected network on synthetic data — tracking the full spectrum is entirely feasible. Showing that the theoretical mean-square critical step size precisely matches the empirical stability threshold in such a controlled setting would significantly strengthen the paper's rigor.
> >
> > Without this exact empirical verification, the current validation remains limited to bound estimations. Taking this into consideration, I am inclined to keep my score as a **Weak Accept** for the current version.

---

> > > ### Author Response · Authors · 2026-04-02
> > >
> > > Thank you for the continued engagement and for the valuable feedback. We agree that an exact empirical check in a very small controlled setting would strengthen the paper.
> > >
> > > Our reason for not including such an experiment is not that we disagree with its value, but that finding a tiny setting where ZO EoS appears clearly is itself nontrivial. In small synthetic problems, optimization often converges too quickly or behaves too simply, so the EoS regime may not appear in a clean or sustained way. More broadly, EoS is not a universal phenomenon that shows up automatically in arbitrary optimization problems; it is usually observed most clearly in realistic deep-learning settings. Even in the literature on standard GD, constructing toy settings where EoS can be clearly exhibited and then rigorously analyzed has itself been the main contribution of several follow-up papers. In that sense, identifying the right minimalist setting is not just a routine verification step, but can be a substantial task in its own right.
> > >
> > > In this paper, we chose to focus on two goals: developing the exact mean-square stability theory, and demonstrating the phenomenon in meaningful neural-network experiments. We agree that a controlled toy experiment matching the exact threshold would be a valuable complement, and we see it as a natural direction for future work.

---

### Decision · Program_Chairs · 2026-04-30

**Decision:**

Accept (regular)

**Comment:**

This paper studies the Edge of Stability (EoS) phenomenon in zeroth-order optimization algorithms. The reviewers acknowledge the rigorous theoretical analysis, the empirical analysis, the clear presentation, the significance and originality of the results, which highlights the specific EoS phenomenon occurring when using zeroth-order algorithms. Reviewers also provide some suggestions on presentation, and the experiments. The authors should incorporate these suggestions in their revision.